# Local-scale deposition of surface snow on the Greenland ice sheet

Alexandra M. Zuhr[1,2], Thomas Münch[1], Hans Christian Steen-Larsen[3], Maria Hörhold[4], and
Thomas Laepple[1,5]

[1]Alfred-Wegener-Institut Helmholtz Zentrum für Polar- und Meeresforschung, Research Unit Potsdam, Telegrafenberg A45, 14473 Potsdam, Germany
[2]University of Potsdam, Institute of Geosciences, Karl-Liebknecht-Str. 24-25, 14476 Potsdam-Golm, Germany
[3]Geophysical Institute, University of Bergen and Bjerknes Centre for Climate Research, Bergen, Norway
[4]Alfred-Wegener-Institut Helmholtz Zentrum für Polar- und Meeresforschung, Research Unit Bremerhaven, 27568 Bremerhaven, Germany
[5]University of Bremen, MARUM - Center for Marine Environmental Sciences and Faculty of Geosciences, 28334 Bremen, Germany

**Correspondence:** Alexandra M. Zuhr (alexandra.zuhr@awi.de)

**Abstract.** Ice cores from polar ice sheets and glaciers are an important climate archive. Snow layers, consecutively deposited and buried, contain climatic information from the time of their formation. However, particularly low-accumulation areas are characterised by temporally intermittent precipitation, which can be further redistributed after initial deposition, depending on the local surface features at different spatial scales. Therefore, the accumulation conditions at an ice core site influence the quantity and quality of the recorded climate signal in proxy records. This study aims to characterise the local accumulation patterns and the evolution of the snow height to describe the contribution of the snow (re-)deposition to the overall noise level in climate records from ice cores. To this end, we applied a Structure-from-Motion photogrammetry approach to generate near-daily elevation models of the surface snow for a 195 m$^2$ area in the vicinity of the deep drilling site of the East Greenland Ice Core Project in northeast Greenland. Based on the snow height information we derive snow height changes on a day-to-day basis throughout our observation period from May to August 2018 and find an average snow height increase by ∼11 cm. The spatial and temporal data set also allows an investigation of snow deposition versus depositional modifications. We observe irregular snow deposition and erosion causing uneven snow accumulation patterns, a removal of more than 60 % of the deposited snow, and a negative relationship between the initial snow height and the amount of accumulated snow. Furthermore, the surface roughness decreased by approximately a factor of two throughout the spring and summer season at our study site. Finally, our study shows that Structure-from-Motion is a relatively simple method to demonstrate the potential influences of depositional processes on proxy signals in snow and ice.

## 1   Introduction

Ice cores from polar ice sheets and glaciers are one of the most important climate archives. Physical and chemical characteristics preserved in the ice store information on past climatic conditions and are used as proxy data, for example, to reconstruct past temperatures (e.g., Dansgaard, 1964; Jouzel and Merlivat, 1984) or snow accumulation rates (e.g., Mosley-Thompson et al., 2001; Dethloff et al., 2002).

The accuracy and interpretability of reconstructed parameters depend on the understanding of the initial signal formation and the processes that potentially thereafter change the original signal imprinted in the deposited precipitation. Amongst these are local processes such as snow-air exchange processes, alteration of the isotopic composition (i.e. $\delta^{18}$O or $\delta$D) by e.g. diffusion, sublimation, vapor deposition or metamorphism (Steen-Larsen et al., 2014; Dadic et al., 2015; Ritter et al.), depositional losses of chemical compounds (Weller et al., 2004), local to regional processes such as the spatial variability in snowfall and wind-driven redeposition leading with the local topography to stratigraphic noise (Fisher et al., 1985; Münch et al., 2016), and larger scale processes such as precipitation intermittency (Persson et al., 2011).

One major obstacle is the apparent gap between precipitation - as determined from model approaches, reanalysis data, and remote sensing products - and the net snow accumulation at the local scale relevant for firn and ice core records. This gap is caused by processes such as snow erosion, drift, and redistribution which depend on the wind speed, wind direction, and duration of wind events, as well as on the conditions of the surface snow (Li and Pomeroy, 1997a,b; Sturm et al., 2001). While dunes and ripple marks are snow bedforms resulting from snow deposition by wind, sastrugi are the result of erosional processes at the snow surface and are very common at locations with high wind speeds (Filhol and Sturm, 2015; Kochanski et al., 2018). Furthermore, loose snow on top of consolidated features can easily be picked up, transported by wind, and redeposited at other locations which results in spatially variable snow accumulation (Fisher et al., 1985; Libois et al., 2014; Naaim-Bouvet et al., 2016), rendering wind an important meteorological parameter driving the changes in the observed snow surface (Albert and Hawley, 2002; Groot Zwaaftink et al., 2013). In order to understand the temporal and spatial variability of snow accumulation and to ascertain their contribution to accumulation intermittency and to the observed variability in proxy records (van der Veen and Bolzan, 1999; Ekaykin et al., 2016; Picard et al., 2019), quantification of the spatial and temporal snowfall events and of the changes in surface structure and surface roughness are therefore of crucial importance.

The acquisition of reliable snow height data is, however, still a challenge (Eisen et al., 2008; Chakra et al., 2019). Methods to measure the amount of snow accumulation include stake lines and farms, snow height sensors, remote sensing products (e.g., satellites, lidar and radar measurements, photogrammetry alone or in combination with Structure-from-Motion (SfM) approaches), as well as laser scanning approaches. Stake lines, grids, and farms are robust and low-cost ways to manually document snow height evolution (e.g., Kuhns et al., 1997; Mosley-Thompson et al., 1999; Schlosser et al., 2002); however, these methods require time and personnel in the field. Snow height sensors, often mounted next to an automatic weather station (AWS) (e.g., Steffen and Box, 2001; van de Wal et al., 2005), require less manual work, can provide measurements at high temporal resolution, but are restricted to a single point. Remote sensing products provide large spatial coverage up to several hundreds of kilometres with spatial resolutions of e.g. 0.7 m pixel size for laser altimeter systems (Herzfeld et al., 2021); however, their large spatial resolution is not suitable for small or local-scale studies (e.g., van der Veen and Bolzan, 1999; Rignot and Thomas, 2002; Arthern et al., 2006). To obtain snow height changes on the scale from centimetres to kilometres, various forms of SfM photogrammetry (Keutterling and Thomas, 2006; Westoby et al., 2012; Nolan et al., 2015; Basnet et al., 2016; Cimoli et al., 2017), laser scanners (Baltsavias et al., 2001; Picard et al., 2016, 2019), and large grids of snow stakes (Mosley-Thompson et al., 1999; Schlosser et al., 2002) are used. SfM is a widely used technique, also with applications in glaciology (e.g., Westoby et al., 2012; Chakra et al., 2019), and it can overcome some of the limitations of laser scanners, for

example, the wind disturbance by the fixed scanning tower as well as the limited range of the laser scan (Picard et al., 2016, 2019).

In this study, we apply a custom-made SfM photogrammetry approach to explore the snow accumulation behaviour for a study site in northeast Greenland next to the deep drilling site of the East Greenland Ice-Core Project (EGRIP). The overall aims of this study are (1) to show that our method reliably characterises the temporal and spatial pattern of snow erosion and accumulation, (2) to provide insights into the temporal and spatial changes of the surface structures, and (3) to investigate the effect of wind, subsequent snow erosion and transport on the internal layering of the upper snowpack and the resulting implications for climate proxy analyses.

## 2 Data and Methods

### 2.1 Study Site

Our study site is located next to the EGRIP camp site in northeast Greenland (75° 38' N, 36° W, 2708 m a.s.l., Fig. 1a) (Dahl-Jensen et al., 2019). The location has a mean annual temperature of -29 °C and is characterised by prevailing westerly winds (Madsen et al., 2019) with a mean wind direction of 252° during our observation period (Appendix Figs. A1 and A2). Accumulation rates in the vicinity of EGRIP are 13.9 cm w.eq. yr$^{-1}$ as estimated over a period of ∼5 years from 2011 to 2015 (using density data of the upper 2 m of the snowpack, Schaller et al., 2016), while shallow ice cores and geophysical surveys indicate between 10 and 12 cm w.eq. yr$^{-1}$ (Vallelonga et al., 2014; Karlsson et al., 2020). An AWS from the Program for the Monitoring of the Greenland Ice Sheet (PROMICE) (Fausto and van As, 2019) was installed in 2016 ∼500 m southeast of the camp (Fig. 1b) and provides meteorological data with a 10-minute resolution (Appendix Fig. A1).

### 2.2 SfM photogrammetry and snow height reconstruction

We apply a SfM photogrammetry approach to map the daily snow accumulation patterns by reconstructing the daily surface snow heights from digital images. To this end, we took images of the snow surface covering a 39 x 10 m rectangular area, with the long x-axis set up perpendicular to the main wind direction and the short y-axis pointing towards it (referred to as *Photogrammetry Area*; Fig. 1b). We set up 35 glass fiber sticks surrounding the area (30 sticks along the x-axis and five distributed on the surrounding edges, Fig. 1c) to provide absolute reference points for the snow height reconstructions. Furthermore, all sticks were levelled to the same relative height by using a theodolite.

Photos were taken almost daily from 16[th] of May to 1[st] of August, 2018 (77 days), mostly between 6 and 8 pm (local camp time, GMT-3) to ensure the best light contrast and similar light conditions on all photos (Nolan et al., 2015; Cimoli et al., 2017). No photos were taken on days with very cloudy or whiteout conditions, because these conditions do not allow a DEM generation from optical images. The photos were taken using a Sony $\alpha$ 6000 camera with a fixed lens of 20 mm focal length and a focal ratio of f/16. The ISO value was set to 100. These parameters were chosen to get as much contrast in the images as possible. The camera was mounted at a height of ∼ 1.6 m above ground on a setup consisting of a sledge, an ice core box, a

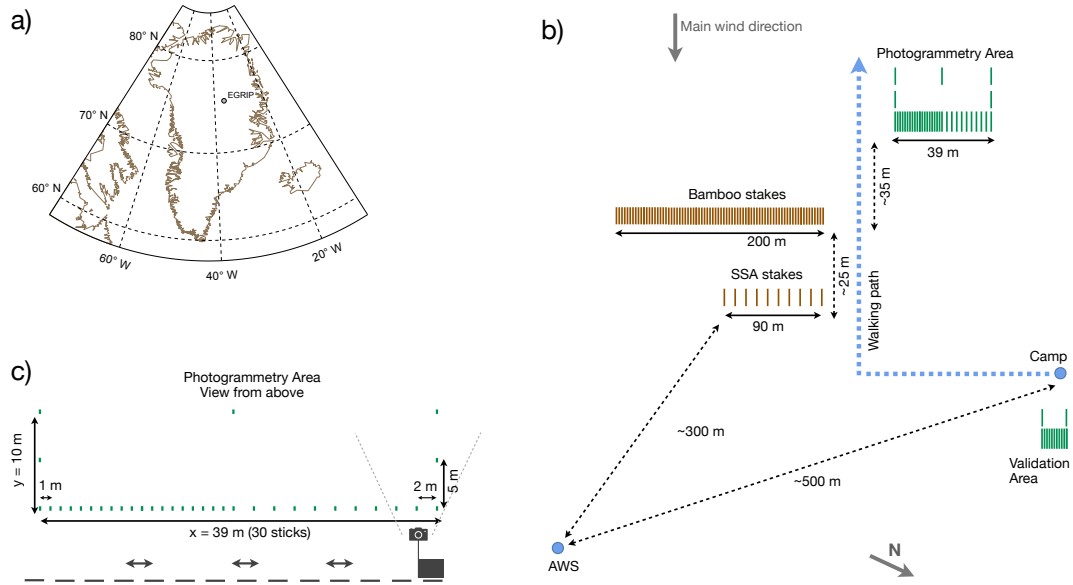

**Figure 1.** Overview maps for all relevant locations and transects in our study. a) Map of Greenland with the location of the EGRIP camp site in northeast Greenland. b) Schematic map of the EGRIP area including all relevant study sites. This area is approximately 200 m south of the deep drilling site at the EGRIP camp and about 300 m from the AWS. The map is not to scale. Data from the AWS, the Bamboo stakes, and the SSA transect (SSA = Specific Surface Area) are used for the comparison of snow height estimates. Data from the Validation Area are described in the Appendix B. c) Schematic illustration of the Photogrammetry area with respective distances. 30 glass fiber sticks were set along the walking line (x-axis), four sticks were positioned on the edges towards the main wind direction (y-axis), and one in the back of the study area. The sledge with the camera is located as is shown in Fig. 2. The approximate field of view of the camera is illustrated with grey lines.

plexiglass plate, and a metal pole (Fig. 2). During image acquisition, photos were taken every second using an automatic shutter control while the sledge was dragged on foot by a person along the downwind main side (Fig. 1, x-axis). This provided about
90 60 consecutive images with an overlap of ∼70 %. If less than 50 photos were available, no DEM could be generated because of insufficient overlap between successive images. We obtained an effective data set of 37 out of 77 days (48 %, Table A1) due to overcast conditions affecting the light contrast, the inability to detect surface structures, failures in the image processing, or insufficient overlap of consecutive photos.

We used the software AgiSoft PhotoScan Professional (Version 1.4.3 Software, 2018, retrieved from http://www.agisoft.
com/downloads/installer/) for the SfM workflow including the digital elevation model generation (hereafter referred to as DEM (digital elevation model), archived under https://doi.org/10.1594/PANGAEA.936082, Zuhr et al., 2021b). The generated DEMs have a resolution of 1 x 1 cm. For reliable geo-referencing, we manually added ground control points (GCPs) with known coordinates using the top of the glass fiber sticks (Figs. 1c, B2b and c) within Agisoft PhotoScan. The sticks at y = 10 m were, however, not visible in every daily data set and could not always be used as GCPs. Therefore, all 35 sticks were used

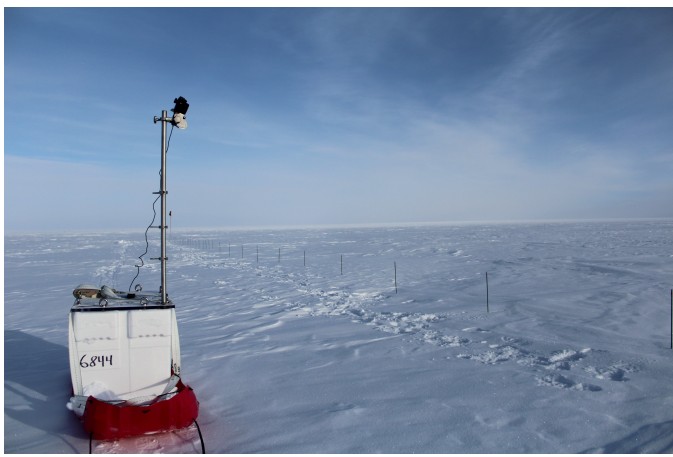

**Figure 2.** Camera setup for the image acquisition. The setup consists of a sledge, an ice core box, a plexiglass plate, a metal pole, and the camera used to take images of the Photogrammetry area. The sledge was dragged along the downwind main side of the study area, during which photos were taken using an automatic shutter control.

as GCPs if they were visible, otherwise the effective number of GCPs varied between 32 and 35. The absence of the GCPs at y = 10 m and the lower image quality might impair the height control in the back of the area. For further analyses, the study area was therefore restricted to y ≤ 5 m and thus a DEM area of $195 \, \text{m}^2$ (instead of $390 \, \text{m}^2$) to ensure constant data availability.

We evaluated our DEMs by analysing the trueness of our DEM-derived snow height estimates compared to reference heights, i.e., manually measured snow heights. For this, we set up a validation area with independent validation points within the area, avoiding using the actual study area for the validation purposes to minimise the disturbances on the snow height evolution in the study area. The detailed validation analyses are presented in the Appendix B with the main findings summarised here:

1. Data quality: We assessed the data quality and uncertainty by comparing DEM-derived snow heights around the stick positions to manually measured snow heights at the stick locations (Appendix B1, Table B1, Fig. B1). We derived the temporal and spatial uncertainty across all stick positions and all measurement days with manual and DEM data and found a mean difference of 0.2 cm, a standard deviation of 1.2 cm, and a root mean square error (RMSE) of 1.3 cm. This uncertainty applies to single points in time and space and will reduce when averaging over the quantities.

2. Sensitivity test on the number of GCPs: We tested the dependency of the number of used GCPs on the DEM uncertainty by using additional sticks in the validation area as an independent measure of the accuracy of DEM-derived snow height estimates (Appendix B2, Table B3). Different numbers of GCPs have a very small effect on the overall accuracy with mean differences between -0.3 and 0.1 cm, standard deviations up to 1.5 cm and RMSEs up to 1.5 cm.

3. Sensitivity test on the alignment of GCPs: We evaluated the accuracy of our DEM-derived snow height estimates to the alignment of the sticks itself by using the detailed information from the validation sticks (Appendix B2, Table B4). The

influence of misaligned sticks causes mean differences between -0.1 and 0.1 cm, standard deviations up to 1.0 cm and RMSEs up to 1.0 cm.

In summary, uncertainties from manually setting up the transect, distributing the GCP coordinates during the processing, as well as the uncertainty of the GCP alignment are small compared to the amplitude of snow height change throughout our observation period (11 cm on average). We therefore conclude that our elevation models provide reliable snow height estimates with a sufficient accuracy for the purpose of our study.

## 2.3   Additional snow height and snowfall data

Complementing the DEM-derived snow height data, four additional snow height evolution estimates are available with different temporal resolutions and spatial coverages (Table 1 and Fig. 1b): manual documentation of the relative snow heights from i) the glass fiber sticks in the photogrammetry area (*PT sticks*, archived under https://doi.pangaea.de/10.1594/PANGAEA.931124, Zuhr et al., 2021a), ii) a 200 m long transect with 200 wooden sticks with 1 m spacing (*Bamboo stakes*, archived under https://doi.pangaea.de/10.1594/PANGAEA.921855, Steen-Larsen, 2020a), iii) a 90 m long transect with 10 sticks and 10 m spacing

(*SSA stakes*, SSA = specific surface area, archived under https://doi.pangaea.de/10.1594/PANGAEA.921853, Steen-Larsen, 2020b), and iv) automatic snow height measurements from the sonic snow height sensor at the nearby AWS (*AWS PROMICE*, http://www.promice.dk). The SSA stakes as well as the Bamboo stakes were aligned in the same orientation as our study area. The high-resolution data from the AWS PROMICE were averaged to daily values.

**Table 1.** Snow height estimates around the EGRIP camp site. The temporal resolution, the spatial extent, and the distance relative to the study area are given. The manual measurement of the 30 sticks at our study area refers to the sticks along the x-axis, because these were measured more frequently than the remainder of the 35 sticks. Estimates form the Bamboo stakes and the SSA stakes are averages across 200 or 10 sticks with 1 or 10 m spacing, respectively. The single point high-resolution data from the AWS PROMICE were averaged to daily values. Locations are illustrated in the overview map (Fig. 1b).

| Name | Temporal resolution | Spatial extent | Distance (m) |
|---|---|---|---|
| DEMs | near-daily | 39 x 5 m | 0 |
| PT sticks | 3 days | 30 sticks, 39 m | 0 |
| Bamboo stakes | 3-5 days | 200 sticks, 200 m | $\sim 50$ |
| SSA stakes | daily | 10 sticks, 90 m | $\sim 70$ |
| AWS PROMICE | daily | single point | $\sim 370$ |

Both, snowfall and snowdrift can lead to an increase in snow height and a differentiation between these can be difficult
in the DEMs. During the study period, snowfall was manually documented when visual snowfall was observed or physical snowfall was collected (Appendix Table A1; snow collection setup described in Steen-Larsen et al., 2014). We use this simple, manual documentation of snowfall as well as the ERA5 snowfall product from the European Centre for Medium-Range

Weather Forecasts (ECMWF, 2017; Hersbach et al., 2020) to obtain information about the time of snowfall events. ERA5 is increasingly used and provides reliable near-surface variables over the Greenland Ice Sheet (Delhasse et al., 2020). The data were downloaded with an hourly resolution from the Climate Data Store (https://cds.climate.copernicus.eu) and summed up to daily values. If the manual documentation indicates snowfall and the DEM data show an increase in surface heights, we consider this as snowfall. This does, however, not exclude the possibility that snowdrift (i.e., mobilised snow by wind) may have also contributed to the increase.

In addition to the SfM photogrammetry study, a snow sampling study was carried out every third day at the windward side of each stick position along the long x-axis of the PT area (resulting data is not part of this study). Even though the positions were filled up with snow after sampling to avoid artificial surface structures and drift, we manually removed these areas in the DEM generation to minimise biased snow height estimates.

## 2.4 Estimation of surface roughness

Surface roughness is often used to describe and analyse the size of landforms and features with respect to a specific scale, and it is therefore a useful tool to investigate the variability of the snow surface in our study area (van der Veen et al., 2009; Grohmann et al., 2011; Veitinger et al., 2014). Here, we use the peak-to-peak amplitude of 2.5 m long, non-overlapping segments following the approach by Albert and Hawley (2002) and average the individual values to a representative surface roughness estimate $R$,

$$R = \frac{1}{n} \sum_{i=1}^{n} \left( h_{\max_i} - h_{\min_i} \right) \tag{1}$$

where $h_{\max_i}$ is the maximum and $h_{\min_i}$ the minimum snow height of an individual segment and where $n$ is the number of considered segments. We analyse the surface roughness perpendicular (along the x-axis) and parallel (along the y-axis) to the main wind direction (schematic overview in Appendix in Fig. D1). Both estimates are averaged across $n$ = ~3800 individual segments. This surface roughness estimate captures variations on spatial scales below 2.5 m. To account for the larger-scale undulations, we additionally compute the standard deviation of the entire DEM area after applying a spatial smoothing (using an isotropic Gaussian smoothing kernel with a standard deviation of 100 cm).

## 3 Results

### 3.1 Relative snow heights from digital elevation models

Each of the 37 DEMs (Fig. 3, Appendix Table A1, Zuhr et al., 2021b) represents a two dimensional map (39 x 5 m) of the relative snow height in the study area for the particular day. The zero-point was chosen arbitrarily to be at the bottom of the first GCP on the day of installation. All further snow heights are referenced to this zero-level.

On the first day of our observation period, 16[th] of May 2018 (hereafter we refer to the Day of Observation Period, *DOP*), the snow height varied from -10.5 cm to +11.3 cm, with a total amplitude of 21.8 cm (Fig. 3 top panel). Two pronounced features of elevated snow heights were elongated along the prevailing wind direction and located in x-direction from ∼12 m to ∼20 m

and around 32 m. Considering the higher wind speeds during the winter (Appendix Fig. A3), these bedforms present in our study area are presumably dunes resulting from snow erosional processes (Filhol and Sturm, 2015). Until the middle of our observation period (20[th] of June 2018, DOP 36), the snow height has generally increased with a maximum increase of 12 cm while the surface structures flattened (Fig. 3 second panel). At the end of our observation period (1[st] of August 2018, DOP 78), snow heights ranged from +2.6 cm to +16.4 cm, thus showing a reduced amplitude of 13.8 cm compared to DOP 1 (Fig. 3 third panel). The bedform morphology is still dominated by undulations, although they are no longer as dominant as in the beginning of the observation period.

## 3.2 Comparison of the mean temporal evolution of different snow height estimates

Over the season, the area affected by the manual snow sampling and leading to missing DEM values is increasing. Thus, we focus our main analyses on an averaged band from $y = 2.5$ to $y = 3$ m which remains largely unaffected by the disturbances from foot steps and snow sampling across the entire season, since it lies upwind of these activities (Fig. 3 red bar in the top panel). We refer to this band in the following as the *2.5 m-band*. By comparing the DEMs between the beginning and the end of our observation period, the change in snow height of the 2.5 m-band amounts to an overall but not homogeneous increase of ∼11 cm (black vertical line in Fig. 4a; 10 and 90 % quantiles are 7.4 and 14.8 cm, respectively).

To investigate the consistency between snow height estimates obtained by different methods and from different locations, we compare our DEM-derived snow height data to the other, independently obtained estimates (Table 1, Fig. 4a and b). Compared to our DEM-derived increase in snow height of ∼11 cm from DOP 1 to DOP 78 (top row in Table 2), the other measurements based on multiple sampling locations show increases from 8.5 to 10.9 cm, while the single snow height sensor mounted to the AWS PROMICE recorded an increase of only 5.8 cm. Part of the differences among the methods are due to the different time periods covered, since comparing the most common time interval yields a better agreement (lower row in Table 2). The remainder of the differences are expected due to the different spatial representativity of the measurement techniques. The individual pixel within our study site also show large spatial variability of snow height increase at the EGRIP site (see Appendix Fig. C1).

**Table 2.** Snow height changes (in cm) for the different snow height estimates (Table 1) throughout the observation period. Two different periods are considered to include first the entire observation period covered by the DEMs (i.e., DOP 1 to 78) and second to compare the most common time interval, especially for the PT sticks and the Bamboo stakes (temporal resolutions are mentioned in Table 1).

| Name | DEM: 2.5 m-band | PT sticks | Bamboo stakes | SSA stakes | AWS PROMICE |
|---|---|---|---|---|---|
| DOP | 1 - 78 | 5 - 78 | 10 - 75 | 2 - 78 | 1 - 78 |
| Change (cm) | 11 | 9.7 | 8.5 | 10.9 | 5.8 |
| DOP | 10 - 73 | 11 - 72 | 10 - 71 | 10 - 73 | 10 - 73 |
| Change (cm) | 10.3 | 10.7 | 8.5 | 10.9 | 7.6 |

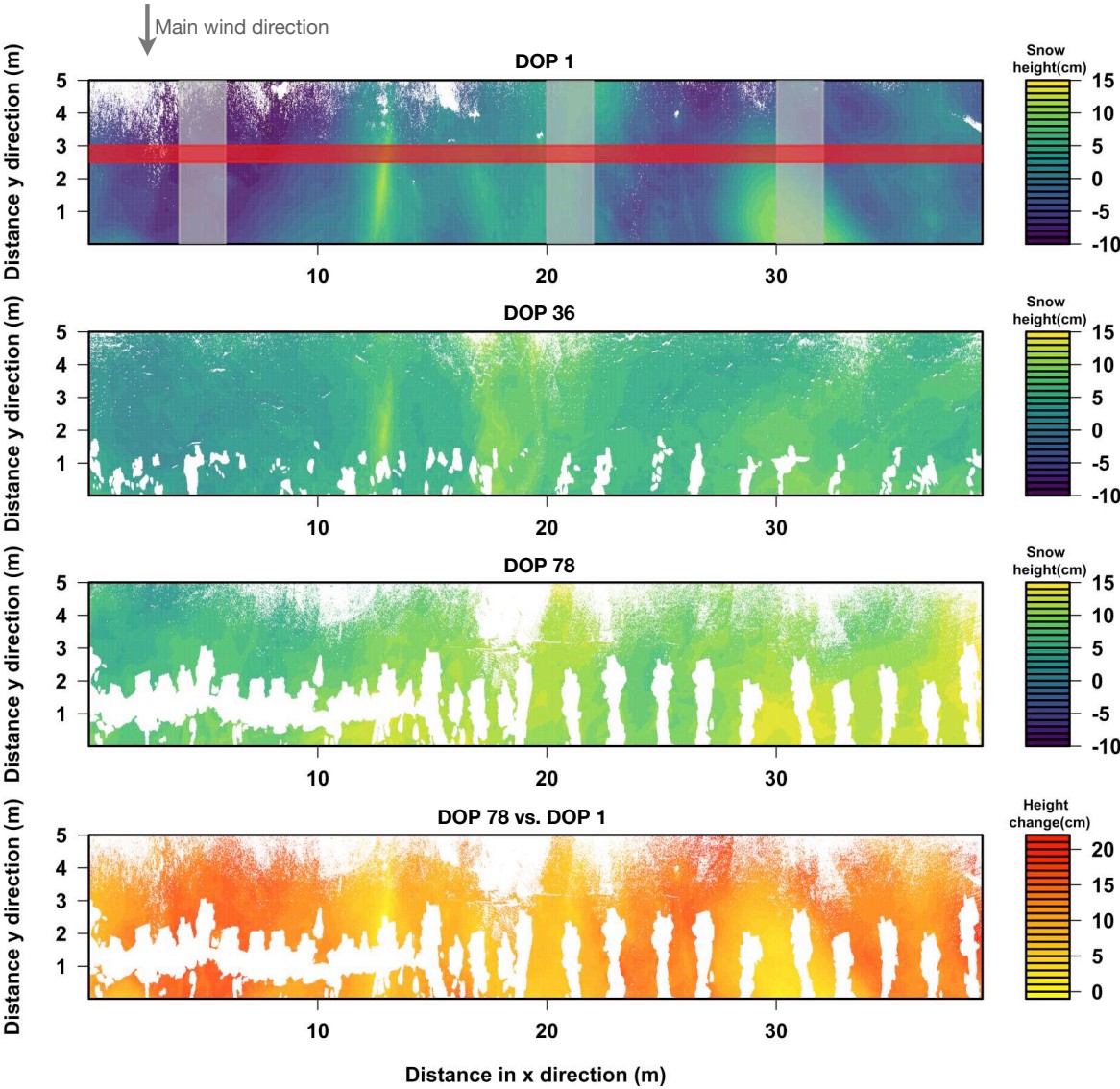

**Figure 3.** DEM-derived relative snow heights presented as two-dimensional maps (39 x 5 m). Shown are the snow heights for the Day of Observation Period 1 (16[th] of May 2018, DOP 1, upper panel), DOP 36 (20[th] of June 2018, second panel), and DOP 78 (1[st] of August 2018, third panel) as well as the change in snow height between DOP 1 and DOP 78 (fourth panel). Snow height estimates are given in cm relative to the snow height at x = 0 m and y = 0 m on the day of installation. The y-direction points towards the main wind direction. The red bar (in the top panel) indicates the band along the x-direction from y = 2.5 to y = 3 m (50 cm width), which is used to obtain average snow heights for each day for further analyses. The grey bars mark three subareas for the analyses in section 3.4 and Fig. 9. Missing data are shown as white areas and are caused either by a snow sampling performed in the same area (white spots close to the lower main line) or by insufficient image quality.

Despite the differences in the average snow height increase between the individual estimates (Fig. 4a), they agree on the overall temporal evolution within their uncertainty ranges (Fig. 4b). The development over time is characterised by a few individual, large events, such as the event around DOP 21 that led to an increase of ∼5 cm. Similar large snow height increases during single events have been reported in other studies (e.g., Libois et al., 2014).

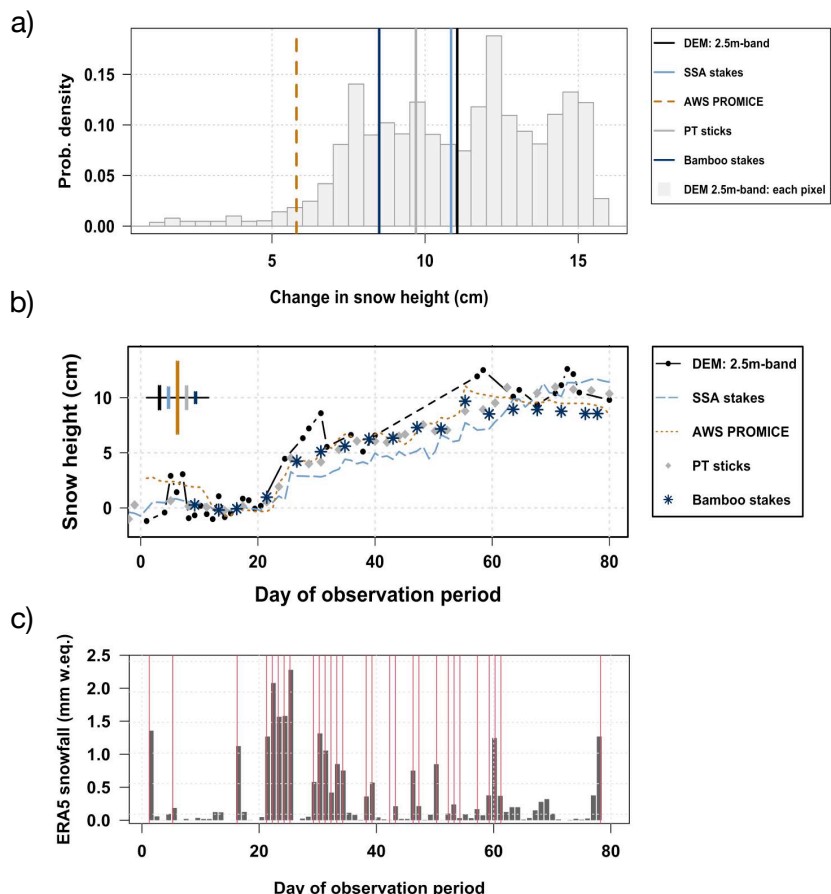

**Figure 4.** Evolution and changes of different snow height estimates throughout the observation period. a) Histogram of the DEM-derived change in snow height (grey bars) for every single pixel of the 2.5 m-band from DOP 1 to DOP 78 (Fig. 3 fourth panel) together with the mean snow height change (black vertical line) in the 2.5 m-band, as well as the mean snow height changes from other methods (vertical lines; Tables 1 and 2). Note that the latter estimates cover different spatial extents. b) Mean temporal evolution of snow height estimates throughout the observation period from the DEMs (2.5 m-band; black), the SSA stakes (light blue), the AWS PROMICE (gold), the PT Sticks (grey), and the Bamboo stakes (dark blue). For a direct comparison, each estimate is referred to its mean value from DOP 10 to DOP 20, which is defined as the zero level. Vertical bars include the uncertainty in cm (± one standard error) due to the limited spatial resolution of each method assuming a spatial decorrelation length of 5 m for our study site. The AWS PROMICE has the largest uncertainty because it is a single point measurement. c) The ERA5 snowfall product (grey bars) and manually documented snowfall during the observation period (red lines, see Appendix Table A1).

Manual documentation of snowfall (Appendix Table A1) contains only the information when snowfall occurred, but no indication on the amount. By contrast, all snow height estimates show only the total snow accumulation including depositional and erosional changes such as snowdrift and redistribution, but not the net amount of snowfall during a single event. To close the gap, we compare these data sets to the ERA5 snowfall product (Fig. 4c). The ERA5 data and the manual documentation agree in general well on the timing of snowfall; however, ERA5 also indicates snowfall on some days without manual notes.

This can have multiple reasons, e.g., no snowfall documentation during the night, snowfall which was directly blown away, or inaccuracies in ERA5. Comparing the ERA5 data with our snow height estimates, we find good agreement for all estimates concerning the strong event around DOP 21 (Fig. 4b). In addition, the many smaller events between DOP 30 and DOP 60 seem to constitute the gradual height increase in our observations.

### 3.3  Day-to-day variations and the erosion of fresh snowfall

To illustrate the nature of accumulation and erosion at the EGRIP site, we analyse the daily changes in the surface topography maps. Three examples are shown here (Fig. 5; the full maps are archived under https://doi.org/10.1594/PANGAEA.936099, Zuhr et al., 2021).

The first example is an overall increase of the snow height; thus a full layer of fresh snow. The snow height increased by $2.9 \pm 1.6$ cm (one standard deviation) and, except for a small area in the bottom left corner, the entire area received snow (DOP

4 to DOP 5, Fig. 5a). The complementary behaviour, that we will later discuss in more detail, is an erosive event from DOP 7 to DOP 8, characterised by a snow height decrease of $-2.9 \pm 2.3$ cm (Fig. 5b). A negative change in snow height can be caused by compaction or erosion. We consider erosion (i.e., physical depositional modifications) as the primary driver of negative snow height changes, neglecting compaction. Finally, some days show a patchy change in snow height as illustrated here for the evolution from DOP 35 to DOP 36 (net change of $-1.1 \pm 1.6$ cm; Fig. 5c).

The spatial and temporal evolution can be investigated in a more continuous way by averaging across the y-direction (in the 2.5 m-band) to show the snow height evolution against time. For DOP 1 to DOP 12 (Fig. 6, for all DEMs see Appendix Fig. C2), large temporal changes in the snow height are visible. In detail, the mean snow height increased by 4.1 cm from the first to the fifth day, consistent with manually documented snowfall. The ERA5 snowfall product agrees in the timing of snowfall (Fig. 4c, Appendix Table A1), but not regarding the amount (0.6 cm when converting the ERA5 snowfall from mm w.eq. using a

density of 290 kg m$^{-3}$). The DEM-derived increase is evenly distributed across the troughs and dunes. The subsequent decrease in snow height from DOP 7 to DOP 8 (-3.6 cm) is similarly homogeneous and, remarkably, exposes the initial surface structure from DOP 1 again. After a few days with rather patchy snow height changes (DOP 9 to DOP 11), the snow surface shows again the initial structure on DOP 12.

To study how systematic these erosion events are, we analyse how the correlation between the surface structures changes

over time (Fig. 7). For this, we provide for each day the correlation between the DEM on this day with all following DEMs, together with the evolution of the overall snow height. Erosive events can lead to an exposure of previous snow structures increasing the correlation between two DEMs while snowfall in combination with wind can cause inhomogeneous snow height increases which reduces the correlation. Indeed, the pattern of increasing and decreasing snow heights between DOP 4 and

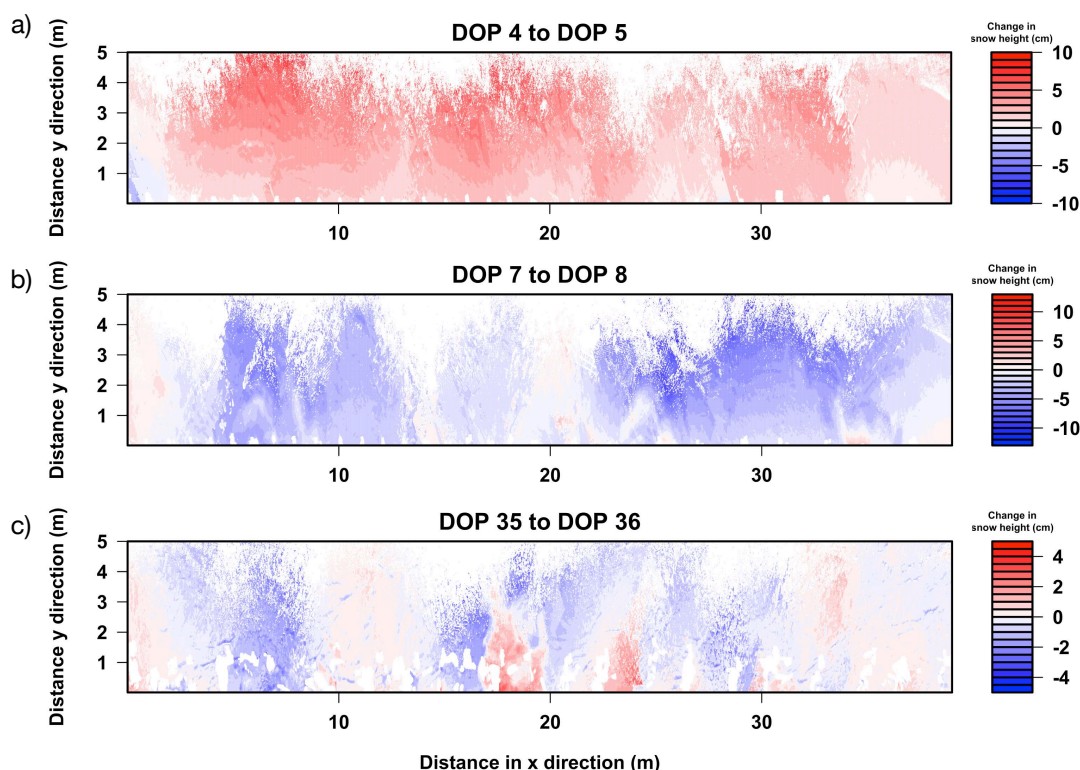

**Figure 5.** Day-to-day variations shown as the change in snow height (in cm) of the entire DEM-area for three periods: a) DOP 4 to DOP 5 shows an increase in the snow height for 78 % of all pixel, b) the change from DOP 7 to DOP 8 is dominated by snow erosion for 65 % of all pixel, and c) the snow height evolution from DOP 35 to DOP 36 is characterised by a spatially patchy pattern with positive and negative changes. For each panel, the zero-line indicates areas without changes in the snow height.

DOP 12 (Fig. 6) is reflected in varying correlations. Similar events occur between DOP 28 and DOP 31 as well as between
DOP 70 and DOP 73, highlighting the contribution of snow erosion on the overall snowpack build-up.

The study of Picard et al. (2019) compared the mean and standard deviation of the daily accumulation to snowfall and wind speed in order to investigate the influence of meteorological parameters on the snow accumulation for a study site on the East Antarctic Plateau, a region which receives only a quarter of the accumulation of the EGRIP site. Following their study, we derive the mean and standard deviation of the daily accumulation from our DEM data and compare these to the ERA5
snowfall and the wind speed from the AWS PROMICE to reproduce their Fig. 5. Surprisingly, we find patterns for our study site in northeast Greenland (Fig. 8) which are comparable to East Antarctica. Especially, we do not find significant relations between snowfall, wind speed, the mean daily accumulation and the standard deviation of the daily accumulation indicating that accumulation is not (solely) determined by snowfall or wind speed. Wind direction is not included in the analysis because the comparison did not indicate any relationship. The relation between the mean daily accumulation and the standard deviation

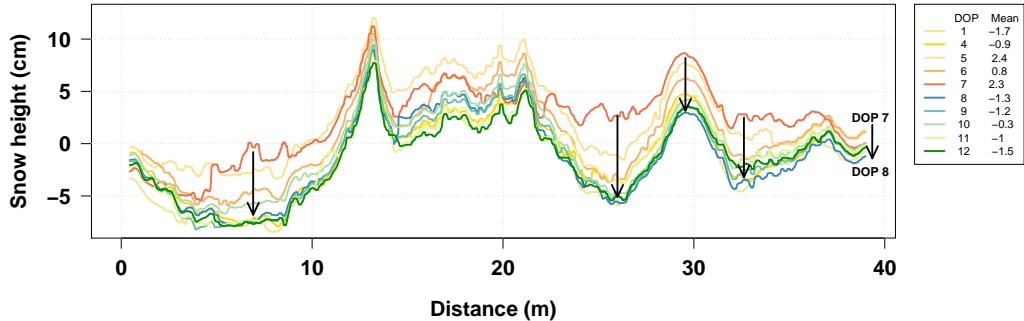

**Figure 6.** Temporal evolution of the relative horizontal snow height profiles from DOP 1 to DOP 12 (20-point running median, averaged 2.5 m-band). Different colours represent the different days as well as the respective mean snow heights in cm, both shown in the legend. Snowfall caused an overall snow height increase from DOP 1 to DOP 7, followed by an erosive event removing the new snow, and exposing the previous surface structure again. Arrows indicate the erosional decrease in the snow height from DOP 7 to DOP 8.

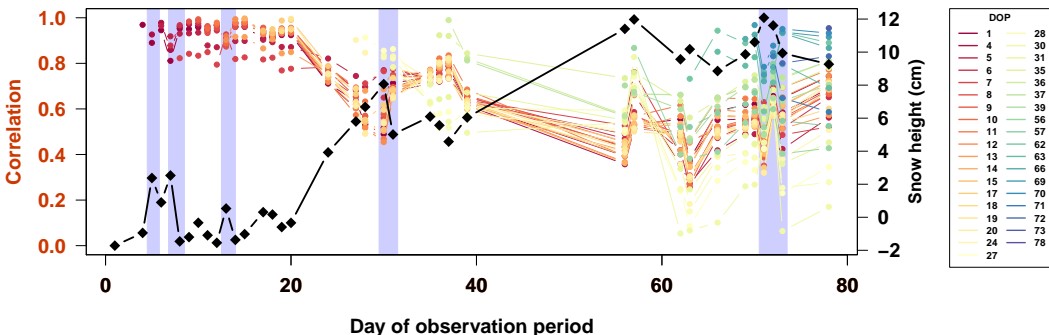

**Figure 7.** Correlation between the DEM-derived surface structures of a particular day and the surface structure on every following day (coloured points) as well as the overall snow height evolution (black diamonds). The colour-code indicates the DOP of the surface structure to which all subsequent structures are correlated to. The entire DEM area is considered for the correlation calculation. Vertical blue bars indicate an increase in the correlation and a decrease in the snow height. The second blue bar to the left shows the decrease in snow height from DOP 7 to DOP 8 which is illustrated in Fig. 6.

of daily accumulation (bottom right plot) is also similar to the pattern observed on the East Antarctic Plateau; however, this funnel-type pattern is reproducible with random data and therefore only a statistical feature with no meaningful information on the physical processes determining the accumulation conditions. However, in contrast to the study by Picard et al. (2019), we find a statistically significant correlation of -0.55 ($p < 0.01$, not accounting for autocorrelation) between wind speed and the daily accumulation (middle plot on the left in Fig. 8) which suggests that snow drift and erosion are important processes

determining the snow accumulation, with higher wind speeds increasing the potential for negative accumulation, i.e. snow erosion.

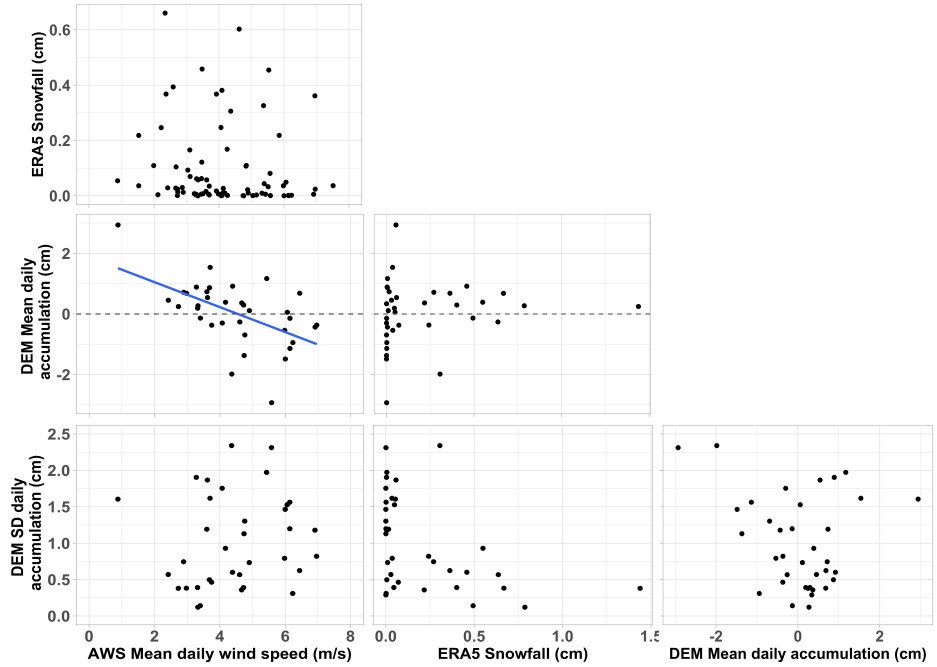

**Figure 8.** Following Fig. 5 in Picard et al. (2019), DEM-derived mean and standard deviation of the daily accumulation are compared to the daily wind speed from the AWS PROMICE and to the ERA5 snowfall product (converted to cm). During data gaps in the DEMs, the amount of snow accumulation was divided by the number of days to derive approximate daily accumulation. The accumulation conditions at the EGRIP site are largely remarkably similar to those on the East Antarctic Plateau, given the differences in accumulation rate, except for the relationship between mean daily wind speed and accumulation, which exhibits a negative correlation (r = -0.55; linear fit in blue) that is not apparent in Picard et al. (2019).

## 3.4 Flattening of the surface and changes in surface roughness

The surface snow becomes flatter towards the end of our observation period (Sect. 3.1, Fig. 3). This change from a heterogeneous to a homogeneous surface structure can be characterised in more detail by analysing the change in snow height between
DOP 1 and DOP 78 relative to the initial snow height on DOP 1 for the 2.5 m-band (Fig. 9a) which shows a variable structure at the beginning and a rather flat snow surface at the end of the observation period. In addition, we investigate three selected subareas with different initial surface structures (grey areas in Fig. 3 upper panel) to account for snow height changes parallel to the main wind direction. On DOP 1, the snow structures in these subareas (Fig. 9b) were characterised by a trough (dotted grey line), the top of a dune (blue), and an undulating surface (gold). While the first and second subareas received a homoge-
neous snow accumulation of ∼14 and ∼6 cm, respectively, the third subarea suggests a spatially variable snow accumulation such that the partial dune undulation present at DOP 1 has nearly vanished at DOP 78. Thus, despite the differences at the beginning of the observation period, all three subareas developed to similar relative snow heights on DOP 78 (Fig. 9b solid lines). Combining all four subareas, we find a strong negative correlation of -0.9 between the change in snow height and the

initial snow height (Fig. 9c) which indicates that areas which started with a relatively high snow height received less snow while areas with a comparably low initial snow height received more accumulation, a pattern that is also evident for the entire study area (see Appendix Fig. C3).

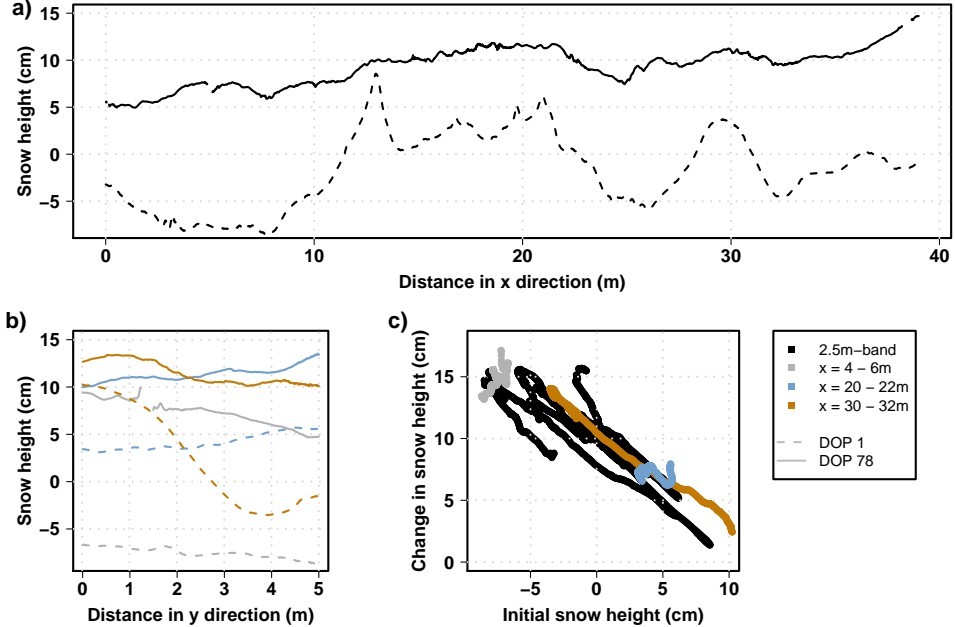

**Figure 9.** DEM-derived relative snow heights for 16[th] of May (DOP 1, dotted lines) and 1[st] of August, 2018 (DOP 78, solid lines) for four subareas of the study area: a) the 2.5 m-band perpendicular to the main wind direction; b) three subareas parallel to the main wind direction (grey: x = 4-6 m, blue: x = 20-22 m, and gold: x = 30-32 m) marked with grey bars in Fig. 3. c) Relationship between the initial snow height on DOP 1 and the change in snow height to DOP 78 for the subareas in a) and b). Note that the legend refers to all panels. The relation for each individual pixel in the study area is shown in the Appendix Fig. C3.

The change from a heterogeneous to a flatter surface structure is also reflected in a change of the surface roughness (Fig. 10). The temporal evolution of the roughness shows a consistent decrease from 4-5 cm to ~2 cm from DOP 1 to DOP 38 for both estimates parallel and perpendicular to the main wind direction (individual peak-to-peak amplitudes are in the Appendix Fig. D2). Interestingly, the same behaviour as found in the roughness (variations inside 2.5 m intervals) is also found for the large scale undulations. After the data gap of the DEM data between DOP 40 and DOP 56, the surface roughness estimates show an increase on DOP 56 followed by a successive decrease towards the end of the observation period. The surface roughness perpendicular to the main wind direction shows a larger amplitude than the surface roughness parallel to the main wind direction as well as the estimates from the entire study area. To investigate the evolution of the surface structure during the gap in the DEMs, we also include the standard deviation of the individual snow height readings from the PT sticks. This measure shows no changes in the strength of surface undulations between DOP 40 and DOP 56. Similarly to the DEM-derived roughness

estimates, the PT stick estimate records an increase in undulations around DOP 60 which indicates that the overall roughness increased at this point before decreasing again towards the end of the observation period.

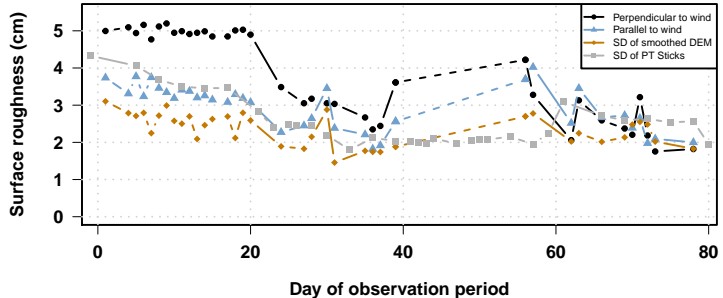

**Figure 10.** Temporal evolution of the surface roughness estimates throughout the observation period as described in section 2.4. The estimates parallel (blue) and perpendicular (black) to the main wind direction show an overall similar behaviour with the latter estimate suggesting higher roughness estimates at the beginning of the observation period. In addition, larger-scale undulations (gold) follow the overall pattern as well with an overall lower roughness estimate. The standard deviation across all PT stick measurements (grey) is used to fill the DEM data gap between DOP 40 and DOP 56 and follows the other estimates.

## 3.5   Implied internal structure of the snowpack

The snow accumulation characteristics presented in the previous sections suggest spatial variability in the snow accumulation which might influence the internal structure of the snowpack. The DEM-derived snow height data can be used to extract the internal structure of the snowpack along the x-axis which we illustrate as a two-dimensional view of the upper snow layers for the last day of our observation period (Fig. 11). A snow height increase between two consecutive DEMs is considered as a positive contribution to the snowpack and adds a new layer to the internal structure. A decrease in the snow height removes previously deposited layers, neglecting snow compaction.

The internal structure is characterised by the fact that only a limited number of days with different layer thicknesses are finally preserved. The prominent snowfall event on DOP 4, for example, is not recorded due to its subsequent erosion (previously discussed in Fig. 6). Other strong events, such as the large increase in snow height at DOP 21, result in nearly continuous layers, albeit with varying thicknesses. Different layer thicknesses transfer the heterogeneity of the initially rough surface to the internal structure of the snowpack with considerable variations at different locations: at $x = 12\,\mathrm{m}$, no snowfall events were recorded prior to DOP 56, while at $x = 8$ or $x = 24\,\mathrm{m}$, for example, larger amounts of snow were accumulated during the first weeks of the observation period. Based on the small number (about five to six) of distinct days in Fig. 11, we conclude that only a small number of events is actually recorded in the internal structure even though we know that more snowfall events occurred, which suggests that physical and chemical properties might vary at different locations within our study area.

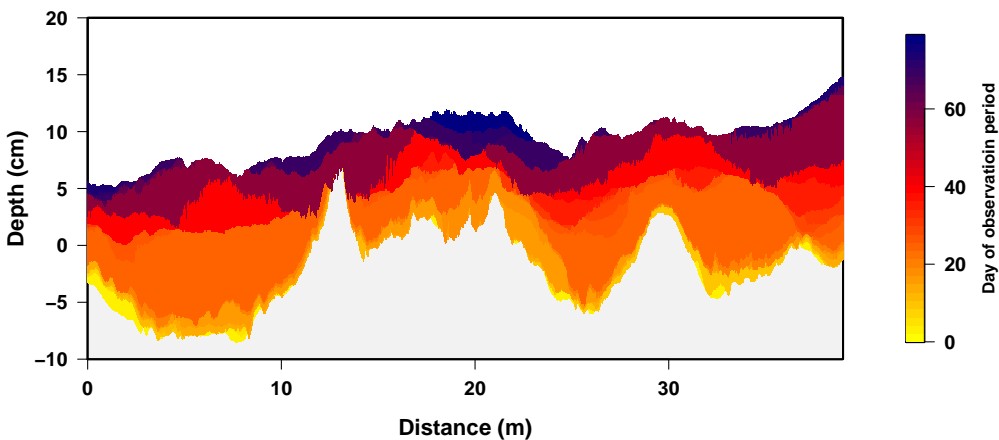

**Figure 11.** Two-dimensional view of the internal structure of our study area based on DEM-derived snow height variations along the 2.5 m-band for the last day of our observation period (DOP 78). Colours indicate the day of deposition during the season, namely when the DEM data recorded an increase in the snow height at the respective location. The grey background represents older snow and surface undulations prior to the first DEM on DOP 1. The long data gap between DOP 39 and DOP 56 does not cause an unrealistically thick snow layer which suggests that the temporal resolution of our data set does not affect the derived internal structure.

## 4   Discussion

We presented a three dimensional data set of snow heights and their variations derived from elevation models. The data show the temporal and spatial changes of the snow surface for the summer season 2018 at the EGRIP location, provide insights into accumulation conditions and allow a comparison to similar studies from e.g. Antarctica (Picard et al., 2019). In this section, we discuss the changes of surface structures, their implications for the interpretation of proxy data and the determination of accumulation estimates, and we assess the advantages and disadvantages of the SfM approach used.

### 4.1   Temporal and spatial changes of surface structures

The DEM-derived mean snow height increased by ∼11 cm in the 2.5 m-band (Fig. 4b). The total amount of snow input into the area, however, was more than 30 cm, if we consider only the positive contributions from precipitated and drifted snow (Fig. 12). The cumulative ERA5 snowfall results in ∼8 cm of net snowfall (Fig. 12). The DEM-derived net accumulation corresponds only to ∼35 % of the total amount of temporarily deposited snow while the ERA5 snowfall is only ∼24 %. Even though this could suggest that the ERA5 data might be biased towards drier conditions, we assume that both differences between the DEM-derived net accumulation and all positive contributions as well as between the DEM-derived estimate and the ERA5 snowfall are caused by substantial contributions of snowdrift and redistribution which emphasise their influence on the final snow accumulation during the observation period. Thus, the overall accumulation intermittency (Kuhns et al., 1997; Picard et al., 2019), presented here as the combination of snowfall and the intermittent depositional modifications, significantly

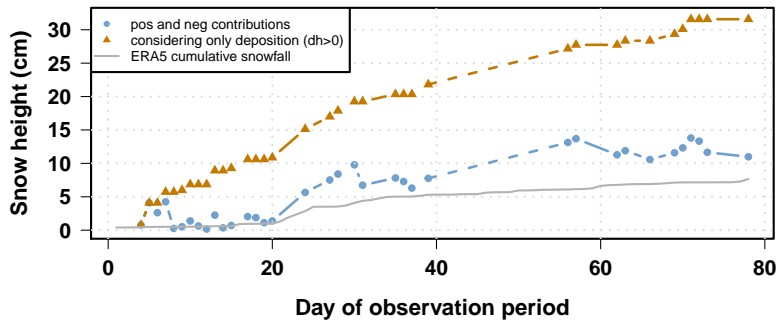

**Figure 12.** Cumulative snow height over the observation period from DEM-derived data and from the ERA5 reanalysis snowfall product. The DEM-derived snow height is shown for the two possibilities of i) counting both positive and negative contributions from one day to the next (blue), and ii) counting only the positive contributions (gold). The ERA5 snowfall (grey) is converted from mm w.eq. to cm assuming a mean snow density of 290 kg m$^{-3}$ which was obtained from daily density measurements of the top 2.5 cm at the stick positions of the SSA transect. Considering only the positive changes in the DEM-derived data accounts for deposition by snowfall or drift, but not for snow removal (e.g., erosion). This indicates that more than half of the snow that arrived at the study site was eroded and redistributed, and was thus transported out of our study area.

influences the recording of climate proxies in the snow and firn and can either cause the removal of snow from single spots and a transport to other locations, or, in turn, the deposition of snow from other locations at our study site.

Varying accumulation rates on a local scale can, especially in combination with wind, influence the surface structure. In our study area, two pronounced dunes were present at the beginning of the observation period which flattened towards the 310 end of the summer season (Figs. 3, 9a and 9b). The process of building up and wearing down of surface undulations has been reported for different locations on ice sheets in Greenland (e.g., Albert and Hawley, 2002) and Antarctica (e.g., Gow, 1965; Groot Zwaaftink et al., 2013; Laepple et al., 2016). The observed flattening in our study is characterised by a negative correlation between the initial snow height and the local accumulation (Fig. 9c), which likely also holds in the long term, i.e., between the accumulation from one year to the next. This suggests that local deviations from the mean accumulation rate 315 quickly average out over time as they cancel each other out (Fisher et al., 1985; Kuhns et al., 1997). Likewise, it explains why accumulation estimates from firn or ice cores that only sample one point in space but average across a large time-window, provide a good estimate of the regional accumulation rate, as already suggested by Kuhns et al. (1997) and van der Veen et al. (2009).

High wind speeds largely determine the growth or reduction of surface features and changes in the snow structure, in addition 320 to smaller contributions from e.g., temperature, humidity and metamorphism (e.g., Gow, 1965; Libois et al., 2014; Kochanski et al., 2018; Filhol and Sturm, 2019). Wind speed thresholds for drift and redistribution are 4 m s$^{-1}$ on average for a 100 hour period, or higher for a shorter time period (Groot Zwaaftink et al., 2013). At our study site, the winter wind speed is generally higher than the summer wind speed (Appendix Fig. A3) which can lead to an enhanced formation of dunes. Even during our observation period, the wind speed exceeded the defined threshold values on some days (Appendix Fig. A2) which can possibly

lead to snowdrift and might explain the observed erosion of entire snow layers and the exposure of previous surface structures (Figs. 6 and 7).

In the first half of our observation period, the surface roughness decreased from 4-5 to 2 cm (Fig. 10) followed by fluctuations around 2 cm. An increase of the surface roughness in winter, followed by a decrease in summer, is often attributed to seasonally changing wind speeds, with higher wind speeds in winter (e.g., Albert and Hawley, 2002). Our observed decrease in

surface roughness towards the summer is comparable to studies from van der Veen et al. (2009) covering a large area of central and northern parts of the Greenland ice sheet, from van der Veen and Bolzan (1999) for the location of the GRIP drilling site (close to Summit station, central Greenland), and to a study from Summit, Greenland, with a comparable spatial extent to our study (Albert and Hawley, 2002). van der Veen and Bolzan (1999) and van der Veen et al. (2009) used different mathematical expressions to calculate the surface roughness and found no reduction in surface roughness during the summer season.

However, Albert and Hawley (2002) used similar spatial scales and found patterns comparable to our results. The spatial and temporal variations in surface roughness estimates highlight the natural complexity of this parameter and the lack of clear information. Even though the considered time period in our study is too short to characterise the seasonal behaviour of surface roughness, our data set contributes to an increasing understanding of this parameter by suggesting a smoothing of the surface and a flattening of surface features towards the summer.

Short-lived and rapidly changing snow structures resulting from wind-driven redistribution have been reported for sites on the East Antarctic Plateau and influence the snow accumulation (Libois et al., 2014; Picard et al., 2019). In contrast to the persistently "patchy" accumulation characteristics in Antarctica and to alpine settings with much higher accumulation rates, we characterise the studied accumulation at the EGRIP location as "layer by layer": we observe alternating and layer-wise increases and decreases of the snow surface between DOP 1 and DOP 20, a filling of troughs (Figs. 3 and 9) and a reduction

of the surface roughness from DOP 20 to DOP 38 (Fig. 10) followed by accumulation in layers from DOP 56 to DOP 78 (Appendix C2). The ratio between the percentage of pixel with a positive and pixel with a negative DEM-derived day-to-day change in the snow height can be used to differentiate more quantitatively between "patchy" and "layer-by-layer" accumulation by defining an arbitrary threshold. In order to obtain a clearer result, we exclude changes between -0.5 and 0.5 cm. A ratio between 0.5 and 1.5 is considered to indicate "patchy" accumulation, i.e., about the same number of pixel show an increase

as a decrease, whereas a ratio below 0.5 or above 1.5 is seen as indicative for "layer-by-layer" accumulation, i.e., an overall increase or decrease of the snow height from one day to the next. Indeed, 12 out of 36 day-to-day changes suggest a ratio below 0.5 and 18 above a ratio of 1.5. Thus, following our simple metric, 30 out of 36 days confirm the proposed "layer-by-layer" accumulation.

Picard et al. (2019) tried to link snow accumulation to meteorological conditions but did not find a robust relationship. In

contrast, our DEM data show a significant negative correlation of -0.55 between the DEM-derived mean daily accumulation and the wind speed (Fig. 8), which suggests enhanced snow erosion during events with higher wind speeds, while lower wind speeds seem to be associated with more snow deposition, potentially leading to the deposition of redistributed snow during calmer conditions. Wind conditions are thus an important parameter for snow accumulation, which might also depend on the local accumulation rate and the amount of loose snow on top of a compacted snow surface. A key difference between the

study by Picard et al. (2019) and our analyses is the considered time period: while their study covers a period of three years including several winter and summer seasons, our data only cover a timespan of three months. Meteorological conditions and their influence on the snow accumulation might depend on the respective season which would not be represented in our study. Repeating our study and extending it to cover a longer time period would be necessary to more thoroughly investigate the seasonal behaviour of the snow surface and the influence of the wind on the snow accumulation.

## 4.2   Implications for the interpretation of proxy data

The large heterogeneity in accumulation, the depositional modifications of the surface snow and their impact on the internal snow structure (Figs. 4a and 11) imply that at sites with similar environmental conditions, parameters measured in a single firn core will not be representative on a seasonal scale (e.g., Ekaykin et al., 2002, 2016; Masson-Delmotte et al., 2015). This result is not only due to precipitation intermittency, a factor often considered in the interpretation of paleoclimate records
(Persson et al., 2011; Sime et al., 2011; Casado et al., 2018), but also due to the erosion of snow layers (Fig. 6). Erosion causes large differences between the total snow input and the net accumulation (Fig. 12) and creates a strong noise level due to an under-sampling of the continuous environmental signal (Casado et al., 2019). In addition, a singular event, such as a singular deposition of a proxy signal from a volcano or a biomass burning peak, might be missed in a local record. Stable water isotopologues, density data and accumulation rates show large interannual variations on local (e.g., Münch et al., 2017) and
also larger scales of e.g. 450 km in North Greenland (e.g. Schaller et al., 2016) which is likely the result of the accumulation heterogeneity.

Following studies on changes in the isotopic composition of surface snow (e.g., Casado et al., 2018; Hughes et al., 2021), we use theoretical snow profiles based on the accumulation history from our DEM data set (Fig. 11) to demonstrate the variability of snow ages at one location for several days (Fig. 13a) and at different locations for a single point in time (Fig. 13b). On a
single day, the derived mean snow age of a depth layer from 0 to 1 cm along our study area can vary by more than 20 days (Fig. 13b orange line). Even though we cannot distinguish between freshly precipitated and eroded or drifted snow in our data set, the variability within these theoretical snow profiles gives insight into the heterogeneous internal structure of the upper snowpack and shows the uneven snow accumulation at the EGRIP camp site and very likely also at other sites. Therefore, individual samples are not representative and should generally be avoided in favour of sampling multiple profiles.

Further long-term observations of the precipitation vs. accumulation statistics (Picard et al., 2019) and spatial studies of the signal recorded in snow and firn (Münch et al., 2016, 2017) are needed to better constrain the effect of both precipitation and accumulation intermittency on the preserved climatic information and to allow for a more reliable interpretation of proxy data from firn and ice cores. Compared to single point measurements, our spatial data set has the advantage of being better able to evaluate the redistribution and final settlement of snow. However, determining the origin and composition, e.g., the
homogeneity, of drifted snow and associated imprinted climatic signals, is essential but still challenging. Measuring the proxy signal at different stages during the deposition process, i.e., freshly precipitated snow, surface snow during vapour exchange with the atmosphere, drifted snow, and buried snow (as exemplary shown in Fig. 13), and combining these data with DEM-

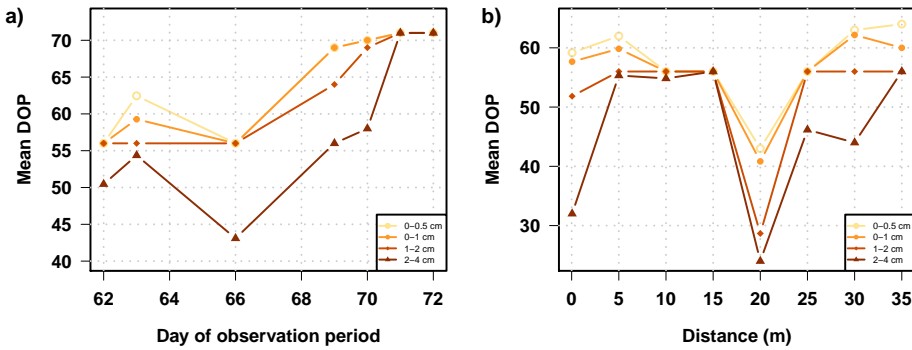

**Figure 13.** Theoretical temporal and spatial sampling of different depth intervals (0-0.5 cm, 0-1 cm, 1-2 cm and 2-4 cm) of the internal snow structure along the 2.5 m-band. a) Temporal sampling for ten consecutive days at x = 12 m. b) Spatial sampling at eight positions with 5 m spacing along the 2.5 m-band on DOP 69. The y-axes in a) and b) represent the average day of snow accumulation for the respective depth interval.

derived snow height information, will help to close the gap between accumulation intermittency and the preserved climatic information.

### 4.3 Implications for the measurement of snow accumulation

Typically, a snow height sensor integrated in an AWS delivers high temporal resolution data for only a single point and measures the accumulation at one specific location on an ice sheet. Our results show that at least at our study site, such a single point measurement would not deliver spatially representative information on a seasonal timescale. If only a single point in our study area was chosen, it would result in a snow accumulation estimate that could range from 7.4 to 14.8 cm (10 and 90 % quantiles, Fig. 4a). The AWS PROMICE estimate is at the lower end of this range (5.8 or 7.6 cm depending on the selected time period, Table 2) and significantly deviates from the DEM-derived average snow height change of ∼11 or 10.3 cm, respectively.

Accumulation estimates from snow stake farms and grids are averaged over multiple sites and are thus more representative (Kuhns et al., 1997; Eisen et al., 2008), but the remaining uncertainty is expected to depend on the number and the spacing of the stakes (Laepple et al., 2016; Münch et al., 2016). We can test this dependency based on our spatio-temporal data set. To this end, we use the DEM-derived snow height data at y = 2.5 m, identify each DEM pixel along this line with a possible snow stake position, and simulate different setups of snow stake samplings with varying stake numbers and spacings. For this, we chose different numbers of stakes between one and more than 200 sampling points with spacings between 10 cm and 10 m (Fig. 14). Depending on the chosen number of stakes and spacing, we determine all possibilities of positioning the stakes along the line at y = 2.5 m. Each possibility yields an accumulation estimate from averaging across the snow height changes at the stake positions, and we calculate the RMSE between the accumulation estimates from all possibilities and the reference accumulation estimate, i.e., the snow height change from averaging across all positions along the y = 2.5 m line.

The RMSE of the simulated snow stake accumulation estimates shows a clear dependency on the number of stakes and the choice of the distance between them (Fig. 14). Averaging ten snow stakes measurements with a one metre distance results in a similar error on the accumulation estimate as using only two stakes with a larger distance of 5 m, while an even larger spacing
does not further reduce the estimation error. This effect can be explained by the typical size of the surface structures at our study site, which are on the scale of several metres. Sampling the same surface feature multiple times does not increase the representativeness of the accumulation estimate, whereas sampling points farther apart to avoid the same feature contain more independent information. Thus, for study sites similar to EGRIP, stake setups for reliable snow accumulation estimates (RMSE <1 cm) could for example consist either of 25 stakes with 1 m distance or of only seven stakes with 5 m distance, significantly
reducing the workload.

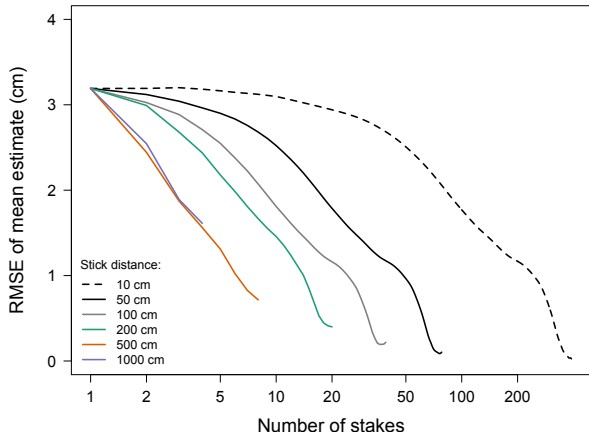

**Figure 14.** The uncertainty of the estimated mean snow height change as a function of the number of sampling points ("stakes") and the distance between them. The mean snow height change is calculated from the DEM data at $y = 2.5$ m for all possible sequences along the x-direction which consist of N sampling points with a given distance from one point to the next. The figure shows the RMSE between the DEM-derived mean snow height change of the respective sequences and the mean snow height change as calculated using all available sampling points.

## 4.4 SfM photogrammetry as an efficient surface snow monitoring tool

We showed that our SfM photogrammetry approach delivers reliable snow height information with an accuracy of $\sim$1.3 cm. The method can be used to characterise the spatio-temporal snow evolution on a spatial decimetre to 100 m scale. Our setup has several advantages in contrast to alternative approaches. Compared to single point measurements, we benefit from spatial
information encompassing an area of 195 m$^2$ which is easily scalable. Previous laser scanner studies covered areas of only 40 m$^2$ (Picard et al., 2016) and 110 m$^2$ (Picard et al., 2019) which can only be extended by placing the laser higher above the ground. However, a laser scanner itself is a high obstacle which can influence the snow redistribution and can thus affect the natural snow accumulation conditions. Our approach also offers the flexibility of repeatedly covering a spatial scale with

specific desired dimensions (e.g., an area with a length of 100 m) and orientations. In contrast, this is not possible for a laser scanner that is fixed to one position with a specific radius, or for manual point measurements, sonic snow height sensors or ground penetrating radar (Basnet et al., 2016; Cimoli et al., 2017). In our specific study, we had to reduce the analysed area by removing parts influenced by a parallel snow sampling campaign, but this can be avoided if one is only interested in the DEM evolution.

Furthermore, our approach does not require expensive equipment, as all the necessary items for image acquisition are commercially available. The method can be easily operated in remote areas and the logistical effort is low. It does not require a permanent power supply, which can be a limiting factor for laser scanners and snow height sensors. No specific training for users is needed as is required for airborne studies with aircraft (e.g., Baltsavias et al., 2001), drones (e.g., Hawley and Millstein, 2019) or LiDAR operations (e.g., Deems et al., 2013). Even though our approach is limited by light availability and visual contrasts, which is also reported in many studies (e.g., Nolan et al., 2015; Harder et al., 2016; Cimoli et al., 2017), it has the advantage of being very easy to operate and that it can be used at other study sites without great effort. However, the approach requires a human operator which can possibly limit our method, especially if an application is planned for a longer time period or year-round, and it also needs a mobile base (here, a sledge) to take multiple images along the area of interest. Using less images or stationary cameras (e.g. two fixed cameras) would probably result in a very small surveyed area since the field of view of one image covers only 3-5 m width at the baseline of the study area. We also had to restrict our analysed area to 5 m towards the wind direction due to insufficient image quality and lack of data points.

Besides the field work, our method requires the use of a software with a graphical user interface, manual work of setting the GCPs during the post-processing and a computer with a strong GPU unit. All data, i.e., the photos and the DEMs, are less than 1 TB. The post-processing time and the effort to generate a DEM are less than four hours per DEM and are comparable or even less than reported by other studies (e.g., James and Robson, 2012).

Since the presented setup was used for the first time for this study, days with missing DEMs were the result of human errors, e.g. insufficient number of images or poor overlap between consecutive photos. To improve future studies, we suggest to use an infrared filter to enhance the image quality and facilitate data acquisition even during cloudy and bad weather conditions (Bühler et al., 2015; Adams et al., 2018). Furthermore, placing the camera higher above the ground could enhance the spatial coverage (Picard et al., 2019). Finally, human errors can be reduced by providing a detailed manual for the data acquisition.

# 5 Conclusions

We presented high-resolution near-daily elevation models monitoring the surface snow evolution over a three-month period (May to August 2018) from the EGRIP campsite in northeast Greenland using a novel SfM photogrammetry approach. Comparing the DEM-derived snow height evolution to other snow height estimates from single or multi-point measurements showed an overall agreement on the general snow height increase of about 11 cm. The comparison emphasised the natural spatial variability of the snow accumulation on a local scale as well as its non-linear and event-driven character. Based on the observed

spatial accumulation field, we recommend that a stake-setup to reliably derive snow height estimates should consist of either 25 sticks with 1 m distance or seven sticks with 5 m distance for locations similar to EGRIP.

Day-to-day changes in the observed snow height provided detailed insights on snowfall and erosion which are essential processes that shape the surface structure and contribute to the internal snowpack structure. The spatial information on wind-driven snow erosion allowed us to showcase the removal of entire snowfall layers which caused an exposure of previous surface structures. The inhomogeneous snow accumulation within our study area led to a flattening of the snow surface and a reduction in surface roughness from ~4-5 to 2 cm as a result of a negative correlation between the initial snow height and the amount of accumulated snow.

Based on the daily snow height information, we simulated the internal snowpack structure for our study area. Main features are the spatially and temporally varying layer thicknesses and the complete absence of snow layers for specific time periods. Extracting single profiles (in space or in time) from this internal layering illustrates the expected variability in proxy data. Averaging samples from several locations is therefore suggested to reduce the local noise and to receive a representative signal.

Proxy data from ice cores are typically interpreted as precipitation-weighted signals. However, we showed that there are significant differences between precipitation and accumulation and that depositional modifications considerably change the structure of the surface snow. Investigating the dependency of proxy signals on the surface structures and on the general depositional processes leading to the signal imprint at different locations would therefore enhance the understanding and interpretability of proxy records. Combining snow height information, as provided by our study, with proxy data from the same area would be helpful to determine the influence of the internal snowpack heterogeneity on reconstructions from firn and ice cores.

*Data availability.* All snow height information data are available in the PANGAEA data base (https://www.pangaea.de): the photogrammetry SfM data under https://doi.org/10.1594/PANGAEA.923418, derived DEMs under https://doi.org/10.1594/PANGAEA.936082, DEM day-to-day changes under https://doi.org/10.1594/PANGAEA.936099, the PT stick data under https://doi.pangaea.de/10.1594/PANGAEA.931124, the SSA stake data under https://doi.pangaea.de/10.1594/PANGAEA.921853, and the Bamboo stake data under https://doi.pangaea.de/10.1594/PANGAEA.921855. Data from the Programme for Monitoring of the Greenland Ice Sheet (PROMICE) were provided by the Geological Survey of Denmark and Greenland (GEUS) at http://www.promice.dk.

**Appendix A: Additional meteorological and photogrammetric information**

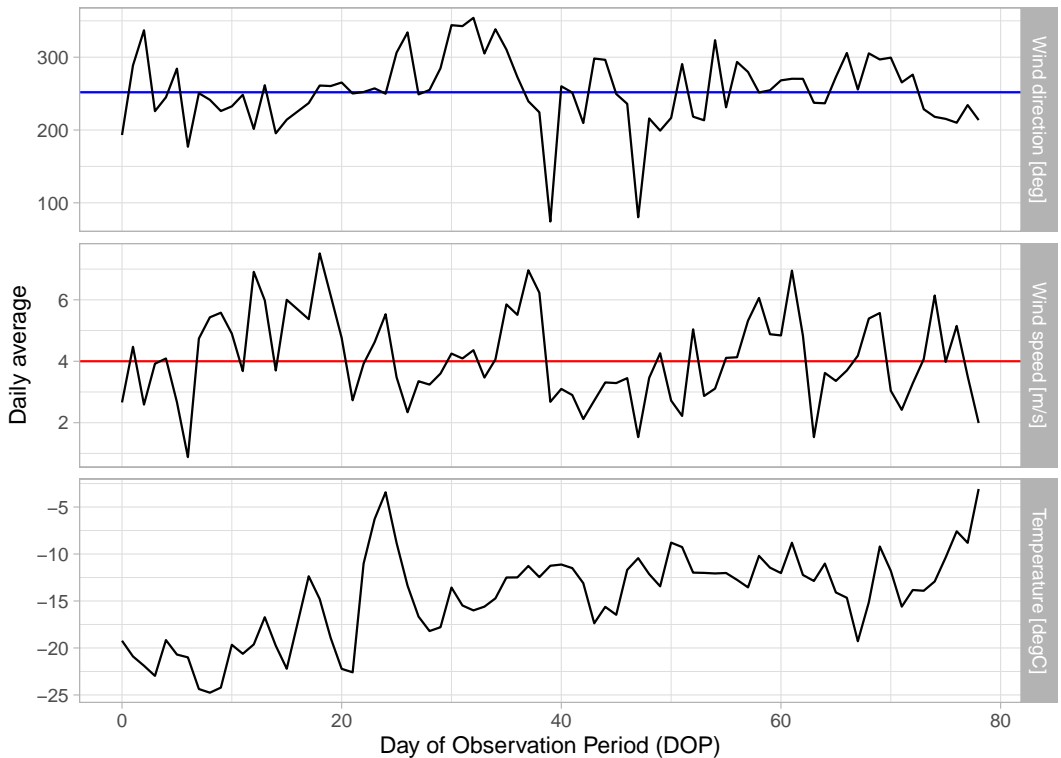

**Figure A1.** Daily averages of meteorological parameters, i.e., wind direction in [°], wind speed in [m s$^{-1}$] and air temperature in [°C], measured at 2 m height at the AWS PROMICE for the observation period from 16[th] of May (DOP 1) to 1[st] of August 2018 (DOP 78). In the first panel, the horizontal blue line indicates the average wind direction of 252° during the observation period. The horizontal red line in the second panel marks the threshold wind speed of 4 m s$^{-1}$. Snow transport is enabled when the threshold is exceeded for a 100 hour average (Groot Zwaaftink et al., 2013).

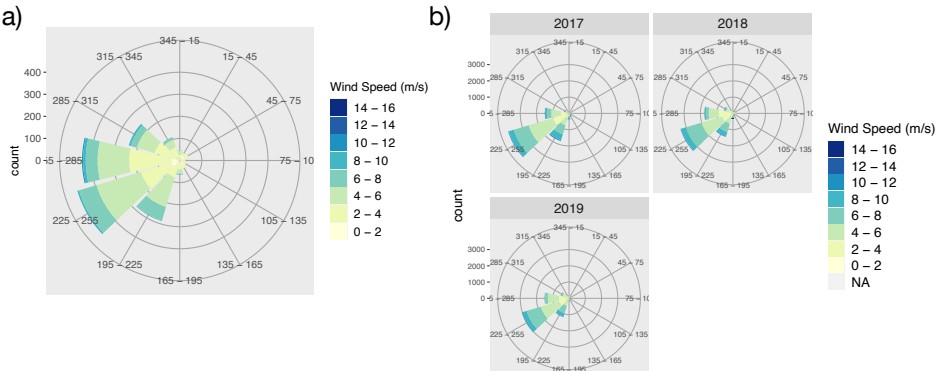

**Figure A2.** Wind characteristics (speed and direction) for the EGRIP camp site recorded by the PROMICE AWS for a) the observation period (16[th] of May to 1[st] of August 2018), and b) the years 2017, 2018 and 2019, respectively.

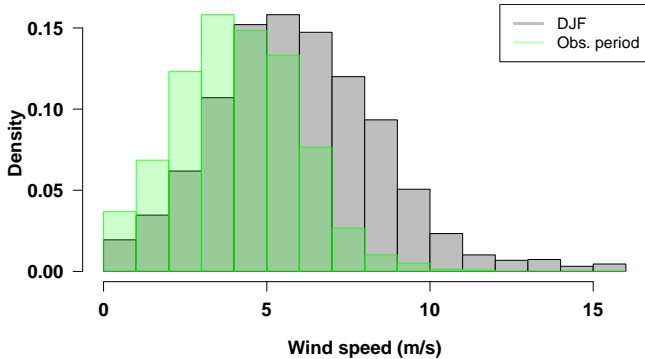

**Figure A3.** Hourly wind speed data from the PROMICE AWS for two different time periods. Wind speeds during the winter months of 2017 to 2019 (December, January, February; DJF; grey) are compared to wind speeds during our observation period (green). The winter months are characterised by higher wind speeds with a mean of ~6 m s[-1] while the average wind speed during our observation period was 4.1 m s[-1].

**Table A1.** Detailed information on the fieldwork campaign, including Day of Observation Period (DOP), Day Of Year (DOY), the date, the availability of a DEM, manual snow height measurements at the PT sticks (PT), and manually documented snowfall. We refer to the DOP in the text, figures and tables.

| DOP | DOY | Date | DEM | PT | Snowfall | DOP | DOY | Date | DEM | PT | Snowfall |
|---|---|---|---|---|---|---|---|---|---|---|---|
| 1 | 136 | 16.05.2018 | x | | x | 40 | 175 | 24.06.2018 | | | |
| 2 | 137 | 17.05.2018 | | | | 41 | 176 | 25.06.2018 | | x | |
| 3 | 138 | 18.05.2018 | | | | 42 | 177 | 26.06.2018 | | x | x |
| 4 | 139 | 19.05.2018 | x | | | 43 | 178 | 27.06.2018 | | x | x |
| 5 | 140 | 20.05.2018 | x | x | x | 44 | 179 | 28.06.2018 | | x | |
| 6 | 141 | 21.05.2018 | x | | | 45 | 180 | 29.06.2018 | | | |
| 7 | 142 | 22.05.2018 | x | | | 46 | 181 | 30.06.2018 | | | x |
| 8 | 143 | 23.05.2018 | x | x | | 47 | 182 | 01.07.2018 | | x | x |
| 9 | 144 | 24.05.2018 | x | | | 48 | 183 | 02.07.2018 | | | |
| 10 | 145 | 25.05.2018 | x | | | 49 | 184 | 03.07.2018 | | x | |
| 11 | 146 | 26.05.2018 | x | x | | 50 | 185 | 04.07.2018 | | x | x |
| 12 | 147 | 27.05.2018 | x | | | 51 | 186 | 05.07.2018 | | | |
| 13 | 148 | 28.05.2018 | x | | | 52 | 187 | 06.07.2018 | | | x |
| 14 | 149 | 29.05.2018 | x | x | | 53 | 188 | 07.07.2018 | | | x |
| 15 | 150 | 30.05.2018 | x | | | 54 | 189 | 08.07.2018 | | x | x |
| 16 | 151 | 31.05.2018 | | | x | 55 | 190 | 09.07.2018 | | | |
| 17 | 152 | 01.06.2018 | x | x | | 56 | 191 | 10.07.2018 | x | | |
| 18 | 153 | 02.06.2018 | x | | | 57 | 192 | 11.07.2018 | x | x | x |
| 19 | 154 | 03.06.2018 | x | | | 58 | 193 | 12.07.2018 | | | |
| 20 | 155 | 04.06.2018 | x | | | 59 | 194 | 13.07.2018 | | x | x |
| 21 | 156 | 05.06.2018 | | x | x | 60 | 195 | 14.07.2018 | | | x |
| 22 | 157 | 06.06.2018 | | | x | 61 | 196 | 15.07.2018 | | x | x |
| 23 | 158 | 07.06.2018 | | x | x | 62 | 197 | 16.07.2018 | x | | |
| 24 | 159 | 08.06.2018 | x | | x | 63 | 198 | 17.07.2018 | x | | |
| 25 | 160 | 09.06.2018 | | | x | 64 | 199 | 18.07.2018 | | | |
| 26 | 161 | 10.06.2018 | | x | | 65 | 200 | 19.07.2018 | | | |
| 27 | 162 | 11.06.2018 | x | | | 66 | 201 | 20.07.2018 | x | x | |
| 28 | 163 | 12.06.2018 | x | x | | 67 | 202 | 21.07.2018 | | | |
| 29 | 164 | 13.06.2018 | | | x | 68 | 203 | 22.07.2018 | | | |
| 30 | 165 | 14.06.2018 | x | x | x | 69 | 204 | 23.07.2018 | x | x | |
| 31 | 166 | 15.06.2018 | x | | x | 70 | 205 | 24.07.2018 | x | | |
| 32 | 167 | 16.06.2018 | | | x | 71 | 206 | 25.07.2018 | x | | |
| 33 | 168 | 17.06.2018 | | x | x | 72 | 207 | 26.07.2018 | x | x | |
| 34 | 169 | 18.06.2018 | | | x | 73 | 208 | 27.07.2018 | x | | |
| 35 | 170 | 19.06.2018 | x | | | 74 | 209 | 28.07.2018 | | | |
| 36 | 171 | 20.06.2018 | x | x | | 75 | 210 | 29.07.2018 | | x | |
| 37 | 172 | 21.06.2018 | x | | | 76 | 211 | 30.07.2018 | | | |
| 38 | 173 | 22.06.2018 | | | x | 77 | 212 | 31.07.2018 | | | |
| 39 | 174 | 23.06.2018 | x | x | x | 78 | 213 | 01.08.2018 | x | x | x |

## Appendix B: Accuracy estimates and validation

We evaluated our DEMs by analysing the trueness of our DEM-derived snow height estimates compared to reference heights, i.e., manually measured snow heights. For this, we first analysed the bias, i.e., the mean difference between DEM-derived estimates and manually measured reference data. We also investigated the variability and dispersion as well as the overall accuracy of our data by calculating the standard deviation and the root mean square error (RMSE) between DEM-derived and manually measured snow heights, respectively. Here, we report about two different evaluation schemes: 1) DEM-derived snow heights in the vicinity of the PT sticks are compared to manually measured snow heights at the PT sticks to assess the general data quality and uncertainty in the study area; and 2) a sensitivity test on the number and dependency of GCPs by analysing DEM-derived and manually measured reference snow heights from a second, independent validation area.

### B1   Data quality assessment via ground control analysis within the study area

To assess the quality of DEM-derived snow height information, we compared manually measured data to DEM-derived snow heights at the PT stick locations ($\pm 10\,$cm in x- and +10$\,$cm in y-direction) for all days on which both data are available (in total 14 days). We consider the manually measured data as reference values, i.e., as the true snow heights. We find a mean difference of 0.2$\,$cm, a standard deviation of 1.2$\,$cm, and a root mean square error (RMSE) of 1.3$\,$cm. Single daily mean differences, standard deviations, and RMSEs are listed in Table B1 and illustrated in Fig. B1. Note that some estimates in Table B1 are based on less than 30 data points due to missing data caused by the snow sampling. Since manual data can be influenced by individual persons carrying out the measurement, for comparison we also analysed independent snow height estimates measured on the same day at the same locations by different people which resulted in a mean difference of 0.2$\,$cm, a standard deviation of 0.3$\,$cm, and a RMSE of 0.4$\,$cm, showing that the DEM RMSE of 1.3$\,$cm is a conservative estimate.

**Table B1.** Accuracy estimates for the DEM-derived snow heights in our study area. Mean differences, standard deviations, and RMSEs between the DEM-derived snow heights in the vicinity of the PT stick locations ($\pm 10\,$cm in x- and +10$\,$cm in y-direction) and the manual snow height measurements at the PT sticks are listed. Data are given for all days of the observation period (DOP) on which both DEM-derived and manually measured snow heights are available (Table A1).

| DOP | 5 | 8 | 11 | 14 | 17 | 28 | 30 | 36 | 39 | 57 | 66 | 69 | 72 | 78 |
|---|---|---|---|---|---|---|---|---|---|---|---|---|---|---|
| Mean difference (cm) | 0.35 | -0.92 | -0.02 | -0.75 | 0.11 | 0.84 | 0.64 | 0.17 | 0.36 | 0.68 | -0.15 | 0.37 | 0.93 | 0.23 |
| Standard deviation (cm) | 0.97 | 0.87 | 1.21 | 0.70 | 0.89 | 1.53 | 1.99 | 0.79 | 1.24 | 1.21 | 1.23 | 1.39 | 1.33 | 1.40 |
| RMSE (cm) | 1.01 | 1.25 | 1.19 | 1.00 | 0.88 | 1.73 | 2.06 | 0.79 | 1.27 | 1.37 | 1.22 | 1.42 | 1.60 | 1.39 |

### B2   Validation

The quality of the DEMs can be affected by many aspects during the image acquisition, the GCP allocation, and the DEM processing. During the image acquisition, the camera resolution, the camera-to-object distance, and the angle of the camera

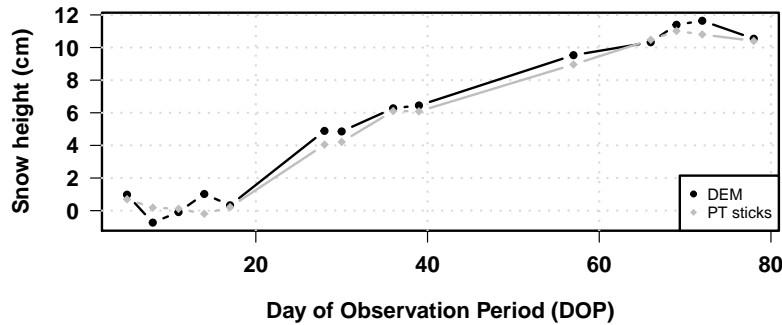

**Figure B1.** Temporal evolution of the DEM-derived snow heights in the vicinity of the PT stick locations ($\pm10\,\text{cm}$ in x- and $+10\,\text{cm}$ in y-direction, black) and of the manually measured snow heights at the PT sticks (grey). Presented are the data for all days during the observation period (DOP) on which both DEM-derived and manually measured snow heights are available (Table A1).

towards the surface can influence the quality of the images (Basnet et al., 2016; Chakra et al., 2019). Moreover, the introduction

of GCPs is necessary to generate geo-referenced DEMs. However, the models can be biased towards the fixed positions of the GCPs, i.e., the glass fibre sticks, due to a stronger contrast (Fig. B2b and c) (e.g., James and Robson, 2014; Cimoli et al., 2017). Since GCPs are only distributed around the study area, not inside, a detailed analysis on potential biases, such as doming effects inside the area, was performed. Furthermore, we investigate the influence of human mistakes during the aligning of the sticks as well as of potential misalignments of GCP marker points during the processing in Agisoft PhotoScan which can introduce

additional uncertainties.

**Ground control analysis with a validation area**

To assess possible biases by the distribution of GCPs, we set up a second, independent area with a size of $50\,\text{m}^2$ (10 m x 5 m, Fig. B2a). This area was set up with the same procedure as the study area and was surrounded by 13 glass fiber sticks which were used as GCPs. We chose a second area where we physically could walk into, which was not possible for the actual study

area in order to avoid snow height disturbances from the foot steps. Four additional sticks (hereafter called *validation sticks*) were distributed inside the validation area on different local snow structures and with different distances to the main line. Image acquisition of this area was performed on three days to account for varying uncertainties in time (Adams et al., 2018), with photos taken on 16[th] of June, 27[th] of June and 9[th] of July 2018. In order not to disturb the DEM generation, the manual snow height measurements at the validation sticks were performed after the image acquisition. By comparing DEM-derived and

manually measured snow height estimates at the validation sticks for all three dates, we obtained an overall mean difference of -0.3 cm, a standard deviation of 1.2 cm, and a RMSE of 1.3 cm (Table B2).

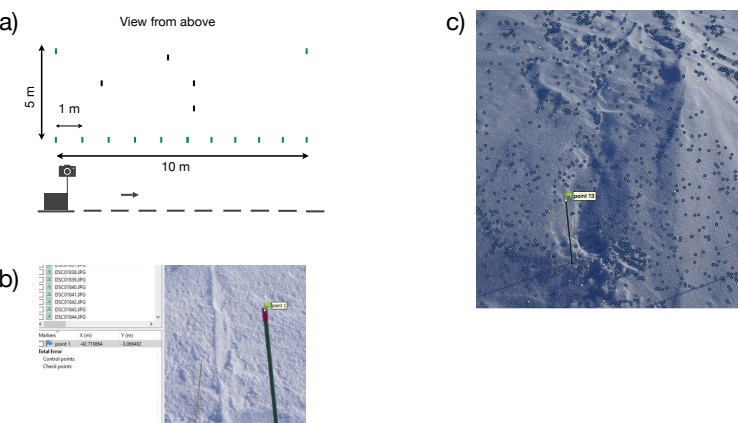

**Figure B2.** Validation area and possible error sources during the processing workflow. a) Schematic illustration of the validation area which was set up according to the same procedure as the main study area. 13 glass fibre sticks were distributed around the validation area (green): eleven sticks were aligned at the downwind side in a straight line with 1 m-spacing and two sticks at the upwind corners at 5 m distance from the main line. Four additional sticks (*validation sticks*, black) were distributed inside the area. b) Allocation of GCPs (ground control points) during the process of DEM (digital elevation model) generation. The top of the glass fibre sticks were used as GCPs and manually checked for correct alignment (c).

**Table B2.** Accuracy measures for the validation area. Mean differences, standard deviations, and RMSEs between DEM-derived and manually measured snow heights are presented for the validation sticks in the validation area.

| Date | 16.06.2018 | 27.06.2018 | 09.07.2018 |
|---|---|---|---|
| Mean difference (cm) | 0.71 | -1.73 | 0.04 |
| Standard deviation (cm) | 0.67 | 0.78 | 0.86 |
| RMSE (cm) | 0.92 | 1.86 | 0.74 |

**Camera-to-object distance and local snow height**

The accuracy of the DEM-derived snow heights depends on all the steps involved in the SfM workflow. This includes the distance between the camera and the object, i.e., the snow surface in our case (Basnet et al., 2016). We therefore assessed the accuracy of the DEM-derived snow height data at the validation sticks for different camera-to-object distances (between 3.8 m and 7 m) and found no dependency on the distance between camera and surface snow.

Since the validation points are distributed on different local surface structures, we also compared the accuracy between varying snow heights but did not find any dependence with respect to the relative snow height.


## Dependency on number and alignment of GCPs

As a final step, we evaluated the accuracy of our DEM-derived snow height estimates depending on the number of used GCPs. It is recommended to use at least three GCPs, however, more GCPs provide a better geo-referencing and a reduced sensitivity to a single point (e.g., James and Robson, 2012; Tonkin et al., 2016). We generated DEMs with five, eight or 13 GCPs and used the detailed snow height information from the validation sticks as ground control analysis. The mean differences between the DEM-derived and the manually measured snow heights at the validation sticks for the different numbers of GCPs are 0.1 cm,

-0.2 cm, and -0.3 cm, respectively, with the standard deviations and RMSEs listed in table B3). The use of more than ten GCPs leads to a decrease in the standard deviation and RMSE, while the mean difference remains comparable.

**Table B3.** Accuracy measures for varying numbers of GCPs used in the DEM generation. Mean differences, standard deviations and RMSEs between DEM-derived and manually measured snow heights using five, eight or 13 GCPs are shown here. Values are averaged for all three dates on which DEMs are available for the validation area (i.e., 16.06., 27.06., and 09.07.2018).

| Number GCPs | 5 | 8 | 13 |
|---|---|---|---|
| Mean difference (cm) | 0.09 | -0.21 | -0.32 |
| Standard deviation (cm) | 1.54 | 1.52 | 1.23 |
| RMSE (cm) | 1.48 | 1.46 | 1.27 |

This analysis, however, assumes that all glass fiber sticks were vertically and horizontally precisely positioned, aligned in a straight line, and that all GCP marker positions were accurately set during the data processing. However, inaccuracies in the GCPs can be caused either by misaligned glass fibre sticks or by misplaced GCP locations during the processing. We therefore

investigated these effects by purposely misaligning GCP positions at the top of the glass fibre sticks. Deviations from the documented coordinates were independently drawn from a normal distribution with a mean of 0 cm and a standard deviation of 2 cm, and were added to the initial input marker coordinates in Agisoft PhotoScan leading to the scenarios a) to e) below. Furthermore, manual misalignment of GCP locations in Agisoft PhotoScan was simulated by deliberately misplacing the GCPs (scenario f).

a) Change of x-coordinates (deviations along the main area).

b) Change of y-coordinates (deviations from the arbitrary chosen zero-line).

c) Change of z-coordinates (deviations in the height of the stick top).

d) Change of x- and y-coordinates by combining the deviations from a) and b).

e) Change of x-, y- and z-coordinates by combining the deviations from a), b) and c).

f) All 13 GCPs were manually set to the left and right margins of the sticks. The normal coordinates without changes were used.

The DEM-derived snow heights from each of these cases is referenced to a DEM assuming perfectly aligned sticks and correct GCP input marker coordinates. Mean deviations between changed and initial DEMs range from -0.1 to 0.1 cm, standard deviations from 0.1 to 1.0 cm, and RMSEs from 0.2 to 1.0 cm (Table B4). Based on this assessment, we conclude that inaccurate distributions of GCPs (e.g., tilted sticks and inaccuracies in the x-, y- or z-coordinates, scenarios a to e) result in an uncertainty of less than 1 cm. Changing the marker position (scenario f) has an even smaller effect on the overall accuracy of the final DEM.

**Table B4.** Accuracy measures for inaccurate GCP coordinates and positions. Mean differences, standard deviations, and RMSEs between normal DEM-derived snow heights and DEM-derived snow heights with altered input marker coordinates or GCP positions. Scenarios a) to f) are explained in the text.

| Scenario | a) | b) | c) | d) | e) | f) |
|---|---|---|---|---|---|---|
| Mean difference (cm) | 0.04 | -0.09 | -0.09 | 0.06 | 0.08 | 0.09 |
| Standard deviation (cm) | 0.12 | 0.17 | 0.92 | 0.14 | 0.98 | 0.10 |
| RMSE (cm) | 0.13 | 0.2 | 0.92 | 0.15 | 0.99 | 0.14 |

## B3 Summary

Uncertainties from manually setting up the transect, distributing the GCP coordinates during the processing, as well as the uncertainty of the GCP alignment are small, especially compared to the amplitude of snow height change throughout our observation period (11 cm on average). We therefore conclude that our elevation models provide reliable snow height estimates with a sufficient accuracy (RMSE of 1.3 cm) for the purpose of our study.

## Appendix C:  Overall snow height evolution

The mean snow height evolution derived from the 2.5 m-band shown in Fig. 4b consists of ~195,000 individual estimates based
on the DEM resolution of 1 x1 cm. The snow height evolution for each individual pixel within this band is illustrated in Fig.
C1 and the relation between the snow height on DOP 1 and the change in snow height from DOP 1 to DOP 78 in Fig. C3.

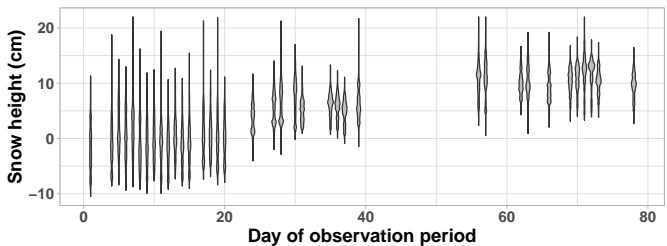

**Figure C1.** Daily variability in snow height throughout the observation period. Each density represents the daily snow height distribution across all individual pixel in the study area. The spatial resolution of the DEM is 1x1 cm. Due to increasingly missing data in the area (footsteps and snow sampling positions), the number of available points per day decreases to about 70 % of the initial number of data points towards the end of the observation period (1,885,506 with 64,494 missing data points due to bad image quality on DOP 1 to 1,319,095 on DOP 78 due to foot steps).

The relative horizontal snow height profiles along the 39 m-long x-axis (Fig. C2) show the overall increase in snow height in our study area.

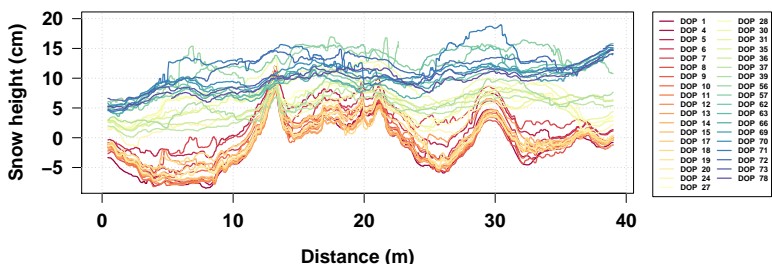

**Figure C2.** Relative horizontal snow height profiles throughout the entire observation period (20-point running median, averaged in y-direction across the 2.5m-band). Different colours represent the different days from DOP 1 to DOP 78, as shown in the legend. Overall, snowfall caused an increase in the snow height at each point in the study area. The first twelve days are shown in more detail in Fig. 6.

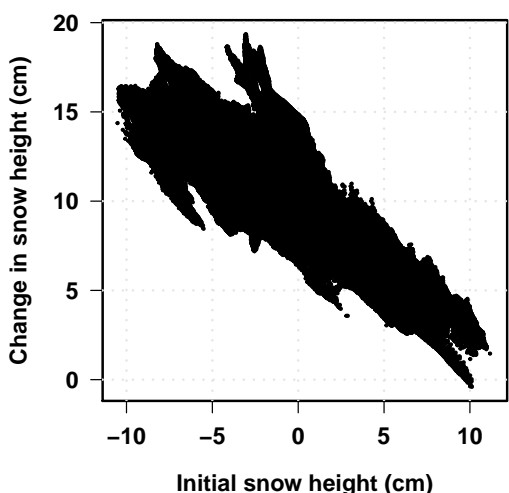

**Figure C3.** Relationship between the initial snow height on DOP 1 and the change in snow height on DOP 78 relative to DOP 1 for each pixel in the study area.

## Appendix D: Surface roughness

The surface roughness is calculated on 2.5 m-long segments along and perpendicular to the main wind direction as illustrated in Fig. D1.

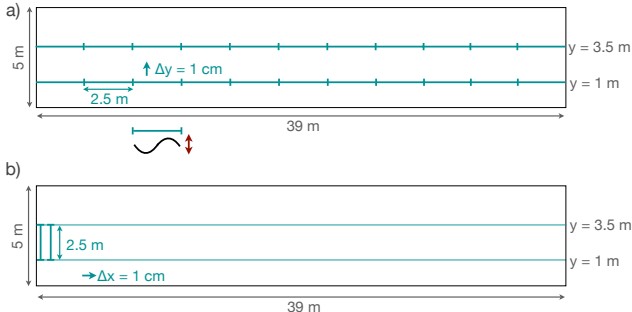

**Figure D1.** Schematic overview of the surface roughness estimates based on Eq. 1. The peak-to-peak amplitude of 2.5 m-long non-overlapping segments is calculated a) perpendicular and b) parallel to the main wind direction. Individual estimates are averaged in each case to obtain a representative surface roughness value. Both estimates cover the same area from $x = 0$ to $x = 39$ m and $y = 1$ to $y = 3.5$ m.

Individual peak-to-peak amplitudes of the 2.5 m-long segments are shown in Fig. D2 and indicate the spread of the individual roughness estimates perpendicular and parallel to the main wind direction as illustrated in the schematic (Fig. D1).

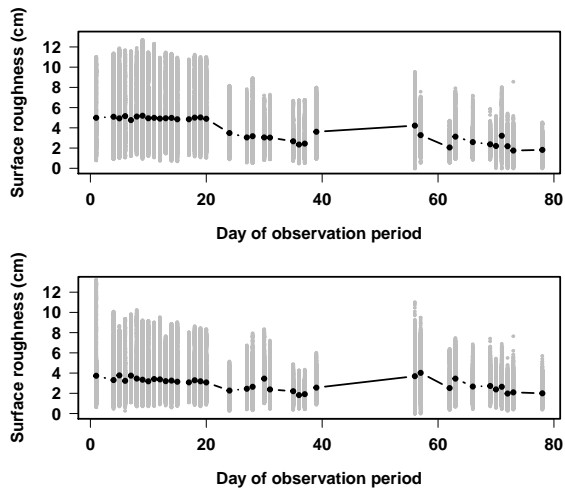

**Figure D2.** Surface roughness estimates throughout the observation period following Eq. 1 (top) perpendicular (Fig. D1a) and (bottom) parallel to the main wind direction (Fig. D1b). Single peak-to-peak amplitudes from the 2.5 m segments are given in grey with the daily average surface roughness in black (Fig. 10).

*Author contributions.* TL, TM, MH and HCSL designed the study. AMZ and HCSL carried it out. AMZ generated the digital elevation models and performed the analyses. AMZ prepared the manuscript with contributions from all co-authors.

*Competing interests.* The authors declare that they have no conflict of interest.

*Acknowledgements.* EGRIP is directed and organized by the Centre for Ice and Climate at the Niels Bohr Institute, University of Copenhagen. It is supported by funding agencies and institutions in Denmark (A. P. Møller Foundation, University of Copenhagen), USA (US National Science Foundation, Office of Polar Programs), Germany (Alfred Wegener Institute, Helmholtz Centre for Polar and Marine Research), Japan (National Institute of Polar Research and Arctic Challenge for Sustainability), Norway (University of Bergen and Trond Mohn Foundation), Switzerland (Swiss National Science Foundation), France (French Polar Institute Paul-Emile Victor, Institute for Geosciences and Environmental research), Canada (University of Manitoba) and China (Chinese Academy of Sciences and Beijing Normal University). We further thank the AWI workshop for the construction of the photogrammetry equipment as well as Sonja Wahl, Anne-Katrine Faber, Melanie Behrens and Tobias Zolles for their help in the data acquisition during the 2018 field campaign. All numerical analyses were carried out by using the software R: A Language and Environment for Statistical Computing. This work has received funding from the European Research Council (ERC) under the European Union's Horizon 2020 research and innovation program: Starting Grant-SNOWISO (grant agreement 759526), Starting Grant-SPACE (grant agreement 716092) and was supported by Helmholtz funding through the Polar Regions and Coasts in the Changing Earth System (PACES) programme of the Alfred Wegener Institute. We thank Quentin Libois and Simon Filhol for their detailed feedback and valuable comments, which have greatly improved this article.

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
