# Peer review of "Local-scale deposition of surface snow on the Greenland ice sheet"

_The Cryosphere, 2021_

## Referee Comment (RC1)

Review of « Local scale depositional processes of surface snow on the Greenland ice sheet », by Alexandra M. Zuhr et al.

**General comments**

This study presents a unique data set of digital elevation models (DEMs) of surface snow near the drilling site of the East Greenland Ice Core Project (EGRIP), acquired near-daily during a whole Summer season (May to August 2018) from a photogrammetry approach. These observations are complemented by more traditional snow height measurements and a variety of meteorological observations. The data are used to extract information on the evolution of average snow height (which increased by 11 cm along the observation period), but also on its spatial variability at the scale of the sampled area (195 m$^2$). It highlights the complexity of spatio-temporal variations, and the poor correlation between precipitation (observed or reported in ERA5 reanalysis) and snow height variations. In particular ~60 % of the deposited snow is at some point removed. This is attributed to the significant role of post-depositional processes, such as snow erosion and subsequent transport by wind. All along Summer, this redistribution results in a reduction of the surface roughness (from 4 to 2 cm), and an overall flattening of the surface. In an extensive discussion section, the impacts of these observations on the proxies used to study climate (e.g. stable water isotopologues) are discussed.

The topic of the study perfectly suits to *The Cryosphere* because it both presents a novel observation methodology, a novel dataset, and interesting results regarding snow processes. The observations are robust and much care is taken to ensure that the observations are valid, and to quantify the uncertainties. The discussion points to relevant questions related to this study, which for some of them (in particular how and when erosion, transport and re-deposition occurs) could have been a bit more explored with the present dataset. The results are not particularly surprising to people familiar with snow physics in polar regions, and it'd be appreciated that more quantitative comparisons be made with previous similar studies. Besides the new technical approach, more insight about how this study complements the existing literature on the topic would be useful as well. The paper is well written and the methodology clearly described. The multiplicity of observations sometimes makes it difficult to follow, and an updated Figure 1 could certainly help the reader. I recommend this manuscript for publication after these minor issues and the technical details below are addressed, and hopefully after a slightly deeper investigation of the data for the physics of snow erosion and transport.

**Specific comments**

1) The detailed quantitative results of the studies are very useful, but should be better put in the context of other studies performed in regions with similar climatic conditions. While it is clearly pointed that the methodology is original (although the differences with classic setups including only 2 cameras should be better highlighted), the novelty of the results, if any, is not sufficiently put forward. It is currently hard to say which ones among the presented results are really unique to this study.

2) Ancillary observations (AWS, snow sampling) are widely mentioned but insufficiently used. Wind speed, and possibly direction, could help interpret the variations of surface roughness (including formation of dunes which may still build up in Summer before to be flattened) or the depositional processes. Snow sampling is not detailed (except for its impact on the study area) but could probably help to stress the spatial heterogeneity displayed in Fig. 11 for instance. The comparison of some snow profiles with this figure would be very useful.

3) In general, the manuscript could be shortened by removing some redundancies (in discussion and conclusions), by clarifying the experiments once for all at the beginning, or by selecting the results. This would leave more room to explore the previous suggestions.

4) Although rich the discussion is a bit long and could probably be shortened. Section 4.3 could be moved to the Results Section because it still contains quantitative results not presented earlier on (e.g. Figs. 10 and 11). Section 4.4 highlights the potential impact of the research on the climatic analysis of ice cores but the conclusions are somehow general. More quantitative estimations of the potential impact would help the reader figure out to which extent the results obtained here can question the current analysis techniques.

**Technical corrections**

l.6-7 : the contribution of snow re-deposition to noise in climate records from ice cores is put as a primary objective of the paper, but I'm not sure strong quantitative conclusions are reached on that question. Consider reformulating the main objective or rephrasing the conclusions.

l.26 : detail briefly how isotopic composition can be changed

l.28 : "larger" is not clear

l.31 : maybe remove "deposited"

l.39 : "mapping" is not clear. Do you mean in space or time ?

l.39 : why is surface roughness important here ?

l.40 : I think precipitation intermittency is completely independent of surface processes, such that accumulation intermittency and precipitation intermittency are two distinct things

l.49 : maybe provide the typical spatial scale of remote sensing observations. For laser altimetry for instance

l.51-52 : it's not clear whether SfM is a particular type of photogrammetry or something different

l.53 : if laser scanners do have limitations, maybe state them here. This will support the use of SfM

l.59-61 : the end of the introduction is incomplete. The objective is not clearly stated, and no outline is provided. Instead some result is provided that should not appear here.

l.65 : "with a mean" is awkward → *where the mean annual temperature is -29°C* ?

l.67 : what should the reader deduce from the comparisons of accumulation rate vs annual layer thickness ? Are these numbers consistent ?

l.71 : are these data used in the study ? If not, this last sentence is useless

l.74 : to achieve this

l.75 : not clear if this is the area covered by one picture or by the whole DEM. Is it dictated by the field of view of the camera ? Clarify the link between the 390 m² and 195 m².

Figure 1 : this figure is central to understand all the measurements that are mentioned in the manuscript. Unfortunately it's not very clear. AWS is loosely positioned because the arrow should point towards the camp which is not shown. The scales are loosely defined (e.g. 90 m, 200 m, 39 m) while they could be consistent. The 10 m width of the SfM method is not shown. X and y axis could be added. What are the 5 sticks above the 35 sticks in the photgrammetry area ? Add the sledge and orientation of the camera

l.76 : "around" does not suggest the sticks are put on a line. Are they ?

l.79 : why "almost"? Are the missing days due to technical issues or were they planned ?

l.84 : how long does it take to take all pictures ? How many pictures are used for each DEM? Why is the width limited to 10 m ? How was the geometry of the study area chosen? Is it necessary to have that many images, compared to standard photogrammetry with only two or three images ?

l.92 : y=10 m was not properly defined, hence this sentence is hard to understand

Figure 2 is hard to relate to Figure 1. Consider adding the footprint of the camera to help

l.95 : does it mean that only a transect is used instead of the full 2D domain ?

l.99 : how do you document the snow height at the glass fibers without perturbing the observed area ? Are the sticks out of the final domain ?

l.105 : "summarised" is unclear. Averaged ?

l.106 : the snow sampling was performed for all 35 glass fibers ? What was measured at this occasion? When was it performed ? Is it used in this study ?

l.110 : how is snowfall documented and how are samples collected?

l.111 : I assume snowdrift can be difficult to distinguish from snowfall in human observations as well

Table 1 : why 30 PT sticks here and not 35 ?

l.115 : show**s**

l.121 : not clear what peak-to-peak means, probably the difference between max and min ?

l.131 : why cannot it be done on the main study area ?

l.133 : redundant with just a few lines above

l.140 : it is not clear what additional information this section provides compared to the previous sections

l.152 : why was not this sensitivity study performed directly on the study site ?

l.170 : *sufficient* accuracy

l.171 : here the final estimation of DEM accuracy should be mentioned. Otherwise it's used later on (1,3 cm) without relevant reference.

l.180 : it seems that on Panel 2 of Figure 3 the dunes have already vanished

Figure 3 : having these x and y axes in Fig. 1 would help a lot. Refer to the section where the areas in grey are used. "Snow sampling *scheme"* sounds awkward, remove scheme ? Clarify in the text (l.106) how frequently such snow sampling were performed, and make it clear whether this corresponds to the readings of snow height at the stakes or not.

l.201 : Reference to Libois et al. (2014) might be relevant here (Figure 2 for instance) or elsewhere

l.206 : any insight/reference about the quality of ERA 5 snowfall reanalysis over Greenland ?

l.216 : not clear whether the consistency is in terms of snowfall occurrence or amount

l.217 : not clear how 0,6 cm should be read in Figure 4c

l.218 : it's hard in Figure 5 to see the successive lines. Maybe consider changing color type when erosion occurs

l.222 : a bit unclear, maybe reformulate : " … on one fixed day and that on any other day "

l.224 : the link between RMSE decrease and erosion is not strict. At least RMSE can decrease without erosion (by smoothing for instance). An interesting quantity could be the RMSE between successive DEMs, after removing mean deviation. Maybe RMSD (deviation) would be more appropriate than RMSE here.

l.232 : not clear what this area is because it often takes a different name in the Figures and in the text. Is it the full domain or only the 0,5 m band ? Maybe give it a name, like x-transect, or "area A"

l.246 : roughness has already been defined earlier

l.246 : not clear where the wind parallel line is (what x ?) and whether 50 cm refers to the length of the segments, in which case why is that different than the 2,5 m used in the other direction ?

l.251 : decrease **with time**

l.258 : where does this 1,3 cm come from ?

l.259 : given the acquisition is probably fast, acquisition could be more frequent than daily. Maybe remove this detail

l.261 : provide references for the 40 m² and 110 m²

l.263 : Ok, but what's the rationale of having such a particular study area (by the way, it'd be helpful to explain earlier on how these dimensions were chosen/constrained, as a square area would be more understandable), compared to a circular area ?

l.266 : the main disadvantage remains the fact that you need an operator, although this could probably be made automatic somehow. What would be the result if only 2 cameras were used in an automatic way ?

l.273 : maybe clarify the human errors, which could be helpful to readers interested in deploying the same kind of instrumentation

l.277 : this title is not clear, maybe just remove reliable

l.288 : please describe where the stakes are placed in these simulations(random distribution, lines etc.)

l.295 : it seems that spacing beyond 5 m is useless in your case, which might be worth pointing. Then, consider providing suggestions, for instance how to maximize the accuracy with a minimum of stakes.

Figure 9 : Is the RMSD computed on a different number of mean values for different spacings ? Maybe clarify this

l.307 : wind speed during the observation period could be advantageously used to explore the drift/deposition events

l.324 : "final snow accumulation" not clear, because precipitation probably governs the final (end of season or yearly average) snow accumulation, but not high frequency variations.

l.325 : 290 kg m$^{-3}$ seems a bit large for fresh snow. Could you provide more details on how this value was chosen

l.327 : how do ERA5 data suggest that build up is very irregular in time? Not clear

l.329 : it'd be helpful to know what "local" means for climate studies, and how far can snow be transported in the study area

l.339 : consider providing the range of snow ages at the end of the observation period

l.339 : does the snow sampling provide valuable information with regards to the spatial heterogeneity of the layering ?

l.345 : the layering does not record each precipitation event, but when snow settles down as a single layer, it probably contains snow with different ages. Somehow there is a "snow reservoir" in between precipitation and settlement, which is fed by precipitation and at some point is incorporated to the snowpack.

l.363 : this idea has already been discussed

l.363 : how much is strong ? Would you have references (if no measurements) regarding snow transport to compare scales ?

l.371 : could the images be used to identify very local re-deposition (within the same observed area) ?

l.390 : are you sure that your observation of dunes vanishing in Summer is representative ? Could it be that you studied a singular year ? Were the wind statistics in agreement with longer term observations ?

l.478 : missing beginning of sentence

**References**

Libois, Q., Picard, G., Arnaud, L., Morin, S., & Brun, E. (2014). Modeling the impact of snow drift on the decameter-scale variability of snow properties on the Antarctic Plateau. *Journal of Geophysical Research: Atmospheres, 119*(20), 11-662.

---

## Author Comment (AC1)

Reply to the Review Comments of Quentin Libois (Referee #1)

on the manuscript

TC-2021-36: Local scale depositional processes of surface snow on the Greenland ice sheet

by Alexandra M. Zuhr et al.

*Thank you for your effort and your careful and detailed review of our manuscript. We appreciate your constructive feedback that will help to improve the manuscript. Below we provide a point-by-point response to all comments. The original referee comments are set in normal font and our answers (author comment, AC) are set in blue.*

**General comments**

This study presents a unique data set of digital elevation models (DEMs) of surface snow near the drilling site of the East Greenland Ice Core Project (EGRIP), acquired near-daily during a whole Summer season (May to August 2018) from a photogrammetry approach. These observations are complemented by more traditional snow height measurements and a variety of meteorological observations. The data are used to extract information on the evolution of average snow height (which increased by 11 cm along the observation period), but also on its spatial variability at the scale of the sampled area (195 m2). It highlights the complexity of spatio-temporal variations, and the poor correlation between precipitation (observed or reported in ERA5 reanalysis) and snow height variations. In particular ~60 % of the deposited snow is at some point removed. This is attributed to the significant role of post-depositional processes, such as snow erosion and subsequent transport by wind. All along Summer, this redistribution results in a reduction of the surface roughness (from 4 to 2 cm), and an overall flattening of the surface. In an extensive discussion section, the impacts of these observations on the proxies used to study climate (e.g. stable water isotopologues) are discussed.

The topic of the study perfectly suits to The Cryosphere because it both presents a novel observation methodology, a novel dataset, and interesting results regarding snow processes. The observations are robust and much care is taken to ensure that the observations are valid, and to quantify the uncertainties. The discussion points to relevant questions related to this study, which for some of them (in particular how and when erosion, transport and re-deposition occurs) could have been a bit more explored with the present dataset. The results are not particularly surprising to people familiar with snow physics in polar regions, and it'd be appreciated that more quantitative comparisons be made with previous similar studies. Besides the new technical approach, more insight about how this study complements the existing literature on the topic would be useful as well. The paper is well written and the methodology clearly described. The multiplicity of observations sometimes makes it difficult to follow, and an updated Figure 1 could certainly help the reader. I recommend this manuscript for publication after these minor issues and the technical details below are addressed, and hopefully after a slightly deeper investigation of the data for the physics of snow erosion and transport.

AC:
We are happy that the manuscript is generally seen as an important and significant work contributing to a better understanding of snow processes. We acknowledge the suggested minor points and carefully address them as outlined below. We will better discuss our results in the context of previous studies (e.g. comparing directly to Picard et al. (2019)) and define more clearly what is unique and similar to the existing literature. We will update Figure 1 for a clearer indication of individual methodologies and improve the technical details as well as the method section. Unfortunately, the temporal resolution and temporal gaps (partly caused by the fact that it was the first season in which this setup was used) limit to robustly determine the origin and timing of the erosion, transport and re-distribution. On the other hand, our dataset is designed to and sufficient to determine the statistics of deposition that are relevant for the formation of environmental and climatic records. We will extend the interpretation and discussion of this point and also adapt our title to "Local scale deposition of surface snow on the Greenland ice sheet" to better define the focus of our study.

**Specific comments**

1) The detailed quantitative results of the studies are very useful, but should be better put in the context of other studies performed in regions with similar climatic conditions. While it is clearly pointed that the methodology is original (although the differences with classic setups including only 2 cameras should be better highlighted), the novelty of the results, if any, is not sufficiently put forward. It is currently hard to say which ones among the presented results are really unique to this study.

AC: We will extend the discussion on how our study relates to other studies from  regions with similar characteristics and also better define the scope of the manuscript (including its title). We will include a sentence to highlight the differences to classical 'stereo' camera setups.

2) Ancillary observations (AWS, snow sampling) are widely mentioned but insufficiently used. Wind speed, and possibly direction, could help interpret the variations of surface roughness (including formation of dunes which may still build up in Summer before to be flattened) or the depositional processes. Snow sampling is not detailed (except for its impact on the study area) but could probably help to stress the spatial heterogeneity displayed in Fig. 11 for instance. The comparison of some snow profiles with this figure would be very useful.

AC: During our research process, we analysed the AWS data (wind speed and direction) in detail. We agree that it would be very useful if one could establish clear links between the meteorological conditions and the development of the surface structure. However, we did not identify any clear links, potentially as our sampling resolution (every day, with some gaps in between) is too low. We will show the AWS data and a comparison to the roughness in the appendix of the revised manuscript. The snow sampling data are still in analysis and thus would be beyond the scope of this manuscript.

3) In general, the manuscript could be shortened by removing some redundancies (in discussion and conclusions), by clarifying the experiments once for all at the beginning, or by selecting the results. This would leave more room to explore the previous suggestions.

AC: In the revised version, we will remove some redundancies in the discussion and conclusion. We think that the presented choice of the results reflects the main analyses concerning the topic of the manuscript, but we will move Figure 11 to the results.

4) Although rich the discussion is a bit long and could probably be shortened. Section 4.3 could be moved to the Results Section because it still contains quantitative results not presented earlier on (e.g. Figs. 10 and 11). Section 4.4 highlights the potential impact of the research on the climatic analysis of ice cores but the conclusions are somehow general. More quantitative estimations of the potential impact would help the reader figure out to which extent the results obtained here can question the current analysis techniques.
AC: As suggested we will move 4.3 to the result sections and rewrite Section 4.4 to be more concrete.

**Technical corrections**

l.6-7 : the contribution of snow re-deposition to noise in climate records from ice cores is put as a primary objective of the paper, but I'm not sure strong quantitative conclusions are reached on that question. Consider reformulating the main objective or rephrasing the conclusions.
AC: We agree that the conclusions are not answering the proposed questions. We will reformulate our conclusions to meet the stated objectives.

l.26 : detail briefly how isotopic composition can be changed
AC: Thank you for pointing out the missing information on changes in the isotopic composition. We will add a comment on this in the introduction.

l.28 : "larger" is not clear
AC: We will rephrase it to "larger scale processes".

l.31 : maybe remove "deposited"
AC: Will be changed as suggested.

l.39 : "mapping" is not clear. Do you mean in space or time ?
AC:  We are referring here to both the spatial and temporal mapping of snowfall. We clarify this by adding "spatial and temporal mapping".

l.39 : why is surface roughness important here ?
AC: Surface roughness is influencing the spatial deposition of snowfall, especially during windy snowfall conditions. We will add this information in the manuscript.

l.40 : I think precipitation intermittency is completely independent of surface processes, such that accumulation intermittency and precipitation intermittency are two distinct things
AC: Precipitation intermittency is not influenced by surface processes, such as erosion or snowdrift. However, precipitation intermittency is part of accumulation intermittency, because it determines the timing and amount of snow available for re-distribution and transport. We will replace 'precipitation' with 'accumulation'.

l.49 : maybe provide the typical spatial scale of remote sensing observations. For laser altimetry for instance
AC: We will specify the spatial scale of current laser altimeter systems to provide a better understanding of the need of small scale methods.

l.51-52 : it's not clear whether SfM is a particular type of photogrammetry or something different
AC: Photogrammetry SfM is a technique for itself and is already widely used, also for similar studies in the field of glaciology (e.g., Chakra et al., 2019; Filhol et al., 2019). We will add a sentence on this technique in the manuscript.

l.53 : if laser scanners do have limitations, maybe state them here. This will support the use of SfM
AC: We will add and elaborate on the limitations of laser scanner studies at this point in the manuscript.

l.59-61 : the end of the introduction is incomplete. The objective is not clearly stated, and no outline is provided. Instead some result is provided that should not appear here.
AC: We will clarify the objectives and remove the results.

l.65 : "with a mean" is awkward. where the mean annual temperature is -29 C ?
AC: Will be changed accordingly.

l.67 : what should the reader deduce from the comparisons of accumulation rate vs annual layer thickness ? Are these numbers consistent ?
AC: We will put them all to the same unit (mm w.eq. yr-1) for easier comparisons. By showing the different accumulation rate estimates, we want to highlight the high spatial variability in local accumulation at our study site.

l.71 : are these data used in the study ? If not, this last sentence is useless
AC: We will remove this sentence.

l.74 : to achieve this
AC: Will be changed as suggested.

l.75 : not clear if this is the area covered by one picture or by the whole DEM. Is it dictated by the field of view of the camera ? Clarify the link between the 390m2 and 195m2.
AC: The total area covered by all images per survey is 390m2. Due to the lack of coverage with images and the lower image quality at the rear of the area, the covered area by the DEMs is only 195m2. We will clarify this in the revised manuscript.

Figure 1 : this figure is central to understand all the measurements that are mentioned in the manuscript. Unfortunately it's not very clear. AWS is loosely positioned because the arrow should point towards the camp which is not shown. The scales are loosely defined (e.g. 90 m,

200 m, 39 m) while they could be consistent. The 10m width of the SfM method is not shown. X and y axis could be added. What are the 5 sticks above the 35 sticks in the photogrammetry area ? Add the sledge and orientation of the camera

AC: We will clarify the positions, make consistent scales and update Figure 1. We will further add the position of the sledge as shown in Figure 2 as well as the approximate field of view of the camera.

l.76 : "around" does not suggest the sticks are put on a line. Are they ?

AC: We agree that the sentence was not precise. The sticks along the x-axis are set on one line, the surrounding sticks are also positioned on line to create a rectangle. We will clarify this in the manuscript.

l.79 : why "almost"? Are the missing days due to technical issues or were they planned ?

AC: No photos were taken on very cloudy days with no visible contrast or with whiteout conditions. These conditions do not allow any snow height reconstructions with optical images only. NIR would be necessary to extract more information during these weather conditions. Further missing days are caused by (human) errors in the data acquisition. We will clarify this in the manuscript.

l.84 : how long does it take to take all pictures ? How many pictures are used for each DEM? Why is the width limited to 10 m ? How was the geometry of the study area chosen? Is it necessary to have that many images, compared to standard photogrammetry with only two or three images ?

AC: The image acquisition itself took about three to five minutes; however, including the preparation and walking time to and from the camp, the time effort was about 45 minutes. The width of 10 m was chosen based on thoughts about image quality for DEM generation. During the field period, we realised that the images are not good enough for a DEM generation up to 10 m width; thus, we restricted the analysed area to 5 m width. About 60 images were used for one DEM. If less than 50 to 60 photos were used, no DEM could be generated because the overlap between successive images was too small. This can be caused by a lack of available surface features to match the images which is a reason for using that many images, instead of two or three, as other studies did. We will add more information about the number of pictures in the methods part and extend the discussion by comparing our method to setups with e.g. only two cameras.

l.92 : y=10 m was not properly defined, hence this sentence is hard to understand

AC: We will clarify y = 10 m in Figure 1.

Figure 2 is hard to relate to Figure 1. Consider adding the footprint of the camera to help

AC: The footprint of the camera in Figure 2 will be added to Figure 1.

l.95 : does it mean that only a transect is used instead of the full 2D domain ?

AC: No, the analysed area is only restricted to the area from y=0m to y=5 m which equals a size of 195m2.

l.99 : how do you document the snow height at the glass fibers without perturbing the observed area ? Are the sticks out of the final domain ?

AC: The sticks surrounded the observed area and could thus be accessed and measured without perturbing or stepping into the studied area.

l.105 : "summarised" is unclear. Averaged ?

AC: Will be changed as suggested.

l.106 : the snow sampling was performed for all 35 glass fibers ? What was measured at this occasion? When was it performed ? Is it used in this study ?

AC: We apologise for the unclear information on the snow sampling. The sampling was performed at 30 stick positions along the x-axis every third day throughout the entire observation period. No samples were taken at the remaining five stick positions. The snow samples were measured for stable water isotopic composition and are not used in this study. We will add more details on the snow sampling procedure and the resulting surface disturbances in the manuscript.

l.110 : how is snowfall documented and how are samples collected?

AC: Snowfall was manually documented in a spreadsheet. If visual snowfall was observed and/or the snow collection tables (setup described in Steen-Larsen et al., 2014) had snow, we noted this down. We will add more information at this point in the manuscript.

l.111 : I assume snowdrift can be difficult to distinguish from snowfall in human observations as well

AC: We agree that snowfall is visually difficult to distinguish from snowdrift. We therefore analysed the DEMs with regard to manual snowfall documentation and ERA5 indications of snowfall.

Table 1 : why 30 PT sticks here and not 35 ?

AC: 30 sticks were set up along the x-axis, the remaining five sticks are mainly used to provide a reliable geo-referencing of the DEMs and not for further analyses. We will clarify this in the manuscript.

l.115 : shows

AC: Will be changed as suggested.

l.121 : not clear what peak-to-peak means, probably the difference between max and min?

AC: Peak to peak refers to the difference between the minimum and the maximum snow height. We will add a mathematical equation to the manuscript to clarify the calculation of the surface roughness.

l.131 : why cannot it be done on the main study area ?
AC: We did not want to disturb the study area by adding too many footsteps. Therefore, we set up a second area where we physically walked into the area to establish validation points within the area. We will mention this reason in the appendix.

l.133 : redundant with just a few lines above
AC: We will remove the redundant parts.

l.140 : it is not clear what additional information this section provides compared to the previous sections
AC: We tried to extensively validate our method by analysing different error terms which can arise when e.g. only using GCPs outside of the study area. We therefore included a detailed analysis of potential biases (e.g. doming effects) and mentioned the calculated uncertainties. We will restructure the method section and move most of the accuracy estimate to the appendix where an additional validation is already presented.

l.152 : why was not this sensitivity study performed directly on the study site ?
AC: The validation area has additional control points within the area, not only surrounding points as in the study area. Since we did not want to disturb the study area in addition to the snow sampling, we decided to set up a separate validation area to perform this sensitivity study. We will improve the description in the manuscript.

l.170 : sufficient accuracy
AC: Will be changed as suggested.

l.171 : here the final estimation of DEM accuracy should be mentioned. Otherwise it's used later on (1,3 cm) without relevant reference.
AC: We will add the final DEM accuracy here.

l.180 : it seems that on Panel 2 of Figure 3 the dunes have already vanished
AC: On DOP 36, the surface already became smoother compared to the beginning of our observation period. This is also shown in the reduced surface roughness towards this day (Figure 8). We will discuss this behavior in more detail in the revised manuscript.

Figure 3 : having these x and y axes in Fig. 1 would help a lot. Refer to the section where the areas in grey are used. "Snow sampling scheme" sounds awkward, remove scheme ? Clarify in the text
AC: We will remove "scheme" and clarify it in the text accordingly. We will refer to the sections where the grey areas are used and update Figure 1 by adding x- and y-axes.

(l.106) how frequently such snow sampling were performed, and make it clear whether this corresponds to the readings of snow height at the stakes or not.
AC: The snow sampling was performed every third day. Manual reading of the snow height at the stakes did not always correspond to the snow sampling dates. We will clarify this in the text.

l.201 : Reference to Libois et al. (2014) might be relevant here (Figure 2 for instance) or elsewhere
AC: We will add this reference.

l.206 : any insight/reference about the quality of ERA 5 snowfall reanalysis over Greenland?
AC: We will add a reference to Delhasse et al. (2020) showing that ERA5 provides reliable near-surface variables (2 m temperature, 10 m wind speed, energy downward fluxes) for the Greenland Ice Sheet. We find agreement in terms of timing for the ERA5 snowfall product and our documented snowfall (Fig. 4c in the manuscript); however, ERA5 shows slightly more occurrences of snowfall than we noted down. We might have missed snowfall for example when it snowed only during the night.

l.216 : not clear whether the consistency is in terms of snowfall occurrence or amount
AC: With 'timing of snowfall' we refer to the occurrence, not the amount.

l.217 : not clear how 0,6 cm should be read in Figure 4c
AC: The amount of 0.6 cm snowfall is derived from the data presented in Figure 4c and converted from mm w.eq. to cm of snowfall (not shown in Figure 4c). We will clarify this in the text.

l.218 : it's hard in Figure 5 to see the successive lines. Maybe consider changing color type when erosion occurs
AC: We agree that the color code in Figure 5 is not intuitive. We are currently experimenting with different color types and /or to add arrows indicating when erosion starts or occurs (see Figure 1).

[Figure]

Potential new Figure 1: Relative horizontal snow height profiles (20-point running median, averaged in y-direction from the 2.5m-band) along the x-axis of the study area. Different colours represent different days from DOP 1 to DOP 12 as well as respective mean snow heights in cm, both shown in the legend. Snowfall caused an overall snow height increase from DOP 1 to DOP 7, followed by an erosive event removing the new snow, and exposing the previous surface structure again. Arrows indicate the erosional decrease in the snow height from DOP 7 to DOP 8.

l.222 : a bit unclear, maybe reformulate : " . . . on one fixed day and that on any other day "

AC: Will be changed as suggested.

l.224 : the link between RMSE decrease and erosion is not strict. At least RMSE can decrease without erosion (by smoothing for instance). An interesting quantity could be the RMSE between successive DEMs, after removing mean deviation. Maybe RMSD (deviation) would be more appropriate than RMSE here.

AC: We agree that the figure is not yet as intuitive as we are aiming for. We picked the RMSE between DEM's (thus also including the mean deviation) to demonstrate that old layers seem to reappear, thus the mean value gets closer but also the shape, thus the correlation increases. We are experimenting with other metrics to make the figure more intuitive.

l.232 : not clear what this area is because it often takes a different name in the Figures and in the text. Is it the full domain or only the 0,5 m band ? Maybe give it a name, like x-transect, or "area A"

AC: It is a good suggestion to name the different surveyed areas. We will add the name '2.5m-band' for the band from y=2.5 m to y=3 m in the description of the study area and use it throughout the manuscript.

l.246 : roughness has already been defined earlier

AC: We will remove the repetition here.

l.246 : not clear where the wind parallel line is (what x ?) and whether 50 cm refers to the length of the segments, in which case why is that different than the 2,5 m used in the other direction ?

AC: The line parallel to the wind is the y-axis while the x-axis is oriented perpendicular to the main wind direction. For clarity, we will indicate the axes in Figure 1. For the surface roughness estimate parallel to the main wind direction (i.e., along the y-axis), the length of each single segment is 2.5 m (from y=1 m to y=3.5 m).

l.251 : decrease with time

AC: Will be changed accordingly.

l.258 : where does this 1,3 cm come from ?

AC: The accuracy of 1.3 cm is derived from the method validation in section 2.3 and the validation section in Appendix B and C. We will mention the final uncertainty of 1.3 cm here again to highlight this value.

l.259 : given the acquisition is probably fast, acquisition could be more frequent than daily. Maybe remove this detail

AC: Yes, it is possible to acquire images more often than once a day since the data acquisition takes only about 30 to 45 minutes, including the walking time to and from the study area. We will remove this detail.

l.261 : provide references for the 40m$^2$ and 110m$^2$
AC: The sizes of 40m$^2$ and 110m$^2$ are taken from Picard et al. (2016, 2019), respectively. We will add the references next to the numbers.

l.263 : Ok, but what's the rationale of having such a particular study area (by the way, it'd be helpful to explain earlier on how these dimensions were chosen/constrained, as a square area would be more understandable), compared to a circular area ?
AC: A circular area from a laser scanner has the disadvantage that the location of interest cannot be changed as quickly as in our setup. Furthermore, the laser itself is a high obstacle which can influence the snow re-distribution and can thus affect the natural snow accumulation conditions. We will extend the discussion here.

l.266 : the main disadvantage remains the fact that you need an operator, although this could probably be made automatic somehow. What would be the result if only 2 cameras were used in an automatic way ?
AC: Indeed, the need for an operator is a large disadvantage. Using only two cameras would probably result in a very small surveyed area since the field of view of one image does not cover an area of 20 m length with a sufficiently high resolution allowing a DEM generation. We will add the limitation, i.e. the need of a human operator, to the evaluation of our method.

l.273 : maybe clarify the human errors, which could be helpful to readers interested in deploying the same kind of instrumentation
AC: Thank you for pointing out that a more detailed description of the human errors can be helpful for readers. We will add more details on this.

l.277 : this title is not clear, maybe just remove reliable
AC: We do not agree with the reviewer here. In this section, we present an optimal measurement setup including number of sticks and their spacing to reliably determine snow accumulation, which we derive from our study area that has been surveyed at high spatial resolution. We therefore would keep the title as it is.

l.288 : please describe where the stakes are placed in these simulations(random distribution, lines etc.)
AC: We use the line at y = 2.5 m as reference and place the simulated number of stakes on this line with the given spacing (no random distribution). We then use all possible combinations with the chosen number of stakes and spacing. Depending on the number of stakes and spacing, we have a changing number of possible combinations. We will describe the approach in more detail in the manuscript.

l.295 : it seems that spacing beyond 5 m is useless in your case, which might be worth pointing. Then, consider providing suggestions, for instance how to maximize the accuracy with a minimum of stakes.
AC: Yes, based on our simulations, a spacing beyond 5 m is not improving the estimate of the snow height change. We will emphasise this in the text. Moreover, according to our experiment,

seven sticks with 5 m spacing deliver the optimal minimum setup of stakes (as mentioned in the manuscript).

Figure 9 : Is the RMSD computed on a different number of mean values for different spacings ? Maybe clarify this
AC: We will clarify our approach in the manuscript. Depending on the chosen number of sticks and spacing, we have a different number of possible combinations to put the stakes on the line. Each combination provides a mean snow height change using the chosen number of stakes. We then calculate the difference to the reference snow height change (all points along the y = 2.5 m line) resulting in the same number of differences as possible combinations. From these differences, we then calculate the RMSD.

l.307 : wind speed during the observation period could be advantageously used to explore the drift/deposition events
AC: The wind speed during the observation period is illustrated in Figure A1 in the appendix of the manuscript. During the course of the analyses for this manuscript, we also studied the relation between wind characteristics (speed and direction) and changes in the snow height. However, we did not identify any clear links between these parameters. Therefore, we did not add more details on a possible relation between wind characteristics and snow erosion/deposition in the manuscript.

l.324 : "final snow accumulation" not clear, because precipitation probably governs the final (end of season or yearly average) snow accumulation, but not high frequency variations.
AC: We will clarify this by adding "final snow accumulation during the observation period".

l.325 : 290 kg m-3 seems a bit large for fresh snow. Could you provide more details on how this value was chosen
AC: The snow density of 290 kg m-3 is derived from daily density measurements along the SSA transect. At each of the ten stick positions, the snow density of the top 2.5 cm of snow is measured in addition to the specific surface area (SSA). Snow density data are not further used in this study. We will clarify the density value in the manuscript.

l.327 : how do ERA5 data suggest that build up is very irregular in time? Not clear
AC: We refer here to the results from the DEM data (as seen in Figure 10) and the comparison of these to the ERA5 snowfall data. We will clarify it in the text.

l.329 : it'd be helpful to know what "local" means for climate studies, and how far can snow be transported in the study area
AC: We agree that the definition of "local" in this context is not straightforward. We will specify local climate signal by adding "local air temperature". Furthermore, we analysed different resources to estimate the distance of transported snow; unfortunately, we did not find a conclusive distance for snow transport.

l.339 : consider providing the range of snow ages at the end of the observation period
AC: It is a good point to investigate the range of theoretical snow ages at the end of our observation period. Analysing the (virtual) top 5 mm of the snow surface indicates that the average day of accumulation is DOP 66.3 (with a standard deviation of 7.4 days, total range between DOP 55 and DOP 78). Real snow ages are, however, difficult to determine because we cannot distinguish between freshly precipitated and eroded or drifted snow which would already have an older age. We will add information on the (range of) snow age in the manuscript.

l.339 : does the snow sampling provide valuable information with regards to the spatial heterogeneity of the layering ?
AC: Stable water isotopes are measured in the snow samples, but the data are not analysed yet. Thus, we cannot conclude on the (isotopic) heterogeneity of the layering. Adding these data would be beyond the scope of this manuscript.

l.345 : the layering does not record each precipitation event, but when snow settles down as a single layer, it probably contains snow with different ages. Somehow there is a "snow reservoir" in between precipitation and settlement, which is fed by precipitation and at some point is incorporated to the snowpack.
AC: Many thanks for these thoughts. It is correct that a single layer, which can consist of precipitated and drifted snow, may contain snow with different ages which probably has implications for the stored climatic signal in the respective snow layer. We will include this thought in the discussion.

l.363 : this idea has already been discussed
AC: We will reword this section to avoid repetition.

l.363 : how much is strong ? Would you have references (if no measurements) regarding snow transport to compare scales ?
AC: Stable water isotopologues, density data and accumulation rates show large interannual variations on local but also larger scales of e.g. 450 km in North Greenland (e.g. Schaller et al., 2016). The authors also mention the importance of the smoothing of the snow surface. However, data regarding snow transport are difficult to obtain and we have no references about these scales for Greenland. We will extend our discussion at this point in the manuscript.

l.371 : could the images be used to identify very local re-deposition (within the same observed area) ?
AC: In our opinion, the images are not sufficient to identify local re-deposition within our study area. We cannot definitely link a re-deposited snow particle to its origin from a spot within our study area.

l.390 : are you sure that your observation of dunes vanishing in Summer is representative ? Could it be that you studied a singular year ? Were the wind statistics in agreement with longer term observations?

AC: As we only have this single season, we cannot be sure. However, a study from Summit, Greenland (Albert and Hawley, 2002), showed a similar behaviour of vanishing dunes during summer. Our observation period is comparable with regard to wind speed and direction as shown by the wind statistics from the nearby AWS for the years 2017 to 2019 (see Figure 2 and 3). We also compare the wind speed frequency during our observation period to the winter months (December, January, February) and see that the winter months generally show higher wind speeds (see Figure 4) which can enhance the formation of sastrugi. Thus the meteorological data would support the hypothesis that what we see is a representative observation.

[Figure]

Figure 2: Hourly wind characteristics (speed and direction) for the years 2017, 2018 and 2019 recorded by the nearby PROMICE AWS.

[Figure]

Figure 3: Hourly wind characteristics (speed and direction) for the observation period from 16th May to 1st August 2018 recorded by the nearby PROMICE AWS.

[Figure]

Figure 4: Hourly wind speed data from the nearby PROMICE AWS for two different time periods. Wind speeds during the winter months of 2017 to 2019 (December, January, February; DJF) are compared to wind speeds during our observation period (16.05. - 01.08.2018). The winter months are characterised by higher wind speeds with a mean of ~6 m/s while the average wind speed during our observation period was 4.1 m/s.

l.478 : missing beginning of sentence
AC: Thank you for pointing this out. We will update the acknowledgements.

References

Chakra, C. A., Gascoin, S., Somma, J., Fanise, P., and Drapeau, L.: Monitoring the Snowpack Volume in a Sinkhole on Mount Lebanon using Time Lapse Photogrammetry, Sensors, 19, https://doi.org/10.3390/s19183890, https://www.mdpi.com/1424-8220/19/18/3890, 2019.

Delhasse, A., Kittel, C., Amory, C., Hofer, S., van As, D., S. Fausto, R., and Fettweis, X.: Brief communication: Evaluation of the near-surface climate in ERA5 over the Greenland Ice Sheet, The Cryosphere, 14, 957–965, https://doi.org/10.5194/tc-14-957-2020, https://tc.copernicus.org/articles/14/957/2020/, 2020.

Filhol, S. and Sturm, M.: The smoothing of landscapes during snowfall with no wind, Journal of Glaciology, 65, 173–187, https://doi.org/10.1017/jog.2018.104, 2019.

Libois, Q., Picard, G., Arnaud, L., Morin, S., & Brun, E. (2014). Modeling the impact of snow drift on the decameter-scale variability of snow properties on the Antarctic Plateau. Journal of Geophysical Research: Atmospheres, 119(20), 11-662.

Schaller, C. F., Freitag, J., Kipfstuhl, S., Laepple, T., Steen-Larsen, H. C., and Eisen, O.: A representative density profile of the North Greenland snowpack, The Cryosphere, 10, 1991–2002, https://doi.org/10.5194/tc-10-1991-2016, 2016.

Steen-Larsen, H. C., Masson-Delmotte, V., Hirabayashi, M.,Winkler, R., Satow, K., Prié, F., Bayou, N., Brun, E., Cuffey, K. M., Dahl-Jensen, D., Dumont, M., Guillevic, M., Kipfstuhl, S., Landais, A., Popp, T., Risi, C., Steffen, K., Stenni, B., and Sveinbjörnsdóttir, A. E.: What controls the isotopic composition of Greenland surface snow?, Climate of the Past, 10, 377–392, https://doi.org/10.5194/cp-10-377-2014, 2014.

---

## Author Comment (AC2)

**Reply to the Review Comments of Simon Filhol (Referee #2)**

on the manuscript

TC-2021-36: Local scale depositional processes of surface snow on the Greenland ice sheet

by Alexandra M. Zuhr et al.

We are very thankful for the detailed and thorough review of our manuscript and for the many constructive comments. Below is a point-by-point response to the general, detailed and line specific comments. The original referee comments are set in normal font and our answers (author comment, AC) are set in blue.

**General comments**

The manuscript entitled "Local scale depositional processes of surface snow on the Greenland ice sheet" by Zuhr et al. present a technique based on ground-based photogrammetry to survey daily the geometry of the snow surface. The experiment was conducted over the course of 78 days, covering an area of 195m2.

Overall the manuscript is well written and present an interesting novel dataset, which to my knowledge only one study in Antarctica had realized. The data show at 1 cm scale the deposition and erosion of snow in a dry cold environment, in which wind is a driving force to the overall landscape. The snow surface is marked by bedforms which are assumed to play a crucial role in the deposition of snow to the ice sheet. Bedforms would be responsible for spatial variability which consequently should be looked at attentively if one is to reconstruct seasonal accumulation rate of precipitation or chemical from past and deeper ice cores. The relevance of the study is therefore justified by the advent of high-resolution description of ice core compositions.

The intent of the manuscript is in its current form unclear. While the title suggests a study on processes concerning snow deposition on the ice sheet, the content corresponds more to a method paper. There is little to no background on snow deposition processes, the description of depositional/erosional events is scattered around various parts focusing on validating a method. In that sense, I would either suggest to change the title to clarify that the content has to do with the validation of a method to measure snow deposition, or the content of the paper should be deeply revised to fit the title meaning and focus on a description of processes. Moreover, this study repeats very closely work done by Picard et al. presented in two prior papers from which very little has been used to inform this study, or even perform a comparative analysis in between Antarctica and Greenland.

If the intent of the manuscript is to validate the method, then the analysis presented seems robust and demonstrating the capability of this simple setup. Though, a thorough and unbiased discussion on advantages and limitations is necessary, including the entire pipeline, from instrumentation design to the final processing of the data, all taking place in the lab, the field

and the office (post processing). However, while the setup has some contextual particularity, using photogrammetry for snow is by now a well-established technique.

Overall, this manuscript presents an interesting dataset from which the authors fall short to extract information describing accumulation processes, informed from the recent literature investigating snow surface morphology shaped by wind. For instance, figure 11 of the manuscript holds interesting data about the snowpack internal structure which is barely exploited. Finally, the main conclusion of the authors is arguable given the data presented. The authors claim that the snow surface become smoother which would corroborate past studies, but the data do not show such trend. Figure 8 shows a surface roughness oscillating twice over the coarse of the study from 4 to 2cm. Little is discussed about it and how this oscillation happens.

**Simon Filhol**

AC: We thank the reviewer for the detailed evaluation of our manuscript and the in-depth comments on the objectives of our study. We will adjust the title to "Local scale deposition of surface snow on the Greenland ice sheet" and focus the manuscript more on the investigation of the processes at the snow surface, and less on the method and the validation thereof. We will shorten and focus the method part on the general photogrammetry SfM method and our specific setup. We will keep only a short accuracy estimate in the method section and move most of the detailed validation (section 2.3) to the appendix where we will summarise the already presented additional validation of the method and associated uncertainties of the data. We agree with the reviewer that our study is similar to the laser scanner study performed by Picard et al. (2016, 2019) at Dome C, East Antarctic Plateau. Even though our data are collected with a slightly different method, covering a much shorter time span and having a study site with different accumulation conditions, our data show comparable behaviors between ERA5 snowfall, wind speed and accumulation characteristics (see Figure 1) as shown in Picard et al. (2019). However, our snow accumulation occurs differently than in East Antarctica. More information on this behavior is presented below and will be added in the manuscript. Overall, we provide a temporal and spatial data set from a study site in Greenland which is relevant for snow accumulation and deposition statistics on a local scale (tens of metres). The data further allow the analysis of (accumulation) noise in ice cores and contribute thereby to a better understanding of proxy data.

**Detailed comments**

The introduction presents well the techniques available to accomplish the goal of the research question, but little attention is given to providing background on the geomorphological parameters of interest. Such addition would help refining the research question. For instance, the recent work by Picard et al. could be used further in grounding this study and using the data presented here to expand our understanding of snow surface morphology evolution in dry, cold, and windy environment of high ice sheet plateau.

AC: Thank you for the detailed feedback on the introduction. Comparing our study to Picard et al. (2019) and referencing our results to his work will be added in more detail in the manuscript. However, our data set is shorter and not as detailed as the one used by Picard and due to data gaps, a detailed comparison with wind characteristics (wind speed, direction and frequency) does not provide reliable conclusions on snow erosion and transport. Nevertheless, our data provide valuable information on local accumulation and deposition statistics for a new study site in northeast Greenland. Figure 1 shows similar patterns as presented in Figure 5 by Picard et al. (2019). In contrast to the data from the East Antarctic Plateau (EAP), our study site receives much more snowfall and we therefore expect different accumulation conditions. According to Picard, accumulation on the EAP is more patchy, whereas our data are more indicative of layer-by-layer accumulation. The similarity of Figure 1 here and Figure 5 in Picard et al. (2019) might indicate similar erosional patterns. We will add this figure as well as a discussion on the comparison to our manuscript. However, we cannot conclusively determine what processes are driving snow accumulation at our study site and how they differ from other areas. More high-resolution data sets from a variety of (polar and alpine) study sites would be needed to quantify the driving processes.

Figure 1: Comparison of different environmental parameters and snowfall specific characteristics. Following Picard et al. (2019), we compare DEM-derived mean daily accumulation as well as the standard deviation of daily accumulation to daily snowfall from

ERA5 and mean daily wind speeds from the nearby PROMICE automatic weather station for the observation period (16.05. - 01.08.2018).

In section 3, the authors chose to first present the signal of interest and explain posteriorly the reason of these changes. I would suggest doing the opposite; first describe the meteorological events relevant to snow accumulation/erosion, and then present the actual change occurring at the snow surfaces as a consequence to the weather events. The connection in between the two can then be drawn and interpreted. Also, a careful description of the wind conditions (speed, direction, frequency) for the period of the study but also at this season in other years would also help contextualizing the relevance of this study. Figure 4 and appendix A1 hint to this direction, the information could be reorganized more effectively to the reader's advantage. AC: Thank you for the detailed thoughts. The starting point of our research was indeed to

compare the meteorological conditions (e.g. wind speed and direction) with our dataset. We agree that it would be very useful if we would be able to predict the snow surface evolution based on the meteorological conditions. However, similar to Picard et al. (2019), we did not find any clear relationships which might be partly due to the shortness of our dataset. Our aim is thus to provide a statistical description of the snow accumulation and redistribution at this site; and for this aim, we think that the current structure is more useful. We agree with your point to add a more detailed description of the wind characteristics and we

We agree with your point to add a more detailed description of the wind characteristics and we will include a wind rose for our observation period as well as for the years 2017 to 2019 (see Figures 2 and 3).

In a second part, you may provide a qualitative description of the surface geomorphology accompanied by one or more photos. The description of surface bedforms in the current second paragraph could be refined. Such description can consider an area with a larger extent than the mapping area itself. It would add context to the maps (that are smaller than many of the typical snow bedforms and deliberately across the main wind direction).

AC: A description of the wider surface geomorphology is not the aim of our study. Nevertheless, we will include a short description of our area in section 3.1 complementing our DEMs. In general, the main focus of our manuscript is on the statistics of snow deposition and accumulation. We deliberately do not focus on single morphological structures and, thus, have not described and discussed them in more detail in the manuscript.

The discussion reads like an in-depth analysis of the method rather than an in-depth discussion of the processes generating the local scale deposition variability. Here the authors argue that the snow surface is being smoothed, which in itself is arguable given the data presented in Figure 8, nevertheless we find no discussion of why and how a rough surface would become smoother (which implies an uneven snow deposition). Therefore, linking the results presented as is (snow height change and surface roughness) to climatic proxies is not clear and missing key connections.

AC: We see the evolution of the surface roughness together with the negative relation between initial snow height and amount of accumulated snow as indicators for a smoothing of the snow surface during our observation period. We agree that the discussion can be improved by adding details on how the snow surface can smooth (e.g. the contribution of wind). Nevertheless, we

think that our results and especially Figure 11 (in the manuscript) allow a discussion of the effects of snow accumulation intermittency on climatic proxies.

In the discussion, there tends to be a bias towards the method used with little of the disadvantages discussed (e.g. need of an operator, possibility of interreferences with the snowpack microstructure, post-processing effort in terms of computational power and manual work). Also, little discussion is done on how it could be improved in terms of processing, or experimental setup (to the exception of NIR). Is it credible to expand this protocol for an entire year? Is this credible given the resources required to accomplish data acquisition to deliverable DEMs?

AC: We agree that the discussion of the method is highlighting the advantages of it and is lacking limitations and disadvantages, e.g. the need for an operator and the difficulty to extend our approach to year-round surveys. We will assess and include disadvantages of the method and present a more profound analysis on the future use and possible extensions of our method.

**Line specific comments**

- L22: "accumulation rates", accumulation rates of what? AC: We will change the wording to "snow accumulation rates".

- L33-35: the self-organization of snow in dunes, ripples and sastrugi results in a heterogenous snow surface and spatial variability of snow depth deposition. There are now more relevant sources describing processes shaping the snow surface in windy environment. I invite the authors reading more carefully papers by Filhol and Sturm, Kochansky et al. and Picard et al. AC: We agree that adding more references to the formation of surface snow structures will improve this section. We are thankful for the references and will add these to the manuscript.

- L43: the terms photogrammetry and structure from motion are often mistaken for being two independent methods, which in the field glaciology most often refer to the same technique. Also, what is meant by "remote sensing products"? photogrammetry and lidar are remote sensing techniques themselves

AC: We agree that our current list of methods is not very consistent. We will rephrase this part and better differentiate between the individual methods.

- L46: are sonic ranger newer than laser scanner or the recent rise of user-friendly photogrammetric software?

AC: Sonic rangers are frequently used at automatic weather stations (AWS) and are easy to operate at remote locations with little maintenance (e.g. Cohen and Dean 2013, Castellani et al. 2015); however, they are not new. We will change 'A newer technique' to 'Another technique' in the manuscript. At the EGRIP camp site, a Campbell Scientific SR50A instrument has been used to record the snow height at the nearby AWS since 2016 as part of the PROMICE network (https://www.promice.dk/WeatherStations.html).

- L56: There is a number of studies demonstrating time-lapse photogrammetry, and some actually applied to snow (See Eltner et al 2017, Filhol et al 2019 and Chakra et al 2019 to cite a few). Despite the illumination challenge to expose snow properly and the logistical challenge to perform measurement in the cold, there is little inherent differences applying such technique to snow than any other type of surfaces.

AC: Thank you for these suggestions. We will extend this part and add more references.

- L66-69: Can you provide the precipitation estimate in the same units (either in SWE or ice layer thickness)

AC: Yes, we will provide all accumulation estimates in the same unit (mm w.eq. yr-1).

- L86: How many images per survey on average? AC: We used on average 60 images per survey/day to generate one DEM. We will add the number of used images to the manuscript.

- L94: 32 to 35 sticks per image or total? How many were you able to obtain GCPs per image? AC: Since our sledge was dragged 1.5-2 m apart from the row of sticks, only two to four sticks (dependent on the spacing between sticks) are visible per image. We will add the field of view of the camera to Figure 1 for a better illustration of the coverage on the photos. After arranging all ~60 images, we were able to detect between 32 and 35 sticks (depending on the image quality and light conditions), all are used as GCPs. This is mentioned in the manuscript as well.

L116-117: The sentences are unclear as the terms have not been clearly defined prior. What do the authors mean by snowdrift? This can either be interpreted as snow accumulation behind an obstacle or the action of snow being mobilized by wind. "Snow drift, erosion and re-distribution" are not three independent processes. If the snow height changes in negative (e.g. the snow surface subsides) this would indicate either erosion or compaction.
AC: Thank you for this observation. We will add this to clarify that we refer to snowdrift as snow being mobilised by wind. We will further indicate that negative snow height changes can be erosion or compaction.

- L120: There exist a variety of ways to describe surface roughness, can you provide the mathematical expression of the surface roughness of choice? AC: We will provide a mathematical definition of the used surface roughness.

- L136: variance is fine, but standard deviation has the advantage to be expressed in the same unit as mean and RMSE. a choice to be considered throughout the manuscript AC: We agree that mentioning the standard deviation is more useful. We will change this throughout the manuscript.

- Figure 3:

- Is the colorscale choice of the first panels a linear gradient (e.g. viridis)? If not the colorscale can introduce representational artefacts. Or why not choosing a divergent colorscale centered over the median value of the first map?

AC: The used color scale was a default topographic color scale. Thank you for the suggestion to use the viridis color scale. We will adapt the color scale to this color palette to reduce representational artefacts.

- I would suggest overlaying a hillshade to the elevation colormap to highlight the surface texture of the three upper panels.

AC: We avoid the use of a hillshade to keep the illustration simple and clear.

- Indicate the main wind direction with an arrow to ease reading of the graph, as wind direction is a prevalent variable to surface texture development and anysotropy. A wind rose showing the period prior and during the time of the study would be even more interesting.

AC: Thank you for the very helpful comments. We will add an arrow indicating the main wind direction as well as wind roses showing the wind conditions during our observation period (Figure 2) and for the years 2017, 2018 and 2019 for comparison (Figure 3).

Figure 2: Hourly wind characteristics (speed and direction) for the observation period from 16.05. - 01.08.2018 recorded by the nearby PROMICE AWS.

---

## Author Response (AR1)

Reply to the Review Comments of Quentin Libois (Referee #1)

on the manuscript

TC-2021-36: Local scale depositional processes of surface snow on the Greenland ice sheet

by Alexandra M. Zuhr et al.

*Thank you for your effort and your careful and detailed review of our manuscript. We appreciate your constructive feedback that will help to improve the manuscript. Below we provide a point-by-point response to all comments. The original referee comments are set in normal font and our answers (author comment, AC) are set in* blue.

**General comments**

This study presents a unique data set of digital elevation models (DEMs) of surface snow near the drilling site of the East Greenland Ice Core Project (EGRIP), acquired near-daily during a whole Summer season (May to August 2018) from a photogrammetry approach. These observations are complemented by more traditional snow height measurements and a variety of meteorological observations. The data are used to extract information on the evolution of average snow height (which increased by 11 cm along the observation period), but also on its spatial variability at the scale of the sampled area (195 m2). It highlights the complexity of spatio-temporal variations, and the poor correlation between precipitation (observed or reported in ERA5 reanalysis) and snow height variations. In particular ~60 % of the deposited snow is at some point removed. This is attributed to the significant role of post-depositional processes, such as snow erosion and subsequent transport by wind. All along Summer, this redistribution results in a reduction of the surface roughness (from 4 to 2 cm), and an overall flattening of the surface. In an extensive discussion section, the impacts of these observations on the proxies used to study climate (e.g. stable water isotopologues) are discussed.

The topic of the study perfectly suits to The Cryosphere because it both presents a novel observation methodology, a novel dataset, and interesting results regarding snow processes. The observations are robust and much care is taken to ensure that the observations are valid, and to quantify the uncertainties. The discussion points to relevant questions related to this study, which for some of them (in particular how and when erosion, transport and re-deposition occurs) could have been a bit more explored with the present dataset. The results are not particularly surprising to people familiar with snow physics in polar regions, and it'd be appreciated that more quantitative comparisons be made with previous similar studies. Besides the new technical approach, more insight about how this study complements the existing literature on the topic would be useful as well. The paper is well written and the methodology clearly described. The multiplicity of observations sometimes makes it difficult to follow, and an updated Figure 1 could certainly help the reader. I recommend this manuscript for publication after these minor issues and the technical details below are addressed, and hopefully after a slightly deeper investigation of the data for the physics of snow erosion and transport.

AC:
We are happy that the manuscript is generally seen as an important and significant work contributing to a better understanding of snow processes. We acknowledge the suggested minor points and carefully addressed them in the revised manuscript as outlined below. We now better discuss our results in the context of previous studies (e.g. comparing directly to Picard et al., 2019) and define more clearly what is unique and similar to the existing literature. We updated Figure 1 for a clearer indication of individual methodologies and improved the technical details as well as the method section. Unfortunately, the temporal resolution and temporal gaps (partly caused by the fact that it was the first season in which this setup was used) limit a robust determination of the origin and the timing of snow erosion, transport and re-distribution. On the other hand, our dataset is designed to and sufficient to determine the statistics of deposition that are relevant for the formation of environmental and climatic records. We extended the interpretation and discussion of this point and also adapted our title to "Local-scale deposition of surface snow on the Greenland ice sheet" to better define the focus of our study.

**Specific comments**

1) The detailed quantitative results of the studies are very useful, but should be better put in the context of other studies performed in regions with similar climatic conditions. While it is clearly pointed that the methodology is original (although the differences with classic setups including only 2 cameras should be better highlighted), the novelty of the results, if any, is not sufficiently put forward. It is currently hard to say which ones among the presented results are really unique to this study.

AC: We extended the discussion on how our study relates to other studies from regions with similar characteristics such as locations in Greenland, in Antarctica but also alpine regions, and also better define the scope of the manuscript (including its title). We also mention the differences to classical 'stereo' camera setups.

2) Ancillary observations (AWS, snow sampling) are widely mentioned but insufficiently used. Wind speed, and possibly direction, could help interpret the variations of surface roughness (including formation of dunes which may still build up in Summer before to be flattened) or the depositional processes. Snow sampling is not detailed (except for its impact on the study area) but could probably help to stress the spatial heterogeneity displayed in Fig. 11 for instance. The comparison of some snow profiles with this figure would be very useful.

AC: During our research process, we analysed the AWS data (wind speed and direction) in detail. We agree that it would be very useful if one could establish clear links between the meteorological conditions and the development of the surface structure. However, we did not identify any clear links, despite a significant negative correlation between the daily wind speed and the mean daily accumulation derived from our data set. We now show more data from the AWS in the appendix of the revised manuscript and an extended discussion on the relation between meteorological parameters and accumulation characteristics. The snow sampling data are still in analysis and thus would be beyond the scope of this manuscript.

3) In general, the manuscript could be shortened by removing some redundancies (in discussion and conclusions), by clarifying the experiments once for all at the beginning, or by selecting the results. This would leave more room to explore the previous suggestions.

AC: In the revised version, we removed redundancies and moved Figure 11 to the result section. We think that the presented choice of the results reflects the main analyses concerning the topic of the manuscript.

4) Although rich the discussion is a bit long and could probably be shortened. Section 4.3 could be moved to the Results Section because it still contains quantitative results not presented earlier on (e.g. Figs. 10 and 11). Section 4.4 highlights the potential impact of the research on the climatic analysis of ice cores but the conclusions are somehow general. More quantitative estimations of the potential impact would help the reader figure out to which extent the results obtained here can question the current analysis techniques.

AC: As suggested we moved Fig. 11 to the result sections and modified the discussion part on the implications for proxy data.

**Technical corrections**

l.6-7 : the contribution of snow re-deposition to noise in climate records from ice cores is put as a primary objective of the paper, but I'm not sure strong quantitative conclusions are reached on that question. Consider reformulating the main objective or rephrasing the conclusions.

AC: We agree that the conclusions are not answering the proposed questions. We reformulated the objectives and revised the conclusions.

l.26 : detail briefly how isotopic composition can be changed

AC: Thank you for pointing out the missing information on changes in the isotopic composition. We added a comment on this in the introduction.

l.28 : "larger" is not clear

AC: We rephrased it to "larger scale processes".

l.31 : maybe remove "deposited"

AC: Is changed as suggested.

l.39 : "mapping" is not clear. Do you mean in space or time ?

AC:  This sentence was removed while rephrasing the paragraph.

l.39 : why is surface roughness important here ?

AC: Surface roughness is influencing the spatial deposition of snowfall, especially during windy snowfall conditions. We added this explanation in the manuscript.

l.40 : I think precipitation intermittency is completely independent of surface processes, such that accumulation intermittency and precipitation intermittency are two distinct things

AC: Precipitation intermittency is not influenced by surface processes, such as erosion or snowdrift. However, precipitation intermittency is part of accumulation intermittency,

because it determines the timing and the amount of snow that is available for re-distribution and transport. We replaced 'precipitation' with 'accumulation'.

l.49 : maybe provide the typical spatial scale of remote sensing observations. For laser altimetry for instance
AC: We specified the spatial scale of current laser altimeter systems to provide a better understanding of the need of small scale methods.

l.51-52 : it's not clear whether SfM is a particular type of photogrammetry or something different
AC: SfM photogrammetry is a technique which uses the photogrammetry approach for the data acquisition and SfM algorithms for the DEM generation. It is already widely used, also for similar studies in the field of glaciology (e.g., Chakra et al., 2019; Filhol et al., 2019). We added a sentence on the technique in the manuscript.

l.53 : if laser scanners do have limitations, maybe state them here. This will support the use of SfM
AC: We extended this section with limitations of laser scanner studies.

l.59-61 : the end of the introduction is incomplete. The objective is not clearly stated, and no outline is provided. Instead some result is provided that should not appear here.
AC: We clarified the objectives and removed the results. However, we do not provide an outline because we see this as a redundant.

l.65 : "with a mean" is awkward. where the mean annual temperature is -29 C ?
AC: Is changed accordingly.

l.67 : what should the reader deduce from the comparisons of accumulation rate vs annual layer thickness ? Are these numbers consistent ?
AC: We put them all to the same unit (mm w.eq. yr-1) for easier comparisons. By showing the different accumulation rate estimates, we want to highlight the high spatial variability in local snow accumulation at our study site.

l.71 : are these data used in the study ? If not, this last sentence is useless
AC: We removed this last sentence.

l.74 : to achieve this
AC: Is changed as suggested.

l.75 : not clear if this is the area covered by one picture or by the whole DEM. Is it dictated by the field of view of the camera ? Clarify the link between the 390m2 and 195m2.
AC: The total area covered by all images per survey is 390m2. Due to the lack of coverage with images and the lower image quality at the rear of the area, we reduced the size of the analysed DEMs to 195m2. We clarified this in the revised manuscript.

Figure 1 : this figure is central to understand all the measurements that are mentioned in the manuscript. Unfortunately it's not very clear. AWS is loosely positioned because the arrow should point towards the camp which is not shown. The scales are loosely defined (e.g. 90 m, 200 m, 39 m) while they could be consistent. The 10m width of the SfM method is not shown. X and y axis could be added. What are the 5 sticks above the 35 sticks in the photogrammetry area ? Add the sledge and orientation of the camera

AC: We clarified the positions and updated Figure 1. However, Figure 1 is still not to scale because it would minimise the size of the study area by expanding the distance to the automatic weather station. However, we want to focus the overview on the relevant sites (Bamboo stakes, SSA stakes and the photogrammetry area). We added the position of the sledge as shown in Figure 2, the approximate field of view of the camera as well as the position of the camp and the validation area (mentioned in the Appendix).

l.76 : "around" does not suggest the sticks are put on a line. Are they ?

AC: We agree that the sentence was not precise. The sticks along the x-axis are set on one line, the surrounding sticks are also positioned on straight lines to create a rectangle. We clarified this in the manuscript.

l.79 : why "almost"? Are the missing days due to technical issues or were they planned ?

AC: No photos were taken on very cloudy days with no visible contrast or during whiteout conditions. These conditions do not allow any snow height reconstructions with optical images. Near infrared would be necessary to extract more information during these weather conditions. Further missing days are caused by (human) errors in the data acquisition. We clarified this in the manuscript.

l.84 : how long does it take to take all pictures ? How many pictures are used for each DEM? Why is the width limited to 10 m ? How was the geometry of the study area chosen? Is it necessary to have that many images, compared to standard photogrammetry with only two or three images ?

AC: The image acquisition itself took about five minutes; however, including the preparation and walking time to and from the camp, the total time effort was about 45 minutes. The width of 10 m was chosen based on thoughts about image quality for DEM generation. During the field period, we realised that the images are not good enough for a DEM generation up to 10 m width; thus, we restricted the analysed area to 5 m width. About 60 images were used for one DEM. If less than 50 to 60 photos were used, no DEM could be generated due to insufficient overlap between successive images. This can be caused by a lack of available surface features to match the images which is a reason for using that many images, instead of two or three, as other studies did. We extended the information on the number of photos in the methods part as well as in the discussion.

l.92 : y=10 m was not properly defined, hence this sentence is hard to understand

AC: We clarified the scales in Figure 1.

Figure 2 is hard to relate to Figure 1. Consider adding the footprint of the camera to help
AC: The footprint of the camera from Figure 2 is added to Figure 1.

l.95 : does it mean that only a transect is used instead of the full 2D domain ?
AC: No, the analysed area is only restricted to the area from y = 0 m to y = 5 m which equals a size of 195m2.n However, due to increasing missing data caused by the snow sampling, many analyses are performed on a band from y = 2.5 m to y = 3 m to ensure constant data quality throughout the observation period.

l.99 : how do you document the snow height at the glass fibers without perturbing the observed area ? Are the sticks out of the final domain ?
AC: The sticks surrounded the observed area and could thus be accessed and measured without perturbing or stepping into the study area.

l.105 : "summarised" is unclear. Averaged ?
AC: Is changed as suggested.

l.106 : the snow sampling was performed for all 35 glass fibers ? What was measured at this occasion? When was it performed ? Is it used in this study ?
AC: We apologise for the unclear information on the snow sampling. The sampling was performed at 30 stick positions along the x-axis every third day throughout the entire observation period. No samples were taken at the remaining five stick positions which are surrounding the study area. The snow samples were measured for stable water isotopic composition and are not used in this study. We added more details on the snow sampling procedure and the resulting surface disturbances in the manuscript.

l.110 : how is snowfall documented and how are samples collected?
AC: Snowfall was manually documented in a spreadsheet. If visual snowfall was observed and/or the snow collection tables (setup described in Steen-Larsen et al., 2014) had snow, we noted this down and sampled the snow. We extended the information on this in the manuscript.

l.111 : I assume snowdrift can be difficult to distinguish from snowfall in human observations as well
AC: We agree that snowfall is visually difficult to distinguish from snowdrift. We therefore analysed the DEMs with regard to the manual snowfall documentation and the ERA5 snowfall as additional indications of snowfall.

Table 1 : why 30 PT sticks here and not 35 ?
AC: 30 sticks were set up along the x-axis, the remaining five sticks are mainly used to provide a reliable geo-referencing of the DEMs and not for further analyses. We clarified the entire setup in the manuscript.

l.115 : shows
AC: Is changed as suggested.

l.121 : not clear what peak-to-peak means, probably the difference between max and min?
AC: Peak to peak refers to the difference between the minimum and the maximum snow height. We added the mathematical equation in the manuscript to clarify the calculation of the surface roughness. We further added a schematic illustrating the individual segments which are used for the surface roughness calculation.

l.131 : why cannot it be done on the main study area ?
AC: We did not want to disturb the study area by adding too many footsteps. Therefore, we set up a second area where we physically walked into the area to establish validation points within the area. We extended the provided information on the validation in the appendix.

l.133 : redundant with just a few lines above
AC: The detailed description on the data validation and uncertainty estimation is moved to the appendix. We kept only a short summary of the data validation in the method part.

l.140 : it is not clear what additional information this section provides compared to the previous sections
AC: We tried to extensively validate our method by analysing different error terms which can arise when e.g. only using GCPs outside of the study area. We therefore included a detailed analysis of potential biases (e.g. doming effects) and mentioned the calculated uncertainties. We restructured the method section and moved the detailed accuracy estimation to the appendix.

l.152 : why was not this sensitivity study performed directly on the study site ?
AC: The validation area has additional control points within the area, not only surrounding points as for the study area. Since we did not want to disturb the study area in addition to the snow sampling, we decided to set up a separate validation area to perform this sensitivity study. We updated the description in the manuscript.

l.170 : sufficient accuracy
AC: Is changed as suggested.

l.171 : here the final estimation of DEM accuracy should be mentioned. Otherwise it's used later on (1,3 cm) without relevant reference.
AC: We added the final DEM accuracy here.

l.180 : it seems that on Panel 2 of Figure 3 the dunes have already vanished
AC: On DOP 36, the snow surface became smoother compared to the beginning of the observation period. This is also shown in the reduced surface roughness towards this day (Figure 10 in the revised manuscript). We discuss this behavior in more detail in the revised manuscript.

Figure 3 : having these x and y axes in Fig. 1 would help a lot. Refer to the section where the areas in grey are used. "Snow sampling scheme" sounds awkward, remove scheme ? Clarify in the text

AC: We removed "scheme" and clarified it in the text accordingly. We now refer to the sections where the grey areas are used and updated Figure 1 by adding x- and y-axes.

(l.106) how frequently such snow sampling were performed, and make it clear whether this corresponds to the readings of snow height at the stakes or not.

AC: The snow sampling was performed every third day. Manual reading of the snow height at the stakes did not always correspond to the snow sampling dates. We clarified this in the text.

l.201 : Reference to Libois et al. (2014) might be relevant here (Figure 2 for instance) or elsewhere

AC: The reference is added.

l.206 : any insight/reference about the quality of ERA 5 snowfall reanalysis over Greenland?

AC: We added a reference to Delhasse et al. (2020) mentioning that ERA5 provides reliable near-surface variables (2 m temperature, 10 m wind speed, energy downward fluxes) for the Greenland Ice Sheet. We find agreement in terms of timing for the ERA5 snowfall product and our documented snowfall (Fig. 4c in the manuscript); however, ERA5 shows slightly more occurrences of snowfall than our documentation. We might have missed snowfall when, for example, snowfall only occurred during the night.

l.216 : not clear whether the consistency is in terms of snowfall occurrence or amount

AC: With 'timing of snowfall' we refer to the occurrence, not the amount. We clarified this in the text.

l.217 : not clear how 0,6 cm should be read in Figure 4c

AC: The amount of 0.6 cm snowfall is derived from the data presented in Figure 4c and converted from mm w.eq. to cm of snowfall (not shown in Figure 4c). We clarified this in the text.

l.218 : it's hard in Figure 5 to see the successive lines. Maybe consider changing color type when erosion occurs

AC: We agree that the color code in Figure 5 was not intuitive. We changed the colours of the initial figure with colours showing a stronger change between DOP 7 (red) and DOP 8 (blue) highlighting the erosive event. Additionally, we added arrows to guide the reader (see revised Figure 6).

[Figure]

Revised Figure 6: Temporal evolution of the relative horizontal snow height profiles from DOP 1 to DOP 12 (20-point running median, averaged 2.5 m-band). Different colours represent the different days as well as the respective mean snow heights in cm, both shown in the legend. Snowfall caused an overall snow height increase from DOP 1 to DOP 7, followed by an erosive event removing the new snow, and exposing the previous surface structure again. Arrows indicate the erosional decrease in the snow height from DOP 7 to DOP 8.

l.222 : a bit unclear, maybe reformulate : " . . . on one fixed day and that on any other day "
AC: This paragraph was rephrased and that specific sentence reworded.

l.224 : the link between RMSE decrease and erosion is not strict. At least RMSE can decrease without erosion (by smoothing for instance). An interesting quantity could be the RMSE between successive DEMs, after removing mean deviation. Maybe RMSD (deviation) would be more appropriate than RMSE here.
AC: We agree that the figure was not as intuitive as we were aiming for. We changed the metric from RMSE to correlation (Fig. 7 in the revised manuscript) and show estimates for all days throughout the season. The plot is now better representing the behaviour of the surface structures and it is also illustrating the relation to changes in the snow height.

[Figure]

Revised Fig. 7: Correlation between the DEM-derived surface structures of a particular day and the surface structure on every following day (coloured points) as well as the overall snow height evolution (black diamonds). The colour-code indicates the DOP of the surface structure to which all subsequent structures are correlated to. The entire DEM area is considered for the correlation calculation. Vertical blue bars indicate an increase in the correlation and a decrease

in the snow height. The second blue bar from the left shows the decrease in snow height from DOP 7 to DOP 8 which is illustrated in Fig. 6.

l.232 : not clear what this area is because it often takes a different name in the Figures and in the text. Is it the full domain or only the 0,5 m band ? Maybe give it a name, like x-transect, or "area A"
AC: It is a good suggestion to name the different surveyed areas. We now refer to the name '2.5m-band' for the averaged area from y = 2.5 m to y = 3 m in the description of the study area and use it throughout the manuscript.

l.246 : roughness has already been defined earlier
AC: We removed the repetition here and defined surface roughness in more detail in Section 2.4 of the revised manuscript.

l.246 : not clear where the wind parallel line is (what x ?) and whether 50 cm refers to the length of the segments, in which case why is that different than the 2,5 m used in the other direction ?
AC: The line parallel to the wind is the y-axis while the x-axis is oriented perpendicular to the main wind direction. For clarity, we now indicate the axes in Figure 1. For the surface roughness estimate parallel to the main wind direction (i.e., along the y-axis), the length of each single segment is 2.5 m (from y = 1 m to y = 3.5 m). We added a schematic to illustrate the considered segments for the surface roughness estimation (Appendix Fig. D1).

l.251 : decrease with time
AC: Changed accordingly.

l.258 : where does this 1,3 cm come from ?
AC: The accuracy of 1.3 cm is the root mean square error between all manually and DEM-derived snow heights at the stick positions throughout the observation period. We moved the entire description of the validation to Appendix B and clarified the different accuracy estimates.

l.259 : given the acquisition is probably fast, acquisition could be more frequent than daily. Maybe remove this detail
AC: Yes, it is possible to acquire images more often than once a day since the data acquisition takes only about 45 minutes, including the walking time to and from the study area. We removed this detail.

l.261 : provide references for the $40m^2$ and $110m^2$
AC: The sizes of $40m^2$ and $110m^2$ are taken from Picard et al. (2016, 2019), respectively. We added the references next to the numbers.

l.263 : Ok, but what's the rationale of having such a particular study area (by the way, it'd be helpful to explain earlier on how these dimensions were chosen/constrained, as a square area would be more understandable), compared to a circular area ?

AC: A circular area from a laser scanner has the disadvantage that the location of interest cannot be changed as quickly as in our setup. Furthermore, the laser itself is a high obstacle which can influence the snow re-distribution and can thus affect the natural snow accumulation conditions. We added these points to the discussion.

l.266 : the main disadvantage remains the fact that you need an operator, although this could probably be made automatic somehow. What would be the result if only 2 cameras were used in an automatic way ?

AC: Indeed, the need of an operator is a large disadvantage. Using only two cameras would probably result in a very small surveyed area since the field of view of one image does not cover an area of 20 m length with a sufficiently high resolution to allow a DEM generation. We extended the discussion addressing the limitations of our method, e.g. the need of a human operator.

l.273 : maybe clarify the human errors, which could be helpful to readers interested in deploying the same kind of instrumentation

AC: Thank you for pointing out that a more detailed description of the human errors can be helpful for readers. We added more details on this.

l.277 : this title is not clear, maybe just remove reliable

AC: We do not agree with the reviewer here. In this section, we present an optimal measurement setup including the number of sticks and their spacing to reliably determine snow accumulation, which we derive from our study area that has been surveyed at high spatial resolution. We reformulated the title to 'Implications for the measurement of snow accumulation' (Section 4.3 in the revised manuscript).

l.288 : please describe where the stakes are placed in these simulations(random distribution, lines etc.)

AC: We use the a line at y=2.5 m as reference and place the chosen number of stakes with the given spacing on this line (no random distribution). We then use all possible combinations with the chosen number of stakes and spacing. Depending on the number of stakes and spacing, we have a changing number of possible combinations which is averaged in the end to simulate a traditionally used stake line. We added more details on the approach in the manuscript.

l.295 : it seems that spacing beyond 5 m is useless in your case, which might be worth pointing. Then, consider providing suggestions, for instance how to maximize the accuracy with a minimum of stakes.

AC: Yes, based on our simulations, a spacing beyond 5 m is not improving the estimate of the snow height change. We now emphasise this in the text. Moreover, according to our experiment, seven sticks with 5 m spacing deliver the optimal setup of stakes which is now mentioned in the manuscript.

Figure 9 : Is the RMSD computed on a different number of mean values for different spacings ? Maybe clarify this

AC: We clarified our approach in the manuscript. Depending on the chosen number of sticks and spacing, we have a different number of possible combinations to put the stakes on the line. Each combination provides a mean snow height change using the chosen parameters (number of sticks and stick distance). We then calculate the RMSE of the DEM-derived mean snow height change (all points along the y = 2.5 m line) to the mean of the simulated stake line.

l.307 : wind speed during the observation period could be advantageously used to explore the drift/deposition events
AC: During the course of the analyses for this manuscript, we also studied the relation between wind characteristics (speed and direction) and changes in the snow height. Following Picard et al. (2019), we performed a more detailed analysis of the relation between wind and accumulation characteristics related to Fig. 5 in their study (Fig. 8 in the revised manuscript). We find similar patterns for our study site compared to the study site on the East Antarctic Plateau. However, we also observe a significant correlation of -0.55 between the daily wind speed and the DEM-derived mean daily accumulation which was not reported by Picard et al (2019). We also added more plots on the wind conditions during our observation period (Appendix Figs. A2 and A3) as well as an extended discussion on possible relationships between wind and snow accumulation.

[Figure]

New Figure 8: Following Fig. 5 in Picard et al. (2019), DEM-derived mean and standard deviation of the daily accumulation are compared to the daily wind speed from the AWS PROMICE and to the ERA5 snowfall product (converted to cm). During data gaps in the DEMs,

the amount of snow accumulation was divided by the number of days to account for daily accumulation. The accumulation conditions at the EGRIP site are remarkably similar to those on the East Antarctic Plateau in some cases, given the differences in accumulation rate. A linear fit (blue line) between wind speed and mean daily accumulation shows a negative correlation of -0.55 between both parameters which is not apparent in Picard et al. (2019).

l.324 : "final snow accumulation" not clear, because precipitation probably governs the final (end of season or yearly average) snow accumulation, but not high frequency variations.
AC: We rephrased this section and moved this part to the beginning of the discussion.

l.325 : 290 kg m-3 seems a bit large for fresh snow. Could you provide more details on how this value was chosen
AC: The snow density of 290 kg m-3 is derived from daily density measurements along the SSA transect. At each of the ten stick positions, the snow density of the top 2.5 cm of snow is measured in addition to the specific surface area (SSA). Snow density data are not further used in this study. We clarified the density value in the revised manuscript.

l.327 : how do ERA5 data suggest that build up is very irregular in time? Not clear
AC: We refer here to the results from the DEM data (as seen in Fig. 12 in the revised manuscript) and the comparison of these to the ERA5 snowfall data. We updated the text.

l.329 : it'd be helpful to know what "local" means for climate studies, and how far can snow be transported in the study area
AC: We agree that the definition of "local" in this context is not straightforward. We now specify local climate signals in the revised manuscript. Furthermore, we analysed different resources to estimate the distance of transported snow; unfortunately, we did not find a conclusive distance for snow transport.

l.339 : consider providing the range of snow ages at the end of the observation period
AC: It is a good point to investigate the range of theoretical snow ages at the end of our observation period. We use the detailed information on snow ages for theoretical profiles of temporal and spatial samplings (Fig. 13 in the revised manuscript) to investigate possible implications for the interpretation of proxy data.

[Figure]

[Figure]

New Figure 13: Theoretical temporal and spatial sampling of different depth intervals (0-0.5 cm, 0-1 cm, 1-2 cm and 2-4 cm) of the internal snow structure along the 2.5 m-band. a) Temporal sampling for ten consecutive days at x=12 m. b) Spatial sampling at eight positions with 5 m spacing along the 2.5 m-band on DOP 69. The y-axes in a) and b) represent the average day of snow accumulation for the respective depth interval.

l.339 : does the snow sampling provide valuable information with regards to the spatial heterogeneity of the layering ?
AC: Stable water isotopes are measured in the snow samples, but the data are not analysed yet. Thus, we cannot conclude on the (isotopic) heterogeneity of the layering. Adding these data would be beyond the scope of this manuscript.

l.345 : the layering does not record each precipitation event, but when snow settles down as a single layer, it probably contains snow with different ages. Somehow there is a "snow reservoir" in between precipitation and settlement, which is fed by precipitation and at some point is incorporated to the snowpack.
AC: Many thanks for these thoughts. It is correct that a single layer, which can consist of precipitated and drifted snow, may contain snow with different ages which probably has implications for the stored climatic signal in the respective snow layer. We extended the discussion about snow settlement.

l.363 : this idea has already been discussed
AC: We restructured the discussion and removed repetitions.

l.363 : how much is strong ? Would you have references (if no measurements) regarding snow transport to compare scales ?
AC: Stable water isotopologues, density data and accumulation rates show large interannual variations on local but also larger scales of e.g. 450 km in North Greenland (e.g. Schaller et al., 2016). The authors also mention the importance of the smoothing of the snow surface. However, data which provide an estimate of snow transport are generally difficult to obtain and we have no references about these scales for Greenland. We restructured and extended the discussion on implications for the reconstructions of proxy data.

l.371 : could the images be used to identify very local re-deposition (within the same observed area) ?
AC: In our opinion, the images are not sufficient to identify local re-deposition within our study area. We cannot definitely link a re-deposited snow particle to its origin from a spot within our study area.

l.390 : are you sure that your observation of dunes vanishing in Summer is representative ? Could it be that you studied a singular year ? Were the wind statistics in agreement with longer term observations?
AC: Since our data cover only three months, we cannot conclude on the seasonal behaviour of surface features. However, a study from Summit, Greenland (Albert and Hawley, 2002), showed

a similar behaviour of vanishing dunes during summer. Our observation period is comparable with regard to wind speed and direction as shown by the wind statistics from the nearby AWS for the years 2017 to 2019. We also compare the wind speed frequency during our observation period to the winter months (December, January, February) and see that the winter months generally show higher wind speeds (see Appendix Fig. A2 and A3 in the revised manuscript) which can enhance the formation of sastrugi. Thus, the meteorological data would support the hypothesis that our data present representative observations.

l.478 : missing beginning of sentence
AC: Thank you for pointing this out. We updated the acknowledgements.

References

Chakra, C. A., Gascoin, S., Somma, J., Fanise, P., and Drapeau, L.: Monitoring the Snowpack Volume in a Sinkhole on Mount Lebanon using Time Lapse Photogrammetry, Sensors, 19, https://doi.org/10.3390/s19183890, https://www.mdpi.com/1424-8220/19/18/3890, 2019.

Delhasse, A., Kittel, C., Amory, C., Hofer, S., van As, D., S. Fausto, R., and Fettweis, X.: Brief communication: Evaluation of the near-surface climate in ERA5 over the Greenland Ice Sheet, The Cryosphere, 14, 957–965, https://doi.org/10.5194/tc-14-957-2020, https://tc.copernicus.org/articles/14/957/2020/, 2020.

Filhol, S. and Sturm, M.: The smoothing of landscapes during snowfall with no wind, Journal of Glaciology, 65, 173–187, https://doi.org/10.1017/jog.2018.104, 2019.

Libois, Q., Picard, G., Arnaud, L., Morin, S., & Brun, E. (2014). Modeling the impact of snow drift on the decameter-scale variability of snow properties on the Antarctic Plateau. Journal of Geophysical Research: Atmospheres, 119(20), 11-662.

Schaller, C. F., Freitag, J., Kipfstuhl, S., Laepple, T., Steen-Larsen, H. C., and Eisen, O.: A representative density profile of the North Greenland snowpack, The Cryosphere, 10, 1991–2002, https://doi.org/10.5194/tc-10-1991-2016, 2016.

Steen-Larsen, H. C., Masson-Delmotte, V., Hirabayashi, M.,Winkler, R., Satow, K., Prié, F., Bayou, N., Brun, E., Cuffey, K. M., Dahl-Jensen, D., Dumont, M., Guillevic, M., Kipfstuhl, S., Landais, A., Popp, T., Risi, C., Steffen, K., Stenni, B., and Sveinbjörnsdóttir, A. E.: What controls the isotopic composition of Greenland surface snow?, Climate of the Past, 10, 377–392, https://doi.org/10.5194/cp-10-377-2014, 2014.

on the manuscript

TC-2021-36: Local scale depositional processes of surface snow on the Greenland ice sheet

by Alexandra M. Zuhr et al.

*We are very thankful for the detailed and thorough review of our manuscript and for the many constructive comments. Below is a point-by-point response to the general, detailed and line specific comments. The original referee comments are set in normal font and our answers (author comment, AC) are set in* blue*.*

**General comments**

The manuscript entitled "Local scale depositional processes of surface snow on the Greenland ice sheet" by Zuhr et al. present a technique based on ground-based photogrammetry to survey daily the geometry of the snow surface. The experiment was conducted over the course of 78 days, covering an area of 195m2.

Overall the manuscript is well written and present an interesting novel dataset, which to my knowledge only one study in Antarctica had realized. The data show at 1 cm scale the deposition and erosion of snow in a dry cold environment, in which wind is a driving force to the overall landscape. The snow surface is marked by bedforms which are assumed to play a crucial role in the deposition of snow to the ice sheet. Bedforms would be responsible for spatial variability which consequently should be looked at attentively if one is to reconstruct seasonal accumulation rate of precipitation or chemical from past and deeper ice cores. The relevance of the study is therefore justified by the advent of high-resolution description of ice core compositions.

The intent of the manuscript is in its current form unclear. While the title suggests a study on processes concerning snow deposition on the ice sheet, the content corresponds more to a method paper. There is little to no background on snow deposition processes, the description of depositional/erosional events is scattered around various parts focusing on validating a method. In that sense, I would either suggest to change the title to clarify that the content has to do with the validation of a method to measure snow deposition, or the content of the paper should be deeply revised to fit the title meaning and focus on a description of processes. Moreover, this study repeats very closely work done by Picard et al. presented in two prior papers from which very little has been used to inform this study, or even perform a comparative analysis in between Antarctica and Greenland.

If the intent of the manuscript is to validate the method, then the analysis presented seems robust and demonstrating the capability of this simple setup. Though, a thorough and unbiased discussion on advantages and limitations is necessary, including the entire pipeline, from instrumentation design to the final processing of the data, all taking place in the lab, the field

and the office (post processing). However, while the setup has some contextual particularity, using photogrammetry for snow is by now a well-established technique.

Overall, this manuscript presents an interesting dataset from which the authors fall short to extract information describing accumulation processes, informed from the recent literature investigating snow surface morphology shaped by wind. For instance, figure 11 of the manuscript holds interesting data about the snowpack internal structure which is barely exploited. Finally, the main conclusion of the authors is arguable given the data presented. The authors claim that the snow surface become smoother which would corroborate past studies, but the data do not show such trend. Figure 8 shows a surface roughness oscillating twice over the coarse of the study from 4 to 2cm. Little is discussed about it and how this oscillation happens.

Simon Filhol

AC: We thank the reviewer for the detailed evaluation of our manuscript and the in-depth comments on the objectives of our study. We adjust the title to "Local-scale deposition of surface snow on the Greenland ice sheet" and focus the manuscript more on the investigation of the processes at the snow surface, and less on the method and the validation thereof.
We shortened and focused the method part on the general SfM photogrammetry method and our specific setup. We moved most of the validation details to the Appendix (see Appendix B in the revised manuscript) summarising the validation and accuracy estimation of the method. A short summary is however still presented in section 2.2.
We agree with the reviewer that our study is similar to the laser scanner study performed by Picard et al. (2016, 2019) at Dome C on the East Antarctic Plateau. Even though our data are collected with a slightly different method, covering a much shorter time span and having a study site with different accumulation conditions, our data show comparable patterns between ERA5 snowfall, wind speed and accumulation characteristics (see Figure 8 in the revised manuscript) as shown in Picard et al. (2019). More information on this analysis is presented below and discussed in the revised manuscript.
Overall, we provide a temporal and spatial data set from a study site in Greenland which is relevant for snow accumulation and deposition statistics on a local scale (tens of metres). Our data allow the analysis of potential accumulation noise in ice cores and contribute thereby to a better understanding of proxy data.

**Detailed comments**

The introduction presents well the techniques available to accomplish the goal of the research question, but little attention is given to providing background on the geomorphological parameters of interest. Such addition would help refining the research question. For instance, the recent work by Picard et al. could be used further in grounding this study and using the data presented here to expand our understanding of snow surface morphology evolution in dry, cold, and windy environment of high ice sheet plateau.

AC: Thank you for the detailed feedback on the introduction. Comparing our study to Picard et al. (2019) and referencing our results to his work is presented in the revised manuscript (Fig. 8 and in the discussion). However, our data set is shorter, covering only one season compared to several years, and not as detailed as the one used by Picard et al. (2019). Nevertheless, our data provide valuable information on local accumulation and deposition statistics for a new study site in northeast Greenland. Figure 8 in the revised manuscript shows similar patterns as presented in Figure 5 by Picard et al. (2019). In contrast to the data from the East Antarctic Plateau (EAP), our study site receives much more snowfall and we therefore expect different accumulation conditions. We find a negative correlation between wind speed and the mean daily accumulation which is not the case in the laser scanner study.

Accumulation on the EAP is rather "patchy" while alpine accumulation conditions are rather "layer-wise". Our study area presents accumulation conditions in between antarctic and alpine conditions and shows both behaviours. We discuss the different accumulation and erosion characteristics in more detail in the discussion. But we cannot conclusively determine what processes are driving snow accumulation at our study site and how they differ from other areas. More long-term high-resolution data sets from a variety of (polar and alpine) study sites would be needed to thoroughly quantify the driving processes.

[Figure]

New Figure 8: Following Fig. 5 in Picard et al. (2019), DEM-derived mean and standard deviation of the daily accumulation are compared to the daily wind speed from the AWS

PROMICE and to the ERA5 snowfall product (converted to cm). During data gaps in the DEMs, the amount of snow accumulation was divided by the number of days to account for daily accumulation. The accumulation conditions at the EGRIP site are remarkably similar to those on the East Antarctic Plateau in some cases, given the differences in accumulation rate. A linear fit (blue line) between wind speed and mean daily accumulation shows a negative correlation of -0.55 between both parameters which is not apparent in Picard et al. (2019).

In section 3, the authors chose to first present the signal of interest and explain posteriorly the reason of these changes. I would suggest doing the opposite; first describe the meteorological events relevant to snow accumulation/erosion, and then present the actual change occurring at the snow surfaces as a consequence to the weather events. The connection in between the two can then be drawn and interpreted. Also, a careful description of the wind conditions (speed, direction, frequency) for the period of the study but also at this season in other years would also help contextualizing the relevance of this study. Figure 4 and appendix A1 hint to this direction, the information could be reorganized more effectively to the reader's advantage.

AC: Thank you for the detailed thoughts. The starting point of our research was indeed a comparison between the meteorological conditions (e.g. wind speed and direction) and our dataset. We agree that it would be very useful if we would be able to predict the snow surface evolution based on the meteorological conditions. However, similar to Picard et al. (2019), we did not find any clear relationships for most of the parameters which might be partly due to the shortness of our dataset. However, we find a negative correlation between wind speed and the DME-derived mean daily accumulation which hints to different accumulation conditions at our site compared to the East Antarctic Plateau.

Our aim is to provide a statistical description of the snow accumulation and redistribution at this site; and for this aim, we think that the current structure is more useful.

We agree with your point and added a more detailed description of the wind characteristics and included a wind rose for our observation period as well as for the years 2017 to 2019 (see Appendix Figures A2 and A3 in the revised manuscript).

In a second part, you may provide a qualitative description of the surface geomorphology accompanied by one or more photos. The description of surface bedforms in the current second paragraph could be refined. Such description can consider an area with a larger extent than the mapping area itself. It would add context to the maps (that are smaller than many of the typical snow bedforms and deliberately across the main wind direction).

AC: A description of the wider surface geomorphology is not the aim of our study. Nevertheless, we extended the description of our area in section 3.1 complementing our DEMs. In general, the main focus of our manuscript is on the statistics of snow deposition and accumulation. We deliberately do not focus on single morphological structures and, thus, have not described and discussed them in more detail in the manuscript.

The discussion reads like an in-depth analysis of the method rather than an in-depth discussion of the processes generating the local scale deposition variability. Here the authors argue that the snow surface is being smoothed, which in itself is arguable given the data presented in Figure 8, nevertheless we find no discussion of why and how a rough surface would become

smoother (which implies an uneven snow deposition). Therefore, linking the results presented as is (snow height change and surface roughness) to climatic proxies is not clear and missing key connections.

AC: We see the evolution of the surface roughness together with the negative relation between initial snow height and amount of accumulated snow as indicators for a smoothing of the snow surface during our observation period. We think that our results and especially Figure 11 allow a discussion of the effects of snow accumulation intermittency on climatic proxies.
We agree that the discussion can be improved. We restructured the entire discussion section and added more details on e.g. the contribution of wind related to changes in surface structures.

In the discussion, there tends to be a bias towards the method used with little of the disadvantages discussed (e.g. need of an operator, possibility of interreferences with the snowpack microstructure, post-processing effort in terms of computational power and manual work). Also, little discussion is done on how it could be improved in terms of processing, or experimental setup (to the exception of NIR). Is it credible to expand this protocol for an entire year? Is this credible given the resources required to accomplish data acquisition to deliverable DEMs?

AC: We agree that the discussion of the method is highlighting the advantages of it and is lacking limitations and disadvantages, e.g. the need for an operator and the difficulty to extend our approach to year-round surveys. We assessed and included disadvantages of the method in the revised manuscript and we now present a more profound analysis on the future of our method.

**Line specific comments**

- L22: "accumulation rates", accumulation rates of what?
AC: We changed the wording to "snow accumulation rates".

- L33-35: the self-organization of snow in dunes, ripples and sastrugi results in a heterogenous snow surface and spatial variability of snow depth deposition. There are now more relevant sources describing processes shaping the snow surface in windy environment. I invite the authors reading more carefully papers by Filhol and Sturm, Kochansky et al. and Picard et al.
AC: We agree that adding more references to the formation of surface snow structures will improve this section. We are thankful for the references and added these to the manuscript.

- L43: the terms photogrammetry and structure from motion are often mistaken for being two independent methods, which in the field glaciology most often refer to the same technique. Also, what is meant by "remote sensing products"? photogrammetry and lidar are remote sensing techniques themselves
AC: We agree that our current list of methods is not very consistent. We revised this part and differentiate now between individual methods.

- L46: are sonic ranger newer than laser scanner or the recent rise of user-friendly photogrammetric software?

AC: Sonic rangers are frequently used at automatic weather stations (AWS) and are easy to operate at remote locations with little maintenance (e.g. Cohen and Dean 2013, Castellani et al. 2015); however, they are not new. We rephrased this sentence in the revised manuscript. At the EGRIP camp site, a Campbell Scientific SR50A instrument has been used to record the snow height at the nearby AWS since 2016 as part of the PROMICE network (https://www.promice.dk/WeatherStations.html).

- L56: There is a number of studies demonstrating time-lapse photogrammetry, and some actually applied to snow (See Eltner et al 2017, Filhol et al 2019 and Chakra et al 2019 to cite a few). Despite the illumination challenge to expose snow properly and the logistical challenge to perform measurement in the cold, there is little inherent differences applying such technique to snow than any other type of surfaces.

AC: Thank you for these suggestions. We extended this part and added more references.

- L66-69: Can you provide the precipitation estimate in the same units (either in SWE or ice layer thickness)

AC: Yes, we now provide all accumulation estimates in the same unit (mm w.eq. yr-1).

- L86: How many images per survey on average?

AC: We used on average 60 images per survey/day to generate one DEM. We added the number of used images to the manuscript.

- L94: 32 to 35 sticks per image or total? How many were you able to obtain GCPs per image?

AC: Since our sledge was dragged 1.5-2 m apart from the row of sticks, only two to four sticks (dependent on the spacing between sticks) are visible per image. We added the field of view of the camera to Figure 1 for a better illustration of the coverage on the photos. After arranging all ~60 images, we were able to detect between 32 and 35 sticks (depending on the image quality and light conditions), all visible sticks were used as GCPs. This is now mentioned in the manuscript as well.

- L116-117: The sentences are unclear as the terms have not been clearly defined prior. What do the authors mean by snowdrift? This can either be interpreted as snow accumulation behind an obstacle or the action of snow being mobilized by wind. "Snow drift, erosion and re-distribution" are not three independent processes. If the snow height changes in negative (e.g. the snow surface subsides) this would indicate either erosion or compaction.

AC: Thank you for this observation. We clarified that we refer to snowdrift as snow being mobilised by wind. We also indicate that negative snow height changes can be erosion or compaction.

- L120: There exist a variety of ways to describe surface roughness, can you provide the mathematical expression of the surface roughness of choice?
AC: We provide a mathematical equation and extended the analysis of the surface roughness in the revised manuscript.

- L136: variance is fine, but standard deviation has the advantage to be expressed in the same unit as mean and RMSE. a choice to be considered throughout the manuscript
AC: We agree that mentioning the standard deviation is more useful. We changed the variance estimates to standard deviation estimates throughout the manuscript.

- Figure 3:
- Is the colorscale choice of the first panels a linear gradient (e.g. viridis)? If not the colorscale can introduce representational artefacts. Or why not choosing a divergent colorscale centered over the median value of the first map?
AC: The used color scale was a default topographic color scale. Thank you for the suggestion to use the viridis color scale. We adapted the color scale to this color palette to avoid representational artefacts.

- I would suggest overlaying a hillshade to the elevation colormap to highlight the surface texture of the three upper panels.
AC: We avoid the use of a hillshade to keep the illustration simple and clear.

- Indicate the main wind direction with an arrow to ease reading of the graph, as wind direction is a prevalent variable to surface texture development and anysotropy. A wind rose showing the period prior and during the time of the study would be even more interesting.
AC: Thank you for the very helpful comments. We added an arrow indicating the main wind direction as well as wind roses showing the wind conditions during our observation period and for the years 2017, 2018 and 2019 for comparison (see Appendix Figures A2 and A3).

- You choose the zero-level to be the bottom left corner, but why not using the median elevation of your first day? Over very large area we can expect the mean and median surface height to converge, but given the scale of the study area and the size of dunes and sastrugi, the median will be less prone to bias. Then the DOP36 and DOP37 maps (if using the same colorscale as it is) would show areas where snow accumulated or was eroded in relation to a reference plane closer to where the average physical "zero plane" is.
AC: The zero-level was chosen arbitrarily but could be changed of course. However, since we are more interested in the relative changes of the snow height and the statistics thereof, and less in the absolute snow height increase, the absolute zero-level does not affect our analysis. However, we changed the color scale to the viridis color palette.

- I found nowhere a plot showing time series of snow height for each pixel (or a substantial subsample) as in Picard et al. (2019). Figure 4 panel b) shows an aggregate of this value, but why not showing in the background the entire area of interest.

AC: This is a good point. We added plots showing the snow height evolution for each pixel of our study area (see Appendix Figures C1 and C3 in the revised manuscript). The number of pixel with a value decreases during the observation period due to increasing missing data caused by footsteps and sampling positions. To avoid a bias by missing data, the aggregate in Figure 4b is the average of all individual pixel.

- L179-180: This statement is not obvious to me. first the amplitude of snow height seems to be reduced greatly in between DOP1 and DOP36, and second,
AC: The amplitude decreased between DOP 1 and DOP 36 from 21.8 to 12 cm. The amplitude on DOP 78 accounts to 13.8 cm, which is similar to DOP 36 and less than in the beginning of the observation period. In the revised manuscript, we elaborate more on different time spans during the observation period and present different phases of snow erosion and accumulation.

- L181: "show" -> showing
AC: Is changed as suggested.

- L187: remove "further"
AC: We removed "further".

- L186-188: Given the small amount of accumulation and the spatial interdependency of snow accumulation in relation the wind field, could the human footsteps have created depression in which the snow would be trapped and therefore affecting the general accumulation/deposition pattern of the snow, or even the snow microstructure (local compaction). Could the authors provide a rough estimate of the volume of snow that this could represent in relation to the volume accumulated in the rest of the area. Also, what about the footsteps along the transect created while pulling the sled? Providing the average footstep depth would be a great indicator of the underlying snow hardness as well.
AC: Thank you for your comments on the effect of the footsteps on accumulation and deposition patterns. These are valid thoughts about snow re-distribution possibly caused by our actions close to the study area. We carefully closed the snow sampling positions and footsteps and tried to smoothen and flatten the surface as much as possible to avoid disturbances. The sampling positions are downwind; thus, we expect the influence of these positions to be minimal since the wind was blowing quite constantly from the same direction. We added a wind rose to the manuscript showing wind speed and directions during the observation period (see Appendix Figure A2). The footsteps along the transect while pulling the sledge were not very deep and the snow surface was not very soft. We therefore do not expect effects on the snow accumulation. We added more information on the footsteps to the manuscript as well.

- L206-212: Is there a precipitation gauge or distrometer in the area?
AC: There is unfortunately no precipitation gauge or distrometer in the area or in the vicinity of the EGRIP camp.

- L214: Why do you not use all the DEMs and only 10 out of the 34?
AC: We use only the first ten DEMs and not all 37 because we want to highlight the increase and subsequent decrease of the snow surface. Including all 37 DEMs into one line-plot reduces the visibility of single lines and events and the focus would rather be on the overall snow height increase instead of the day-to-day variability. We added a graph showing all 37 DEMs to the appendix (see Figure C2).

- Figure 4: the legends do not indicate if the graph show mean or median values. Also, points from the DEMs, bamboo forest, and SSA stick could include error bars.
AC: The graph in Fig. 4b shows mean values. This information is now specifically added to the figure caption. We now also show the uncertainties for each method (e.g. finite number of averaged data points).

- Figure 6: The intent of this figure is unclear as it shows similar information than 4b
AC: The intention of Figure 6 was to illustrate the similarity of surface structures from one day to other days, thereby indicating changes in the overall surface structure due to specific events. Figure 6 contains only data from the DEM data set while Figure 4b also includes the overall evolution of individual snow height estimates from other methods, however, no information on the surface structure within the study area. We revised the content of this figure and present now the correlation between DEM-derived structures on different days (Figure 7 in the revised manuscript).

[Figure]

Revised Fig. 7: Correlation between the DEM-derived surface structures of a particular day and the surface structure on every following day (coloured points) as well as the overall snow height evolution (black diamonds). The colour-code indicates the DOP of the surface structure to which all subsequent structures are correlated to. The entire DEM area is considered for the correlation calculation. Vertical blue bars indicate an increase in the correlation and a decrease in the snow height. The second blue bar from the left shows the decrease in snow height from DOP 7 to DOP 8 which is illustrated in Fig. 6.

- Section 3.4: Why not including the entire DEM region in the plot of figure 7c and then focus on the three areas of choice (top of dune, troughs)

AC: We did not include the entire DEM in Figure 7 and especially 7c because the area under consideration would decrease throughout the season with increasing missing data caused by the snow sampling and foot steps. We therefore focus on four sub-areas representing the different snow surface height conditions at the beginning of our observation period in our study area. Nevertheless, we added a plot for the entire study area (see Appendix Figure C3).

- Section 3.5:
- Why the choice of 2.5m peak amplitude?

AC: We chose the 2.5 m peak amplitude for the surface roughness calculation in order to provide results comparable to the study by Albert and Hawley (2002) from Summit, Greenland. Their study site has similar environmental conditions as the EGRIP camp site and is therefore suitable for a comparison of surface roughness estimates. We added the mathematical equation describing the surface roughness calculation and also present a roughness estimate for the larger scale undulations.

- While the surface roughness decrease is more pronounced in the direction perpendicular to the wind, it is very interesting to observe that the roughness had reached 2cm by DOP 36, and then rose back to 4cm to then sharply drop back to 2cm. Would you have further insight as to why? It would actually indicate that the surface roughness magnitude can change sharply and does not necessarily converge to a given value. At least I see no such evidence happening in the data presented here.

AC: The surface roughness indeed decreased to 2 cm already at DOP 36 with a subsequent increase. We also investigated the variability of single 2.5 m-peak amplitudes (see Appendix Figure D2) and observe a decrease in the spread of values from DOP 36 to DOP 78 which indicates an overall decrease in the surface roughness towards the end of our observation period.

To close the data gap between DOP 38 and DOP 56, we added the standard deviation across the manually measured PT sticks and use this as an alternative indicator of surface undulations. This measure shows no changes in the strength of surface undulations suggesting that the decrease is mainly until DOP 38 followed by variations around 2 cm.

- Section 4.1: The authors bring many advantages of the method but too few limitations. For instance, how intrusive is this method? Have you considered the effort (post processing time) and the resources (computational, and data storage) required to process photogrammetric data in comparison to other methods? The method proposed has many advantages, but a fair assessment including the entire workflow (and not simply the fieldwork) is necessary.

AC: We agree that section 4.1 is addressing the advantages of our method rather than presenting an independent evaluation and assessment of our method. We extended the discussion with limitations and disadvantages of our method in this section.

- Section 4.2: prior you used the term RMSE and not RMSD. But overall, I do not see the use of this sub-analysis in respect to the title of the manuscript, as it refers to local scale processes rather than method to estimate local-scale snow deposition.

AC: We do not agree that this section does not fit the title of the manuscript. The small-scale depositional processes, which we visualise and analyse with our photogrammetry method, cause strong spatial variability in local snow accumulation, which affects the representativity of point measurements (snow stakes) in terms of the average regional snow accumulation. Here, we explicitly utilise our data to evaluate the optimal setting of snow stakes to obtain representative snow accumulation measurements. For consistency, we changed RMSD to RMSE.

- L300: providing your own wind analysis would be more convincing, as explained in the comments above.

AC: During the course of the analyses for this manuscript, we also studied the relation between wind characteristics (speed and direction) and changes in the snow height. Following Picard et al. (2019), we performed a more detailed analysis of the relation between wind and accumulation characteristics (Fig. 8 in the revised manuscript). We also added more plots on the wind conditions during our observation period (Appendix Figs. A2 and A3) as well as an extended discussion on possible relationships between wind and snow accumulation.

- L301: Are the spatial scale of the two studies (this one and van der Veen) comparable? And are we actually convinced that this surface roughness decrease is a trend and not a coincidence to the period of experimentation as the intermittent points of figure 8 would indicate?

AC: van der Veen et al. (2009) studied the northern part of the Greenland Ice Sheet; thus, a larger area than covered by our study. They also used a different mathematical expression to calculate surface roughness. Additionally, van der Veen et al. (1998) used airborne laser data from May 1995 to assess the surface roughness for central Greenland including the drilling site GRIP and the Summit location. Their result suggests that no reduction in surface roughness occurs during the summer season in central Greenland. However, the results from Albert and Hawley (2002) who studied local surface roughness on scales similar to our scale, found a similar behaviour as we do. The different results of surface roughness highlight the natural complexity of this parameter and the lack of clear information.

To this end, our study contributes to an increasing understanding of surface roughness by providing our local-scale estimate which suggests a change of the surface roughness from 4-5 to about 2 cm in the end of our observation period as an indication of a smoothing of the surface. We added the mentioned discussion on the different spatial scales to the manuscript. The shortness of our measurement period of only one season does not allow us to clearly distinguish between variability (= the trend is a coincidence) or a seasonal trend. A repetition of our or similar approaches for other years and seasons would be needed to quantify the seasonal behaviour of surface roughness on the Greenland Ice Sheet.

- L305: The link between lower wind speed and smoother surface is unclear as stated. As long as there is snow transport by wind, we know that surface bedforms are generated. Is the link

between wind speed and surface roughness this simple? Snow erodibility is another important variable contributing greatly to the processes at play.

AC: We agree that lower wind speeds in summer are not the only parameter which can lead to a reduction in surface roughness. Thus, the link between wind speed and smoother surface is not as straightforward as it was stated in our manuscript. We removed the sentence and clarified that more parameters (e.g. snowdrift, wind conditions, temperature, humidity, metamorphism) are influencing the conditions and erodibility of snow surfaces as well and may therefore contribute to changes in the surface roughness over the course of a year. Nevertheless, the formation of sastrugi during the winter time with higher wind speeds can still be a significant contribution to the surface structures at our study site. We extended the analysis and the discussion on the relation between wind and accumulation characteristics by e.g. comparing our results to locations with similar conditions (e.g., Antarctica, alpine regions).

- L319: The dataset presented contains no information of micro-scale properties to the exception of the snow surface geometry.

AC: We agree that our data set does not provide information of micro-scale properties and we therefore did not include any analysis about these properties in the manuscript.

- L317 & 331: "surface snow" -> snow surface. (there are multiple instances throughout the manuscript, and the title itself)

AC: On some occasions, we changed the wording throughout the document. For other occasions, we kept 'surface snow' as it fits better the described processes.

- Figure 11: This figure is fascinating! Throughout the paper you focus on the snow surface geometry. In this graph you show the internal structure of the snowpack derived from your surface measurements. This internal structure is a lot more relevant to ice core data than the surface itself, isn't it? Why not focusing the study on the geometrical properties of these internal layers rather than focusing on net snow height changes and snow surface roughness only? If one was to plan a study to retrieve past seasonal precipitation rates, how many ice cores would be needed, or how large of a sample would be needed to overcome the 2D variability of the snow internal layering? Those are simple example questions that could bring relevance to this study linking snow deposition at the surface and the internal structure of the snowpack.

AC: Thank you very much for these thoughts. We agree that this figure is a key result of the manuscript. It highlights the main message of our study, namely the heterogeneous internal structure of the upper snowpack, and provides valuable insights in the deposition structure. We also agree that it should be presented earlier in the manuscript and discussed in more detail. We moved this part to the results section and analysed the implications in more detail by providing a discussion (section 4.2 in the revised manuscript) on theoretical temporal and spatial samplings along our study area and their influence on the interpretation of proxy data from firn and ice cores.

- L365: The statement is unclear. Can wind scouring be defined in relation to previous terminology (aka erosion)? Why are those data not suited for this? They seem quite appropriate given that multiple wind drifting events had occurred during the period of interest.

AC: Wind scouring describes a process where winds persistently carry away snow from the surface by erosion and sublimation, especially at steep surface slopes. This special and complex (depositional) process is most important in low accumulation areas and can be mapped with e.g. radar stratigraphy which detects metamorphosed layers. A fraction of the eroded snow can also fill topographic troughs downwind (Das et al., 2013). With our data set, we are not able to detect whether normal snow erosion and re-distribution or wind scouring occurred because we cannot resolve the micro-properties of the snow surface. We removed this part from the discussion because we cannot contribute to this topic with our data set.

- L366: Was estimating the amount of climatic signal mixing even a stated goal of this study?
AC: One aim of our study is to provide an outlook on the implications of snow erosion on the interpretation of reconstructed climatic signals from ice cores. We agree that the objectives are not clearly outlined. We improved the outline of our objectives as well as the conclusion.

- L387: misuse of "further" like in few other instances in the manuscript
AC: We removed "further" and checked the manuscript thoroughly.

- L387-389: The statement is not evident given data presented in Figure 8 and 11. Both show a variable internal structure of the snowpack that must originate from some processes. If not snow bedforms, then what? In a way this paragraph and the previous one are contradicting each other.
AC: We rephrased the entire conclusion section.

- Figure A1: Can you indicate when snow transport occurred?
AC: We indicate the wind speed threshold of 4 m/s which enables snow transport. However, we cannot confirm that snow transport always occurred during these periods.

References

Eltner, A., Kaiser, A., Abellan, A., & Schindewolf, M. (2017). Time lapse structure-from motion photogrammetry for continuous geomorphic monitoring. Earth Surface Processes and Landforms, 42(14), 2240-2253.

Filhol, S., Perret, A., Girod, L., Sutter, G., Schuler, T. V., & Burkhart, J. F. (2019). Time-Lapse Photogrammetry of Distributed Snow Depth During Snowmelt. Water Resources Research, 55(9), 7916-7926.

Chakra, C. A., Gascoin, S., Somma, J., Fanise, P., & Drapeau, L. (2019). Monitoring the snowpack volume in a sinkhole on Mount Lebanon using time lapse photogrammetry. Sensors, 1(18), 3890.

Picard, G., Arnaud, L., Caneill, R., Lefebvre, E., & Lamare, M. (2019). Observation of the process of snow accumulation on the Antarctic Plateau by time lapse laser scanning. The Cryosphere, 13(7), 1983-1999.

Cohen, L. and Dean, S.: Snow on the Ross Ice Shelf: comparison of reanalyses and observations from automatic weather stations, The Cryosphere, 7, 1399–1410, https://doi.org/10.5194/tc-7-1399-2013, 2013.

Castellani, B. B., M. D. Shupe, D. R. Hudak, and B. E. Sheppard (2015), The annual cycle of snowfall at Summit, Greenland, J. Geophys. Res. Atmos., 120, 6654–6668, doi:10.1002/2015JD023072.

Das, I., Bell, R., Scambos, T. *et al.* Influence of persistent wind scour on the surface mass balance of Antarctica. *Nature Geosci* 6, 367–371 (2013). https://doi.org/10.1038/ngeo1766

---

## Referee Report (RR1)

2nd Review Zuhr et al. 2021

The focus and structure of the manuscript has been tremendously improved following the first set of comments. The manuscript gained in clarity and focus. The choice of the title is now better suited to the stated goal of the manuscript. Th introduction better describe the goal of the study, and the conclusion is more nuanced. With this in mind, the changes brought in response to the initial set of comments are to my judgment satisfactory. Please see the few minor typos caught during this last read.

L78: imaging → images
L78: I still struggle with the term "surface snow" rather than "snow surface". Have the authors run this terminology to an english native speaker to check the grammar correctness of this term?
L133: sakes → stakes
L135: increase the snow height
L170: This is an unclear description. Dunes and sastrugi are two distinct bedforms. Sastrugi are often carved out of previously hardened dunes.
L296: place the verb "used" after "SFM approach"
L310: I think here you mean that 2 dunes were present, not sastrugi.
L325: replace the term "sastrugi" to "bedforms"